# General E(2) - Equivariant Steerable CNNs

**Maurice Weiler**[*]
University of Amsterdam, QUVA Lab
m.weiler@uva.nl

**Gabriele Cesa**[*†]
University of Amsterdam
cesa.gabriele@gmail.com

## Abstract

The big empirical success of group equivariant networks has led in recent years to the sprouting of a great variety of equivariant network architectures. A particular focus has thereby been on rotation and reflection equivariant CNNs for planar images. Here we give a general description of E(2)-equivariant convolutions in the framework of *Steerable CNNs*. The theory of Steerable CNNs thereby yields constraints on the convolution kernels which depend on group representations describing the transformation laws of feature spaces. We show that these constraints for arbitrary group representations can be reduced to constraints under irreducible representations. A general solution of the kernel space constraint is given for arbitrary representations of the Euclidean group E(2) and its subgroups. We implement a wide range of previously proposed and entirely new equivariant network architectures and extensively compare their performances. E(2)-steerable convolutions are further shown to yield remarkable gains on CIFAR-10, CIFAR-100 and STL-10 when used as drop in replacement for non-equivariant convolutions.

## 1 Introduction

The equivariance of neural networks under symmetry group actions has in the recent years proven to be a fruitful prior in network design. By guaranteeing a desired transformation behavior of convolutional features under transformations of the network input, equivariant networks achieve improved generalization capabilities and sample complexities compared to their non-equivariant counterparts. Due to their great practical relevance, a big pool of rotation- and reflection- equivariant models for planar images has been proposed by now. Unfortunately, an empirical survey, reproducing and comparing all these different approaches, is still missing.

An important step in this direction is given by the theory of *Steerable CNNs* [1, 2, 3, 4, 5] which defines a very general notion of equivariant convolutions on homogeneous spaces. In particular, steerable CNNs describe E(2)-equivariant (i.e. rotation- and reflection-equivariant) convolutions on the image plane $\mathbb{R}^2$. The feature spaces of steerable CNNs are thereby defined as spaces of *feature fields*, characterized by a group representation which determines their transformation behavior under transformations of the input. In order to preserve the specified transformation law of feature spaces, the convolutional kernels are subject to a linear constraint, depending on the corresponding group representations. While this constraint has been solved for specific groups and representations [1, 2], no general solution strategy has been proposed so far. In this work we give a general strategy which reduces the solution of the kernel space constraint under arbitrary representations to much simpler constraints under single, *irreducible* representations.

Specifically for the Euclidean group E(2) and its subgroups, we give a general solution of this kernel space constraint. As a result, we are able to implement a wide range of equivariant models, covering regular GCNNs [6, 7, 8, 9, 10, 11], classical Steerable CNNs [1], Harmonic Networks [12], gated Harmonic Networks [2], Vector Field Networks [13], Scattering Transforms [14, 15, 16, 17, 18] and entirely new architectures, in one unified framework. In addition, we are able to build hybrid models, mixing different field types (representations) of these networks both over layers and within layers.

We further propose a group restriction operation, allowing for network architectures which are decreasingly equivariant with depth. This is useful e.g. for natural images which show low level features like edges in arbitrary orientations but carry a sense of preferred orientation globally. An adaptive level of equivariance accounts for the resulting loss of symmetry in the hierarchy of features.

Since the theory of steerable CNNs does not give a preference for any choice of group representation or equivariant nonlinearity, we run an extensive benchmark study, comparing different equivariance groups, representations and nonlinearities. We do so on MNIST 12k, rotated MNIST $SO(2)$ and reflected and rotated MNIST $O(2)$ to investigate the influence of the presence or absence of certain symmetries in the dataset. A drop in replacement of our equivariant convolutional layers is shown to yield significant gains over non-equivariant baselines on CIFAR10, CIFAR100 and STL-10.

Beyond the applications presented in this paper, our contributions are of relevance for general steerable CNNs on homogeneous spaces [3, 4] and gauge equivariant CNNs on manifolds [5] since these models obey the same kind of kernel constraints. More specifically, 2-dimensional manifolds, endowed with an orthogonal structure group $O(2)$ (or subgroups thereof), necessitate *exactly* the kernel constraints solved in this paper. Our results can therefore readily be transferred to e.g. spherical CNNs [19, 5, 20, 21, 22, 23] or more general models of geometric deep learning [24, 25, 26, 27].

## 2 General E(2) - Equivariant Steerable CNNs

Convolutional neural networks process images by extracting a hierarchy of feature maps from a given input signal. The convolutional weight sharing ensures the inference to be translation-equivariant which means that a translated input signal results in a corresponding translation of the feature maps. However, vanilla CNNs leave the transformation behavior of feature maps under more general transformations, e.g. rotations and reflections, undefined. In this work we devise a general framework for convolutional networks which are equivariant under the Euclidean group $E(2)$, that is, under isometries of the plane $\mathbb{R}^2$. We work in the framework of steerable CNNs [1, 2, 3, 4, 5] which provides a quite general theory for equivariant CNNs on homogeneous spaces, including Euclidean spaces $\mathbb{R}^d$ as a specific instance. Sections 2.2 and 2.3 briefly review the theory of Euclidean steerable CNNs as described in [2]. The following subsections explain our main contributions: a decomposition of the kernel space constraint into irreducible subspaces (2.4), their solution for $E(2)$ and subgroups (2.5), an overview on the group representations used to steer features, their admissible nonlinearities and their use in related work (2.6), the group restriction operation (2.7) and implementation details (2.8).

### 2.1 Isometries of the Euclidean plane $\mathbb{R}^2$

The Euclidean group $E(2)$ is the group of isometries of the plane $\mathbb{R}^2$, consisting of translations, rotations and reflections. Characteristic patterns in images often occur at arbitrary positions and in arbitrary orientations. The Euclidean group therefore models an important factor of variation of image features. This is especially true for images without a preferred global orientation like satellite imagery or biomedical images but often also applies to low level features of globally oriented images.

One can view the Euclidean group as being constructed from the translation group $(\mathbb{R}^2, +)$ and the orthogonal group $O(2) = \{O \in \mathbb{R}^{2\times2} \mid O^T O = \mathrm{id}_{2\times2}\}$ via the semidirect product operation as $E(2) \cong (\mathbb{R}^2, +) \rtimes O(2)$. The orthogonal group thereby contains all operations leaving the origin invariant, i.e. continuous rotations and reflections. In order to allow for different levels of equivariance and to cover a wide spectrum of related work we consider subgroups of the Euclidean group of the form $(\mathbb{R}^2, +) \rtimes G$, defined by subgroups $G \leq O(2)$. Specifically, $G$ could be either the special orthogonal group $SO(2)$, the group $(\{\pm1\}, *)$ of the reflections along a given axis, the cyclic groups $C_N$, the dihedral groups $D_N$ or the orthogonal group $O(2)$ itself. While $SO(2)$ describes continuous rotations (without reflections), $C_N$ and $D_N$ contain $N$ discrete rotations by angles multiple of $\frac{2\pi}{N}$ and, in the case of $D_N$, reflections. $C_N$ and $D_N$ are therefore discrete subgroups of order $N$ and $2N$, respectively. For an overview over the groups and their interrelations see Table 6 in the Appendix.

Since the groups $(\mathbb{R}^2, +) \rtimes G$ are semidirect products, one can uniquely decompose any of their elements into a product $tg$ where $t \in (\mathbb{R}^2, +)$ and $g \in G$ [3] which we will do in the rest of the paper.

### 2.2 E(2) - steerable feature fields

Steerable CNNs define feature spaces as spaces of *steerable feature fields* $f : \mathbb{R}^2 \to \mathbb{R}^c$ which associate a $c$-dimensional feature vector $f(x) \in \mathbb{R}^c$ to each point $x$ of a base space, in our case the

plane $\mathbb{R}^2$. In contrast to vanilla CNNs, the feature fields of steerable CNNs are associated with a transformation law which specifies their transformation under actions of E(2) (or subgroups) and therefore endows features with a notion of *orientation*. Formally, a feature vector $f(x)$ encodes the coefficients of a coordinate independent geometric feature relative to a choice of reference frame or, equivalently, image orientation (see Appendix A).

An important example are *scalar* feature fields $s : \mathbb{R}^2 \to \mathbb{R}$, describing for instance gray-scale images or temperature fields. The Euclidean group acts on scalar fields by moving each pixel to a new position, that is, $s(x) \mapsto s\left((tg)^{-1}x\right) = s\left(g^{-1}(x-t)\right)$ for some $tg \in (\mathbb{R}^2, +) \rtimes G$; see Figure 1, left. *Vector* fields $v : \mathbb{R}^2 \to \mathbb{R}^2$, like optical flow or gradient images, on the other hand transform as $v(x) \mapsto g \cdot v\left(g^{-1}(x-t)\right)$. In contrast to the case of scalar fields, each vector is therefore not only moved to a new position but additionally changes its orientation via the action of $g \in G$; see Figure 1, right.

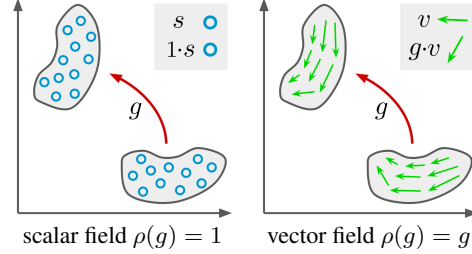

scalar field $\rho(g) = 1$     vector field $\rho(g) = g$

Figure 1: Transformation behavior of $\rho$-fields.

The transformation law of a general feature field $f : \mathbb{R}^2 \to \mathbb{R}^c$ is fully characterized by its *type* $\rho$. Here $\rho : G \mapsto \mathrm{GL}(\mathbb{R}^c)$ is a group representation, specifying how the $c$ channels of each feature vector $f(x)$ mix under transformations. A representation satisfies $\rho(g\tilde{g}) = \rho(g)\rho(\tilde{g})$ and therefore models the group multiplication $g\tilde{g}$ as multiplication of $c \times c$ matrices $\rho(g)$ and $\rho(\tilde{g})$. More specifically, a $\rho$-field transforms under the *induced representation*[1][2] $\left[\mathrm{Ind}_G^{(\mathbb{R}^2,+) \rtimes G}\rho\right]$ of $(\mathbb{R}^2, +) \rtimes G$ as

$$f(x) \;\mapsto\; \left(\left[\mathrm{Ind}_G^{(\mathbb{R}^2,+)\rtimes G}\rho\right](tg) \cdot f\right)(x) \;:=\; \rho(g) \cdot f\left(g^{-1}(x-t)\right). \tag{1}$$

As in the examples above, it transforms feature fields by moving the feature vectors from $g^{-1}(x-t)$ to a new position $x$ and acting on them via $\rho(g)$. We thus find scalar fields to correspond to the *trivial representation* $\rho(g) = 1 \;\forall g \in G$ which reflects that the scalar values do not change when being moved. Similarly, a vector field corresponds to the standard representation $\rho(g) = g$ of $G$.

In analogy to the feature spaces of vanilla CNNs comprising multiple channels, the feature spaces of steerable CNNs consist of multiple feature fields $f_i : \mathbb{R}^2 \to \mathbb{R}^{c_i}$, each of which is associated with its own *type* $\rho_i : G \to \mathrm{GL}(\mathbb{R}^{c_i})$. A stack $f = \bigoplus_i f_i$ of feature fields is then defined to be concatenated from the individual feature fields and transforms under the direct sum $\rho = \bigoplus_i \rho_i$ of the individual representations. A common example for a stack of feature fields are RGB images $f : \mathbb{R}^2 \to \mathbb{R}^3$. Since the color channels transform independently under rotations we identify them as three independent scalar fields. The stacked field representation is thus given by the direct sum $\bigoplus_{i=1}^3 1 = \mathrm{id}_{3\times 3}$ of three trivial representations. While the input and output types of steerable CNNs are given by the learning task, the user needs to specify the types $\rho_i$ of intermediate feature fields as hyperparameters, similar to the choice of channels for vanilla CNNs. We discuss different choices of representations in Section 2.6 and investigate them empirically in Section 3.1.

## 2.3   E(2)-steerable convolutions

In order to preserve the transformation law of steerable feature spaces, each network layer is required to be equivariant under the group actions. As proven for Euclidean groups in [2], the most general *equivariant linear map* between steerable feature spaces, transforming under $\rho_{\mathrm{in}}$ and $\rho_{\mathrm{out}}$, is given by *convolutions* with $G$-steerable kernels[3] $k : \mathbb{R}^2 \to \mathbb{R}^{c_{\mathrm{out}} \times c_{\mathrm{in}}}$, satisfying a kernel constraint

$$k(gx) \;=\; \rho_{\mathrm{out}}(g)k(x)\rho_{\mathrm{in}}(g^{-1}) \quad \forall g \in G,\; x \in \mathbb{R}^2 \,. \tag{2}$$

Intuitively, this constraint determines the form of the kernel in transformed coordinates $gx$ in terms of the kernel in non-transformed coordinates $x$ and thus its response to transformed input fields. It ensures that the output feature fields transform as specified by $\mathrm{Ind}\,\rho_{\mathrm{out}}$ when the input fields are being transformed by $\mathrm{Ind}\,\rho_{\mathrm{in}}$; see Appendix G.1 for a proof.

Since the kernel constraint is linear, its solutions form a linear subspace of the vector space of unconstrained kernels considered in conventional CNNs. It is thus sufficient to solve for a basis of the $G$-steerable kernel space in terms of which the equivariant convolutions can be parameterized. The lower dimensionality of the restricted kernel space enhances the parameter efficiency of steerable CNNs over conventional CNNs similarly to the increased parameter efficiency of CNNs over MLPs.

## 2.4 Irrep decomposition of the kernel constraint

The kernel constraint (2) in principle needs to be solved individually for each pair of input and output types $\rho_{\text{in}}$ and $\rho_{\text{out}}$ to be used in the network. Here we show how the solution of the kernel constraint for arbitrary representations can be reduced to much simpler constraints under *irreducible representations* (irreps). Our approach relies on the fact that any representation of a finite or compact group decomposes under a change of basis into a direct sum of irreps, each corresponding to an invariant subspace of the representation space $\mathbb{R}^c$ on which $\rho$ acts. Denoting the change of basis by $Q$, this means that one can always write $\rho = Q^{-1} \left[ \bigoplus_{i \in I} \psi_i \right] Q$ where $\psi_i$ are the irreducible representations of $G$ and the index set $I$ encodes the types and multiplicities of irreps present in $\rho$. A decomposition can be found by exploiting basic results of character theory and linear algebra [28].

The decomposition of $\rho_{\text{in}}$ and $\rho_{\text{out}}$ in the kernel constraint (2) leads to

$$k(gx) = Q_{\text{out}}^{-1} \left[ \bigoplus_{i \in I_{\text{out}}} \psi_i(g) \right] Q_{\text{out}} \, k(x) \, Q_{\text{in}}^{-1} \left[ \bigoplus_{j \in I_{\text{in}}} \psi_j^{-1}(g) \right] Q_{\text{in}} \qquad \forall g \in G, \, x \in \mathbb{R}^2,$$

which, defining a kernel relative to the irrep bases as $\kappa := Q_{\text{out}} k Q_{\text{in}}^{-1}$, implies

$$\kappa(gx) = \left[ \bigoplus_{i \in I_{\text{out}}} \psi_i(g) \right] \kappa(x) \left[ \bigoplus_{j \in I_{\text{in}}} \psi_j^{-1}(g) \right] \qquad \forall g \in G, \, x \in \mathbb{R}^2.$$

The left and right multiplication with a direct sum of irreps reveals that the constraint decomposes into *independent* constraints

$$\kappa^{ij}(gx) = \psi_i(g) \, \kappa^{ij}(x) \, \psi_j^{-1}(g) \qquad \forall g \in G, \, x \in \mathbb{R}^2 \quad \text{where } i \in I_{\text{out}}, \, j \in I_{\text{in}} \qquad (3)$$

on blocks $\kappa^{ij}$ in $\kappa$ corresponding to invariant subspaces of the full space of equivariant kernels; see Appendix H for a visualization. In order to solve for a basis of equivariant kernels satisfying the original constraint (2), it is therefore sufficient to solve the irrep constraints (3) to obtain bases for each block, revert the change of basis and take the union over different blocks. Specifically, given $d_{ij}$-dimensional bases $\left\{ \kappa_1^{ij}, \cdots, \kappa_{d_{ij}}^{ij} \right\}$ for the blocks $\kappa^{ij}$ of $\kappa$, we get a $d = \sum_{ij} d_{ij}$-dimensional basis

$$\{k_1, \cdots, k_d\} := \bigcup_{i \in I_{\text{out}}} \bigcup_{j \in I_{\text{in}}} \left\{ Q_{\text{out}}^{-1} \, \overline{\kappa}_1^{ij} Q_{\text{in}}, \, \cdots, \, Q_{\text{out}}^{-1} \, \overline{\kappa}_{d_{ij}}^{ij} Q_{\text{in}} \right\} \qquad (4)$$

of solutions of (2). Here $\overline{\kappa}^{ij}$ denotes a block $\kappa^{ij}$ being filled at the corresponding location of a matrix of the shape of $\kappa$ with all other blocks being set to zero; see Appendix H. The completeness of the basis found this way is guaranteed by construction if the bases for each block $ij$ are complete. Note that while this approach shares some basic ideas with the solution strategy proposed in [2], it is computationally more efficient for large representations; see Appendix J. We want to emphasize that this strategy for reducing the kernel constraint to irreducible representations is not restricted to subgroups of $O(2)$ but applies to steerable CNNs in general.

## 2.5 General solution of the kernel constraint for O(2) and subgroups

In order to build isometry-equivariant CNNs on $\mathbb{R}^2$ we need to solve the irrep constraints (3) for the specific case of $G$ being $O(2)$ or one of its subgroups. For this purpose note that the action of $G$ on $\mathbb{R}^2$ is norm-preserving, that is, $|g.x| = |x| \; \forall g \in G, \, x \in \mathbb{R}^2$. The constraints (2) and (3) therefore only restrict the *angular parts* of the kernels but leave their radial parts free. Since furthermore all irreps of $G$ correspond to one unique *angular frequency* (see Appendix I.2), it is convenient to expand the kernel w.l.o.g. in terms of an (angular) Fourier series

$$\kappa_{\alpha\beta}^{ij}\big(x(r,\phi)\big) = A_{\alpha\beta,0}(r) + \sum_{\mu=1}^{\infty} \left[ A_{\alpha\beta,\mu}(r) \cos(\mu\phi) + B_{\alpha\beta,\mu}(r) \sin(\mu\phi) \right] \qquad (5)$$

with real-valued, radially dependent coefficients $A_{\alpha\beta,\mu} : \mathbb{R}^+ \to \mathbb{R}$ and $B_{\alpha\beta,\mu} : \mathbb{R}^+ \to \mathbb{R}$ for each matrix entry $\kappa_{\alpha\beta}^{ij}$ of block $\kappa^{ij}$. By inserting this expansion into the irrep constraints (3) and projecting on individual harmonics we obtain constraints on the Fourier coefficients, forcing most of them to be

zero. The vector spaces of $G$-steerable kernel blocks $\kappa^{ij}$ satisfying the irrep constraints (3) are then parameterized in terms of the remaining Fourier coefficients. The completeness of this basis follows immediately from the completeness of the Fourier basis. Similar approaches have been followed in simpler settings for the cases of $C_N$ in [7], SO(2) in [12] and SO(3) in [2].

The resulting bases for the angular parts of kernels for each pair of irreducible representations of O(2) are shown in Table 1. It turns out that each basis element is harmonic and associated to one unique angular frequency. Appendix I gives an explicit derivation and the resulting bases for all possible pairs of irreps for all groups $G \leq$ O(2) following the strategy presented in this section. The analytical solutions for SO(2), $(\{\pm 1\}, *)$, $C_N$ and $D_N$ are found in Tables 8, 10, 11 and 12. Since these groups are subgroups of O(2), they enforce a weaker kernel constraint as compared to O(2). As a result, the bases for $G <$ O(2) are higher dimensional, i.e. they allow for a wider range of kernels. A higher level of equivariance therefore leads simultaneously to a guaranteed behavior of the inference process under transformations and on the other hand to an improved parameter efficiency.

| $\psi_i$ \ $\psi_j$ | trivial | sign-flip | frequency $n \in \mathbb{N}^+$ | |
|---|---|---|---|---|
| trivial | $\begin{bmatrix} 1 \end{bmatrix}$ | $\varnothing$ | $\begin{bmatrix} \sin(n\phi), & -\cos(n\phi) \end{bmatrix}$ | |
| sign-flip | $\varnothing$ | $\begin{bmatrix} 1 \end{bmatrix}$ | $\begin{bmatrix} \cos(n\phi), & \sin(n\phi) \end{bmatrix}$ | |
| frequency $m \in \mathbb{N}^+$ | $\begin{bmatrix} \sin(m\phi) \\ -\cos(m\phi) \end{bmatrix}$ | $\begin{bmatrix} \cos(m\phi) \\ \sin(m\phi) \end{bmatrix}$ | $\begin{bmatrix} \cos((m{-}n)\phi) & -\sin((m{-}n)\phi) \\ \sin((m{-}n)\phi) & \cos((m{-}n)\phi) \end{bmatrix}$, | $\begin{bmatrix} \cos((m{+}n)\phi) & \sin((m{+}n)\phi) \\ \sin((m{+}n)\phi) & -\cos((m{+}n)\phi) \end{bmatrix}$ |

Table 1: Bases for the angular parts of O(2)-steerable kernels satisfying the irrep constraint (3) for different pairs of input field irreps $\psi_j$ and output field irreps $\psi_i$. The different types of irreps are explained in Appendix I.2.

## 2.6 Group representations and nonlinearities

A question which so far has been left open is which field types, i.e. which representations $\rho$ of $G$, should be used in practice. Considering only the convolution operation with $G$-steerable kernels for the moment, it turns out that any change of basis $P$ to an *equivalent representation* $\widetilde{\rho} := P^{-1}\rho P$ is irrelevant. To see this, consider the irrep decomposition $\rho = Q^{-1} \left[ \bigoplus_{i \in I} \psi_i \right] Q$ used in the solution of the kernel constraint to obtain a basis $\{k_i\}_{i=1}^d$ of $G$-steerable kernels as defined by Eq. (4). Any equivalent representation will decompose into $\widetilde{\rho} = \widetilde{Q}^{-1} \left[ \bigoplus_{i \in I} \psi_i \right] \widetilde{Q}$ with $\widetilde{Q} = QP$ for some $P$ and therefore result in a kernel basis $\{P_{\text{out}}^{-1} k_i P_{\text{in}}\}_{i=1}^d$ which entirely negates changes of bases between equivalent representations. It would therefore w.l.o.g. suffice to consider direct sums of irreps $\rho = \bigoplus_{i \in I} \psi_i$ as representations only, reducing the question on which representations to choose to the question on which types and multiplicities of irreps to use.

In practice, however, convolution layers are interleaved with other operations which are sensitive to specific choices of representations. In particular, nonlinearity layers are required to be equivariant under the action of specific representations. The choice of group representations in steerable CNNs therefore restricts the range of admissible nonlinearities, or, conversely, a choice of nonlinearity allows only for certain representations. In the following we review prominent choices of representations found in the literature in conjunction with their compatible nonlinearities.

All equivariant nonlinearities considered here act spatially localized, that is, on each feature vector $f(x) \in \mathbb{R}^{c_{\text{in}}}$ for all $x \in \mathbb{R}^2$ individually. They might produce different types of output fields $\rho_{\text{out}} : G \to \text{GL}(\mathbb{R}^{c_{\text{out}}})$, that is, $\sigma : \mathbb{R}^{c_{\text{in}}} \to \mathbb{R}^{c_{\text{out}}}$, $f(x) \mapsto \sigma(f(x))$. As proven in Appendix G.2, it is sufficient to require the equivariance of $\sigma$ under the actions of $\rho_{\text{in}}$ and $\rho_{\text{out}}$, i.e. $\sigma \circ \rho_{\text{in}}(g) = \rho_{\text{out}}(g) \circ \sigma \; \forall g \in G$, for the nonlinearities to be equivariant under the action of induced representations when being applied to a whole feature field as $\sigma(f)(x) := \sigma(f(x))$.

A general class of representations are *unitary representations* which preserve the norm of their representation space, that is, they satisfy $|\rho_{\text{unitary}}(g)f(x)| = |f(x)| \; \forall \; g \in G$. As proven in Appendix G.2.2, nonlinearities which solely act on the *norm* of feature vectors but preserve their orientation are equivariant w.r.t. unitary representations. They can in general be decomposed in $\sigma_{\text{norm}} : \mathbb{R}^c \to \mathbb{R}^c$, $f(x) \mapsto \eta(|f(x)|) \frac{f(x)}{|f(x)|}$ for some nonlinear function $\eta : \mathbb{R}_{\geq 0} \to \mathbb{R}_{\geq 0}$ acting on the norm of feature vectors. *Norm-ReLUs*, defined by $\eta(|f(x)|) = \text{ReLU}(|f(x)| - b)$ where $b \in \mathbb{R}^+$ is a learned bias, were used in [12, 2]. In [29], the authors consider *squashing nonlinearities* $\eta(|f(x)|) = \frac{|f(x)|^2}{|f(x)|^2 + 1}$. *Gated nonlinearities* were proposed in [2] as conditional version of norm

nonlinearities. They act by scaling the norm of a feature field by learned sigmoid gates $\frac{1}{1+e^{-s(x)}}$, parameterized by a scalar feature field $s$. All representations considered in this paper are unitary such that their fields can be acted on by norm-nonlinearities. This applies specifically also to all *irreducible representations* $\psi_i$ of $G \leq \mathrm{O}(2)$ which are discussed in detail in Section I.2.

A common choice of representations of *finite* groups like $\mathrm{C}_N$ and $\mathrm{D}_N$ are *regular representations*. Their representation space $\mathbb{R}^{|G|}$ has dimensionality equal to the order of the group, e.g. $\mathbb{R}^N$ for $\mathrm{C}_N$ and $\mathbb{R}^{2N}$ for $\mathrm{D}_N$. The action of the regular representation is defined by assigning each axis $e_g$ of $\mathbb{R}^{|G|}$ to a group element $g \in G$ and permuting the axes according to $\rho_{\mathrm{reg}}^G(\tilde{g})e_g := e_{\tilde{g}g}$. Since this action is just permuting channels of $\rho_{\mathrm{reg}}^G$-fields, it commutes with pointwise nonlinearities like ReLU; a proof is given in Appendix G.2.3. While regular steerable CNNs were empirically found to perform very well, they lead to high dimensional feature spaces with each individual field consuming $|G|$ channels. Regular steerable CNNs were investigated for planar images in [6, 7, 8, 9, 10, 17, 18, 30], for spherical CNNs in [19, 5] and for volumetric convolutions in [31, 32]. Further, the translation of feature maps of conventional CNNs can be viewed as action of the regular representation of the translation group.

Closely related to regular representations are *quotient representations*. Instead of permuting $|G|$ channels indexed by $G$, they permute $|G|/|H|$ channels indexed by cosets $gH$ in the quotient space $G/H$ of a subgroup $H \leq G$. Specifically, they act on axes $e_{gH}$ of $\mathbb{R}^{|G|/|H|}$ as defined by $\rho_{\mathrm{quot}}^{G/H}(\tilde{g})e_{gH} := e_{\tilde{g}gH}$. As permutation representations, quotient representations allow for pointwise nonlinearities; see Appendix G.2.3. Quotient representations were considered in [1, 11].

Regular and quotient fields can furthermore be acted on by nonlinear pooling operators. Via a *group pooling* or projection operation $\max : \mathbb{R}^c \to \mathbb{R}, \ f(x) \to \max(f(x))$ the works [6, 7, 9, 32, 31] extract the maximum value of a regular or quotient field. The invariance of the maximum operation implies that the resulting features form *scalar fields*. Since group pooling operations discard information on the feature orientations entirely, vector field nonlinearities $\sigma_{\mathrm{vect}} : \mathbb{R}^N \to \mathbb{R}^2$ for regular representations of $\mathrm{C}_N$ were proposed in [13]. Vector field nonlinearities do not only keep the maximum response $\max(f(x))$ but also its index $\arg\max(f(x))$. This index corresponds to a rotation angle $\theta = \frac{2\pi}{N}\arg\max(f(x))$ which is used to define a vector field with elements $v(x) = \max(f(x))(\cos(\theta), \sin(\theta))^T$. The equivariance of this operation is proven in G.2.4.

## 2.7 Group restrictions and inductions

The key idea of equivariant networks is to exploit symmetries in the distribution of characteristic patterns in signals. The level of symmetry present in data might thereby vary over length scales. For instance, natural images typically show small features like edges in arbitrary orientations. On a larger length scale, however, the rotational symmetry is broken as manifested in visual patterns exclusively appearing upright but still in different reflections. Each individual layer of a convolutional network should therefore be adapted to the symmetries present in the length scale of its fields of view.

A loss of symmetry can be implemented by *restricting* the equivariance at a certain depth to a subgroup $(\mathbb{R}^2, +) \rtimes H \leq (\mathbb{R}^2, +) \rtimes G$, e.g. from rotations and reflections $G = \mathrm{O}(2)$ to mere reflections $H = (\{\pm 1\}, *)$ in the example above. This requires the feature fields produced by a layer with a higher level of equivariance to be reinterpreted in the following layer as fields transforming under a subgroup. Specifically, a $\rho$-field, transforming according to $\rho : G \to \mathrm{GL}(\mathbb{R}^c)$, needs to be reinterpreted as a $\tilde{\rho}$-field, where $\tilde{\rho} : H \to \mathrm{GL}(\mathbb{R}^c)$ is a representation of the subgroup $H \leq G$. This is naturally achieved by using the *restricted representation* $\tilde{\rho} := \mathrm{Res}_H^G(\rho) : H \to \mathrm{GL}(\mathbb{R}^c), \ h \mapsto \rho(h)$, defined by restricting the domain of $\rho$ to $H$. Since a subsequent $H$-steerable convolution layers can map fields of arbitrary representations we can readily process the resulting $\mathrm{Res}_H^G(\rho)$-field further.

## 2.8 Implementation details

$\mathrm{E}(2)$-steerable CNNs rely on convolutions with $\mathrm{O}(2)$-steerable kernels. Our implementation therefore requires the precomputation of steerable kernel bases according to the analytical solutions in Eq. (4) with arbitrary radial parts. Since the kernel basis is sampled on a discrete pixel grid, care has to be taken that no aliasing artifacts occur. During runtime, the sampled basis is expanded using learned weights. The resulting $G$-steerable kernel is then being used in a standard convolution routine. For more details we refer to Appendix C. Our implementation is provided as a PyTorch extension which is available at `https://github.com/QUVA-Lab/e2cnn`.

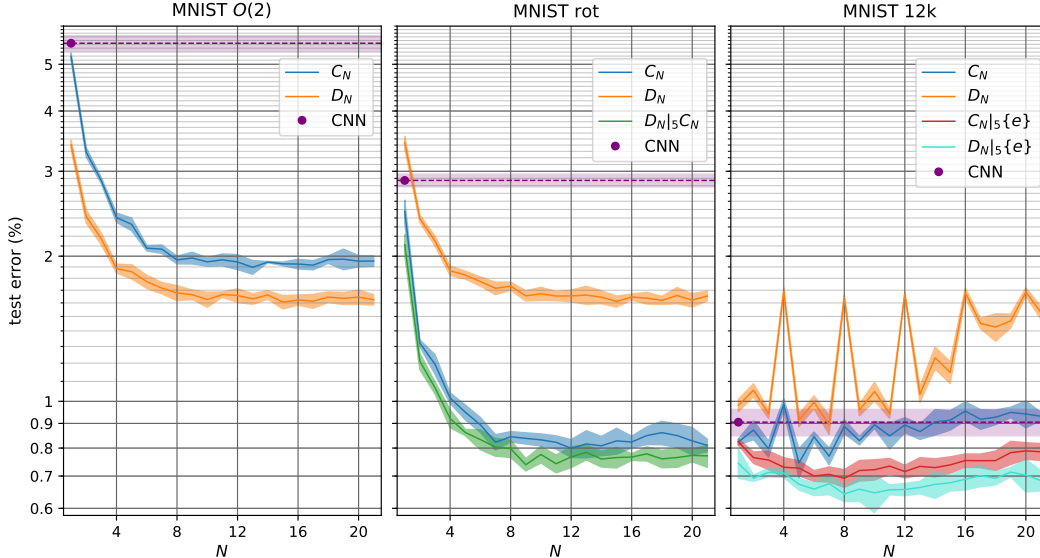

Figure 2: Test errors of $C_N$ and $D_N$ regular steerable CNNs for different orders $N$ for all three MNIST variants. *Left:* All equivariant models improve upon the non-equivariant baseline on MNIST $O(2)$. The error decreases before saturating at around 8 orientations. Since the dataset contains reflected digits, the $D_N$-equivariant models perform better than their $C_N$ counterparts. *Middle:* Since the intraclass variability of MNIST rot is reduced, the performances of the $C_N$ model and the baseline improve. In contrast, the $D_N$ models are invariant to reflections such that they can't distinguish between MNIST $O(2)$ and MNIST rot. For $N = 1$ this leads to a worse performance than that of the baseline. Restricted dihedral models, denoted by $D_N|_5 C_N$, make use of the local reflectional symmetries but are not globally invariant. This makes them perform better than the $C_N$ models. *Right:* On MNIST 12k the globally invariant models $C_N$ and $D_N$ don't yield better results than the baseline, however, the restricted (i.e. non-invariant) models $C_N|_5\{e\}$ and $D_N|_5\{e\}$ do. For more details see Appendix D.1.

## 3 Experiments

Since the framework of general $E(2)$-equivariant steerable CNNs supports many choices of groups, representations and nonlinearities, we first run an extensive benchmark study over the space of supported models in Section 3.1. The insights from these benchmark experiments are then applied to classify CIFAR and STL-10 images in Sections 3.2 and 3.3. All of our experiments are found in a dedicated repository at `https://github.com/gabri95/e2cnn_experiments`.

### 3.1 Model benchmarking on transformed MNIST datasets

We first perform a comprehensive benchmarking to compare the impact of the different design choices covered in this work. All benchmarked models are evaluated on three different versions of the MNIST dataset, each containing 12000 training and 50000 test images. The digits in the three variations MNIST 12k, MNIST rot and MNIST $O(2)$ are left untransformed, are rotated and are rotated and reflected, respectively. These datasets allow us to study the benefit from different levels of $G$-steerability in the presence or absence of certain symmetries. In order to not disadvantage models with lower levels of equivariance, we train all models using data augmentation by the transformations present in the corresponding dataset.

**Representation and nonlinearity benchmarking:** Table 7 in the Appendix shows the test errors of 57 different models on the three MNIST variants. The first four columns state the equivariance groups, representations, nonlinearities and invariant maps which distinguish the models. The invariant maps of each model are applied after the last convolution layer to produce $G$-invariant features. Appendix D.1 compares and analyzes all results in detail. In particular, it discusses regular and quotient models, group pooling and vector field networks, as well as $SO(2)$ and $O(2)$-equivariant irrep models. The latter employ new kinds of gated-nonlinearities and norm-nonlinearities and, in the case of $O(2)$, introduce induced representations as new feature types. The results of all models whose feature fields transform according to regular representations, are summarized in Figure 2.

**Group restriction:** All transformed MNIST datasets show local rotational and reflectional symmetries but differ in the level of symmetry present at the global scale. While $D_N$ and $O(2)$-equivariant

| restriction depth | MNIST rot | | MNIST 12k | | | |
|---|---|---|---|---|---|---|
| | group | test error (%) | group | test error (%) | group | test error (%) |
| (0) | $C_{16}$ | $0.82 \pm 0.02$ | $\{e\}$ | $0.82 \pm 0.01$ | $\{e\}$ | $0.82 \pm 0.01$ |
| 1 | | $0.86 \pm 0.05$ | | $0.79 \pm 0.03$ | | $0.80 \pm 0.03$ |
| 2 | | $0.82 \pm 0.03$ | | $0.74 \pm 0.03$ | | $0.77 \pm 0.03$ |
| 3 | $D_{16}, C_{16}$ | $0.77 \pm 0.03$ | $D_{16}, \{e\}$ | $0.73 \pm 0.03$ | $C_{16}, \{e\}$ | $0.76 \pm 0.03$ |
| 4 | | $0.79 \pm 0.03$ | | $0.72 \pm 0.02$ | | $0.77 \pm 0.03$ |
| 5 | | $0.78 \pm 0.04$ | | $0.68 \pm 0.04$ | | $0.75 \pm 0.02$ |
| no restriction | $D_{16}$ | $1.65 \pm 0.02$ | $D_{16}$ | $1.68 \pm 0.04$ | $C_{16}$ | $0.95 \pm 0.04$ |

Table 2: Effect of the group restriction operation at different depths of the network on MNIST rot and MNIST 12k. All restricted models perform better than non-restricted, and hence globally invariant, models.

| model | group | representation | test error (%) |
|---|---|---|---|
| [6] | $C_4$ | regular/scalar | $3.21 \pm 0.0012$ |
| [6] | $C_4$ | regular | $2.28 \pm 0.0004$ |
| [12] | $SO(2)$ | irreducible | $1.69$ |
| [33] | - | - | $1.2$ |
| [13] | $C_{17}$ | regular/vector | $1.09$ |
| Ours | $C_{16}$ | regular | $0.716 \pm 0.028$ |
| [7] | $C_{16}$ | regular | $0.714 \pm 0.022$ |
| Ours | $C_{16}$ | quotient | $0.705 \pm 0.025$ |
| Ours | $D_{16}|_5 C_{16}$ | regular | $0.682 \pm 0.022$ |

Table 3: Final runs on MNIST rot

| model | | CIFAR-10 | CIFAR-100 |
|---|---|---|---|
| wrn28/10 | [34] | $3.87$ | $18.80$ |
| wrn28/10 | $D_1 D_1 D_1$ | $3.36 \pm 0.08$ | $17.97 \pm 0.11$ |
| wrn28/10* | $D_8 D_4 D_1$ | $3.28 \pm 0.10$ | $17.42 \pm 0.33$ |
| wrn28/10 | $C_8 C_4 C_1$ | $3.20 \pm 0.04$ | $16.47 \pm 0.22$ |
| wrn28/10 | $D_8 D_4 D_1$ | $3.13 \pm 0.17$ | $16.76 \pm 0.40$ |
| wrn28/10 | $D_8 D_4 D_4$ | $2.91 \pm 0.13$ | $16.22 \pm 0.31$ |
| wrn28/10 | [35]        AA | $2.6 \pm 0.1$ | $17.1 \pm 0.3$ |
| wrn28/10* | $D_8 D_4 D_1$ AA | $2.39 \pm 0.11$ | $15.55 \pm 0.13$ |
| wrn28/10 | $D_8 D_4 D_1$ AA | $2.05 \pm 0.03$ | $14.30 \pm 0.09$ |

Table 4: Test errors on CIFAR (AA=autoaugment)

models exploit these local symmetries, their global invariance leads to a considerable loss of information. On the other hand, models which are equivariant to the symmetries present at the global scale of the dataset only are not able to generalize over all local symmetries. The proposed group restriction operation allows for models which are locally equivariant but are globally invariant only to the level of symmetry present in the data. Table 2 reports the results of models which are restricted at different depths. The overall trend is that a restriction at later stages of the model improves the performance. All restricted models perform significantly better than the invariant models. Figure 2 shows that this behavior is consistent for different orders $N$.

**Convergence rate:** In our experiments we find that steerable CNNs converge significantly faster than non-equivariant CNNs. Figure 4 in the Appendix shows this behavior for regular $C_N$-steerable CNNs in comparison to a vanilla CNN. The rate of convergence thereby increases with the order $N$ and, as already observed in Figure 2, saturates at approximately $N = 8$. All models share about the same number of parameters. The faster convergence of equivariant networks is explained by the fact that they generalize over $G$-transformed images by design which reduces the amount of intra-class variability which they have to learn. Conversely, a conventional CNN has to learn to classify all transformed versions of an image explicitly which requires an increased batch size or more training iterations. The enhanced data efficiency of $E(2)$-steerable CNNs thus leads to a reduced training time.

**Competitive runs:** As a final experiment on MNIST rot we are replicating the regular $C_{16}$ model from [7]. It is mostly similar to the models evaluated before but is wider and adds additional fully connected layers; see Table 14 in the Appendix. As reported in Table 3, our reimplementation matches the accuracy of the original model. Replacing the regular feature fields with the quotient representations used in the benchmarking leads to slightly better results. We refer to Appendix F for more insights on the improved performance of quotient model. A further significant improvement and a new state of the art is being achieved by a $D_{16}$ model, which is restricted to $C_{16}$ in the final layer.

### 3.2 CIFAR experiments

The statistics of natural images are typically invariant under global translations and reflections but are not under global rotations. Here we investigate the benefit of $G$-steerable convolutions for such images by classifying CIFAR-10 and CIFAR-100. For this purpose we implement several $D_N$ and $C_N$-equivariant versions of WideResNet [34]. Different levels of equivariance, stated in the model specifications in Table 4, are thereby used in the three main blocks of the network. Regular representations are used throughout the whole model. For a fair comparison we scale the width of all layers such that the number of parameters of the original wrn28/10 model is preserved. We further add a small model, marked by an additional *, which has about the same number of channels

as the non-equivariant wrn28/10. All runs use the same training procedure as reported in [34] and Appendix K.3. We want to emphasize that we perform *no further hyperparameter tuning*.

The results of the $D_1 D_1 D_1$ model confirm that incorporating the global symmetries of the data yields a significant boost in accuracy. Interestingly, the $C_8 C_4 C_1$ model, which is rotation but not reflection-equivariant, achieves better results, which shows that it is worthwhile to leverage local rotational symmetries. Both symmetries are respected simultaneously by the wrn28/10 $D_8 D_4 D_1$ model. While this model performs better than the two previous ones on CIFAR-10, it surprisingly yields slightly worse result on CIFAR-100. The best results are obtained by the $D_8 D_4 D_4$ model which suggests that rotational symmetries are useful even on a larger scale. The small wrn28/10* $D_8 D_4 D_1$ model shows a remarkable gain compared to the non-equivariant wrn28/10 baseline *despite not being computationally more expensive*. To investigate whether equivariance is useful even when a powerful data augmentation policy is available, we further rerun both $D_8 D_4 D_1$ models with *AutoAugment* (AA) [35]. As without AA, both equivariant models outperform the baseline by a large margin.

### 3.3 STL-10 experiments

In order to test whether the previous results generalize to natural images of higher resolution we run experiments on STL-10 [37]. We adapt the experiments in [36] by replacing the non-equivariant convolutions of their wrn16/8 model with regular $D_N$-steerable convolutions. As in the CIFAR experiments, we adopt the training settings and hyperparameters of [36] without changes. Our four adapted models, reported in Table 5, are equivariant under either the action of $D_1$ in all blocks or the actions of $D_8$, $D_4$ and $D_1$. For both choices we build a large

| model | group | #params | test error (%) |
|---|---|---|---|
| wrn16/8 [36] | - | 11M | $12.74_{\pm 0.23}$ |
| wrn16/8* | $D_1 D_1 D_1$ | 5M | $11.05_{\pm 0.45}$ |
| wrn16/8 | $D_1 D_1 D_1$ | 10M | $11.17_{\pm 0.60}$ |
| wrn16/8* | $D_8 D_4 D_1$ | 4.2M | $10.57_{\pm 0.70}$ |
| wrn16/8 | $D_8 D_4 D_1$ | 12M | $9.80_{\pm 0.40}$ |

Table 5: Test errors of different equivariant models on the STL-10 dataset. Models with * preserve the number of channels of the baseline.

model, preserving the number of parameters of the baseline, and a small model, which preserves its number of channels and thus computational requirements. All models improve significantly over the baseline. Due to their extended equivariance, the small $D_8 D_4 D_1$ model performs better than the large $D_1 D_1 D_1$ model. In comparison to the CIFAR experiments, rotational equivariance gives a larger boost in accuracy since the higher resolution of 96px of STL-10 allows for more detailed local patterns which occur in arbitrary orientations. Appendix D.3 reports the results of a data ablation study. The results validate that the gains from incorporating equivariance are consistent over all training set sizes. More information on the training procedures is given in Appendix K.4.

## 4 Conclusions

In this work we presented a general theory of $E(2)$-equivariant steerable CNNs. By analytically solving the kernel constraint for any representation of $O(2)$ or its subgroups we were able to reproduce and compare many different models from previous work. We further proposed a group restriction operation which allows us to adapt the level of equivariance to the symmetries present on the corresponding length scale. When using $G$-steerable convolutions as drop in replacement for conventional convolution layers we obtained significant improvements on CIFAR and STL-10 without additional hyperparameter tuning. While the kernel expansion leads to a small overhead during train time, the final kernels can be stored such that during test time steerable CNNs are computationally not more expensive than conventional CNNs of the same width. Due to the enhanced parameter efficiency of equivariant models it is a common practice to adapt the model width to match the parameter cost of conventional CNNs. Our results show that even non-scaled models outperform conventional CNNs in accuracy.

We believe that equivariant CNNs will in the long term become the default choice for tasks like biomedical imaging, where symmetries are present on a global scale. The impressive results on natural images demonstrate the great potential of applying $E(2)$-steerable CNNs to more general vision tasks which involve only local symmetries. Future research still needs to investigate the wide range of design choices of steerable CNNs in more depth and collect evidence on whether our findings generalize to different settings. We hope that our library will help equivariant CNNs to be adopted by the community and facilitate further research.

### Acknowledgments

We would like to thank Taco Cohen for fruitful discussions on an efficient implementation and helpful feedback on the paper and Daniel Worrall for elaborating on the implementation of Harmonic Networks.

## Footnotes

* Equal contribution, author ordering determined by random number generator.

† This research has been conducted during an internship at QUVA lab, University of Amsterdam.

[1] Induced representations are the most general transformation laws compatible with convolutions [3, 4].

[2] Note that this simple form of the induced representation is a special case for semidirect product groups.

[3] As $k : \mathbb{R}^2 \to \mathbb{R}^{c_{\mathrm{out}} \times c_{\mathrm{in}}}$ returns a matrix of shape $(c_{\mathrm{out}}, c_{\mathrm{in}})$ for each position $x \in \mathbb{R}^2$, its discretized version can be represented by a tensor of shape $(c_{\mathrm{out}}, c_{\mathrm{in}}, X, Y)$ as usually done in deep learning frameworks.

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
