[Supplementary Material]

# General E(2) - Equivariant Steerable CNNs Appendix

## A  Local gauge equivariance of E(2)-steerable CNNs

Figure 3: Different viewpoints on transformations of signals on $\mathbb{R}^2$. *Top Left:* In our work we considered active rotations of the signal while keeping the coordinate frames fixed. *Bottom Left:* The equivalent, passive interpretation views the transformation as a global rotation of reference frames (a global gauge transformation). *Right:* Local gauge transformations rotate reference frames independently from each other. E(2)-steerable CNNs are equivariant w.r.t. both global and local gauge transformations.

The E(2)-equivariant steerable CNNs considered in this work were derived in the classical framework of steerable CNNs on Euclidean spaces $\mathbb{R}^d$ (or more general homogeneous spaces) [1, 2, 3, 4]. This formulation considers *active transformations* of signals, in our case translations, rotations and reflections of images. Specifically, an active transformation by a group element $tg \in (\mathbb{R}^2, +) \rtimes G$ moves signal values from $x$ to $g^{-1}(x-t)$; see Eq. (1) and Figure 3, top left. The proven equivariance properties of the proposed E(2)-equivariant steerable CNNs guarantee the specified transformation behavior of the feature spaces under such active transformations. However, our derivations so far don't prove any equivariance guarantees for *local*, independent rotations or reflections of small patches in an image.

The appropriate framework for analyzing local transformations is given by Gauge Equivariant Steerable CNNs [5]. In contrast to active transformations, Gauge Equivariant CNNs consider *passive gauge transformations*; see Figure 3, right. Adapted to our specific setting, each feature vector $f(x)$ is being expressed relative to a *local reference frame* (or gauge) $\big(e_1(x),\, e_2(x)\big)$ at $x \in \mathbb{R}^2$. A *gauge transformation* formalizes a change of local reference frames by the action of position dependent elements $g(x)$ of the *gauge group* (or structure group), in our case rotations and reflections in $G \leq \mathrm{O}(2)$. Since gauge transformations act independently on each position, they model independent transformation of local patches in an image. As derived in [5], the demand for *local gauge equivariance* results in the same kernel constraint as in Eq. (2). This implies that our models are automatically locally gauge equivariant[4].

More generally, the kernel constraint (2) applies to arbitrary 2-dimensional Riemannian manifolds $M$ with structure groups $G \leq \mathrm{O}(2)$. The presented solutions of the kernel space constraint therefore describe spherical CNNs [14, 5, 15, 16, 17, 18] or convolutional networks on triangulated meshes [19, 20, 21, 22] for different choices of structure groups and group representations.

# B  Overview over subgroups of E(2) and O(2)

The subgroups of $E(2) = (\mathbb{R}^2, +) \rtimes O(2)$ considered in this work are of the form $(\mathbb{R}^2, +) \rtimes G$ with $G \leq O(2)$. An overview over all possible choices is given in the following table.

|  | order $|G|$ | $G \leq O(2)$ | $(\mathbb{R}^2, +) \rtimes G$ |
|---|---|---|---|
| orthogonal | - | $O(2)$ | $E(2) \cong (\mathbb{R}^2, +) \rtimes O(2)$ |
| special orthogonal | - | $SO(2)$ | $SE(2) \cong (\mathbb{R}^2, +) \rtimes SO(2)$ |
| cyclic | N | $C_N$ | $(\mathbb{R}^2, +) \rtimes C_N$ |
| reflection | 2 | $(\{\pm 1\}, *) \cong D_1$ | $(\mathbb{R}^2, +) \rtimes (\{\pm 1\}, *)$ |
| dihedral | 2N | $D_N \cong C_N \rtimes (\{\pm 1\}, *)$ | $(\mathbb{R}^2, +) \rtimes D_N$ |

Table 6: Overview over the different groups covered in our framework.

# C  Implementation details

E(2)-steerable CNNs rely on convolutions with O(2)-steerable kernels. Our implementation therefore involves 1) computing a basis of steerable kernels, 2) the expansion of a steerable kernel in terms of this basis with learned expansion coefficients and 3) running the actual convolution routine. Since the kernel basis depends only on the chosen representations it is precomputed before training.

Given an input and output representation $\rho_{\text{in}}$ and $\rho_{\text{out}}$ of $G \leq O(2)$, we first precompute a basis $\{k_1, \ldots k_d\}$ of $G$-steerable kernels satisfying Eq. (2). In order to solve the kernel constraint we compute the types and multiplicities of irreps in the input and output representations using character theory [23]. The change of basis can be obtained by solving the linear system of equations $\rho(g) = Q^{-1}[\bigoplus_{i \in I} \psi_i(g)]Q \ \forall g \in G$. For each pair $\psi_i, \psi_j$ of irreps occurring in $\rho_{\text{out}}$ and $\rho_{\text{in}}$ we retrieve the analytical solutions $\{\kappa_1^{ij}, \ldots, \kappa_{d_{ij}}^{ij}\}$ listed in Appendix I.3. Together with the change of basis matrices $Q_{\text{in}}$ and $Q_{\text{out}}$, they fully determine the angular parts of the basis $\{k_1, \ldots, k_d\}$ of G-steerable kernels via Eq. (4). Since the kernel space constraint affects only the angular behavior of the kernels we are free to choose any radial profile. Following [7] and [2], we choose Gaussian radial profiles $\exp\left(\frac{1}{2\sigma^2}(r - R)^2\right)$ of width $\sigma$, centered at radii $R = 1, \ldots, \lfloor s/2 \rfloor$.

In practice, we consider digitized signals on a pixel grid[5] $\mathbb{Z}^2$. Correspondingly, we sample the analytically found kernel basis $\{k_1, \ldots, k_d\}$ on a square grid of size $s \times s$ to obtain their numerical representation of shape $(d, c_{\text{out}}, c_{\text{in}}, s, s)$. In this process it is important to prevent aliasing effects. Specifically, each basis kernel corresponds to one particular angular harmonic; see Table 1. When being sampled with a too low rate, a basis kernel can appear as a lower harmonic and might therefore introduce non-equivariant kernels to the sampled basis. For this reason, preventing aliasing is necessary to guarantee (approximate) equivariance. In order to ensure a faithful discretization, note that each Gaussian radial profile defines a ring whose circumference, and thus angular sampling rate, is proportional to its radius. It is therefore appropriate to bandlimit the kernel basis by a cutoff frequency which is chosen in proportion to the rings' radii. Since the basis kernels are harmonics of specific angular frequencies this is easily implemented by discarding high frequency solutions.

In typical applications the feature spaces are defined to be composed of multiple independent feature fields. Since the corresponding representations are block diagonal, this implies that the actual constraint (2) decomposes into multiple simpler constraints[6] which we leverage in our implementation to improve its computational efficiency. Assuming the output and input representations of a layer to be given by $\rho_{\text{out}} = \bigoplus_\gamma \rho_{\text{out}, \gamma}$ and $\rho_{\text{in}} = \bigoplus_\delta \rho_{\text{in}, \delta}$ respectively, the constraint on the full kernel space is equivalent to constraints on its blocks $k^{\gamma\delta}$ which map between the independent fields transforming

under $\rho_{\text{in},\delta}$ and $\rho_{\text{out},\gamma}$. Our implementation therefore computes a sampled basis $\{k_1^{\gamma\delta}, \ldots, k_{d^{\gamma\delta}}^{\gamma\delta}\}$ of $k^{\gamma\delta}$ for each pair $(\rho_{\text{in},\delta}, \rho_{\text{out},\gamma})$ of input and output representations individually.

At runtime, the convolution kernels are expanded by contracting the sampled kernel bases with learned weights. Specifically, each basis $\{k_1^{\gamma\delta}, \ldots, k_{d^{\gamma\delta}}^{\gamma\delta}\}$, realized by a tensor of shape $(d^{\gamma\delta}, c_{\text{out},\gamma}, c_{\text{in},\delta}, s, s)$, is expanded into the corresponding block $k^{\gamma\delta}$ of the kernel by contracting it with a tensor of learned parameters of shape $(d^{\gamma\delta})$. This process is sped up further by batching together multiple occurrences of the same pair of representations and thus block bases.

The resulting kernels are then used in a standard convolution routine. In practice we find that the time spent on the actual convolution of reasonably sized images outweighs the cost of the kernel expansion. In evaluation mode the parameters are not updated such that the kernel needs to be expanded only once and can then be reused. E(2)-steerable CNNs thus have no computational overhead over conventional CNNs at test time.

Our implementation is provided as a PyTorch extension which is available at https://github.com/QUVA-Lab/e2cnn. The library provides equivariant versions of many neural network operations, including G-steerable convolutions, nonlinearities, mappings to produce invariant features, spatial pooling, batch normalization and dropout. Feature fields are represented by *geometric tensors*, which are wrapping a `torch.Tensor` object and augment it, among other things, with their transformation law under the action of a symmetry group. This allows for a dynamic type-checking which prevents the user from applying operations to geometric tensors whose transformation law does not match the transformation law expected by the operation. The user interface hides most complications on group theory and solutions of the kernel space constraint and requires the user only to specify the transformation laws of feature spaces. For instance, a $C_8$-equivariant convolution operation, mapping a RGB image, identified as three scalar fields, to ten regular feature fields, would be instantiated by:

```
1  r2_act = Rot2dOnR2(N=8)
2  feat_type_in  = FieldType(r2_act,  3*[r2_act.trivial_repr])
3  feat_type_out = FieldType(r2_act, 10*[r2_act.regular_repr])
4  conv_op = R2Conv(feat_type_in, feat_type_out, kernel_size=5)
```

Everything the user has to do is to specify that the group $C_8$ acts on $\mathbb{R}^2$ by rotating it (line 1) and to define the types $\rho_{\text{in}} = \bigoplus_{i=1}^3 1$ and $\rho_{\text{out}} = \bigoplus_{i=1}^{10} \rho_{\text{reg}}^{C_8}$ of the input and output feature fields (lines 2 and 3), which are subsequently passed to the constructor of the steerable convolution (line 4).

# D   Further analysis of experimental results

## D.1   Model benchmarking on transformed MNIST datasets

In this section we analyze the benchmarking results of the 57 models in Table 7 in depth. All models in these experiments are derived from the base architecture described in Table 13 in Appendix K. The actual width of each model is adapted such that the number of parameters is approximately preserved. Note that this results in different numbers of channels, depending on the parameter efficiency of the corresponding models. All models apply some form of *invariant mapping* to scalar fields followed by spatial pooling after the last convolutional layer such that the predictions are guaranteed to be invariant under the equivariance group of the model. The number of invariant features passed to the fully connected classifier is approximately kept constant by adapting the width of the last convolutional layer to the invariant mapping used. The statistics of each experiment are averaged over (at least) 6 samples. In the remainder of this subsection we will guide through the results presented in Table 7. For more information on the training setup, see Appendix K.1.

**Regular steerable CNNs:**   Due to their popularity we first cover steerable CNNs whose features transform under *regular* representations of $C_N$ and $D_N$ for varying orders $N$. Note that these models correspond to group convolutional CNNs [6, 7]. For the dihedral models we choose a vertical reflection axis. We use ELUs [34] as pointwise nonlinearities and perform group pooling (see Section 2.6) as invariant map after the final convolution. Overall, regular steerable CNNs perform very well. The reason for this is that feature vectors, transforming under regular representations, can encode any function on the group.

Figure 2 summarizes the results for all regular steerable CNNs on all variants of MNIST (rows 2-10 and 19-27 in Table 7). For MNIST $O(2)$ and MNIST rot the prediction accuracies improve with $N$ but start to saturate at approximately 8 to 12 rotations. On MNIST $O(2)$ the $D_N$ models perform consistently better than the $C_N$ models of the same order $N$. This is the case since the dihedral models are guaranteed to generalize over reflections which are present in the dataset. All equivariant models outperform the non-equivariant CNN baseline.

On MNIST rot, the accuracy of the $C_N$-equivariant models improve significantly in comparison to their results on MNIST $O(2)$ since the intra-class variability is reduced. In contrast, the test errors of the $D_N$-equivariant models is the same on both datasets. The reason for this result is the reflection invariance of the $D_N$ models which implies that they can't distinguish between reflected digits. For $N = 1$ the dihedral model is purely reflection- but not rotation invariant and therefore performs even worse than the CNN baseline. This issue is resolved by restricting the dihedral models after the penultimate convolution to $C_N \leq D_N$, such that the group pooling after the final convolution results in only $C_N$-invariant features. This model, denoted in the figure by $D_N|_5 C_N$, achieves a slightly better accuracy than the pure $C_N$-equivariant model since it can leverage local reflectional symmetries. [7].

For MNIST 12k the non-restricted $D_N$ models perform again worse than the $C_N$ models since they are insensitive to the chirality of the digits. In order to explain the non-monotonic trend of the curves of the $C_N$ and $D_N$ models, notice that some of the digits are approximately related by symmetry transformations[8]. If these transformations happen to be part of the equivariance group w.r.t. which the model is invariant the predictions are more likely to be confused. This is mostly the case for $N$ being a multiple of 2 or 4 or for large orders $N$, which include almost all orientations. Once again, the restricted models, here $D_N|_5\{e\}$ and $C_N|_5\{e\}$, show the best results since they exploit local symmetries but preserve information on the global orientation. Since the restricted dihedral model generalizes over local reflections, its performance is consistently better than that of the restricted cyclic model.

**Quotient representations:** As an alternative to regular representations we experiment with some mixtures of quotient representations of $C_N$ (rows 11-15). These models differ from the regular models by enforcing more symmetries in the feature fields and thus kernels. The individual feature fields are lower dimensional; however, by fixing the number of parameters, the models use more different fields which in this specific case leads to approximately the same number of channels and therefore compute and memory requirements. We do not observe any significant difference in performance between regular and quotient representations. Appendix F gives more intuition on our specific choices of quotient representations and which symmetries they enforce. Note that the space of possible quotient representations and their multiplicities is very large and still needs to be investigated more thoroughly.

**Group pooling and vector field nonlinearities:** For $C_{16}$ we implement a group pooling network (row 16) and a vector field network (row 17). These models map regular feature fields, produced by each convolutional layer, to scalar fields and vector fields, respectively; see Section 2.6. These pooling operations compress the features in the regular fields, which can lead to lower memory and compute requirements. However, since we fix the number of parameters, the resulting models are ultimately much wider than the corresponding regular steerable CNNs. Since the pooling operations lead to a loss of information, both models perform worse than their purely regular counterpart on MNIST $O(2)$ and MNIST rot. Surprisingly, the group pooling network, whose features are orientation unaware, performs better than the vector field network. On MNIST 12k the group pooling network closes up with the regular steerable CNN while the vector field network achieves an even better result. We further experiment with a model which applies vector field nonlinearities to only half of the regular fields and preserves the other half (row 18). This model is on par with the regular model on both transformed MNIST versions but achieves the overall best result on MNIST 12k. Similar to the case of $C_{16}$, the group pooling network for $D_{16}$ (row 28) performs worse than the corresponding regular model, this time also on MNIST 12k.

| | group | representation | | nonlinearity | invariant map | citation | MNIST $O(2)$ | MNIST rot | MNIST 12k |
|---|---|---|---|---|---|---|---|---|---|
| 1 | $\{e\}$ | (conventional CNN) | | ELU | - | - | $5.53\pm0.20$ | $2.87\pm0.09$ | $0.91\pm0.06$ |
| 2 | $C_1$ | | | | | [7, 9] | $5.19\pm0.08$ | $2.48\pm0.13$ | $0.82\pm0.01$ |
| 3 | $C_2$ | | | | | [7, 9] | $3.29\pm0.07$ | $1.32\pm0.02$ | $0.87\pm0.04$ |
| 4 | $C_3$ | | | | | - | $2.87\pm0.04$ | $1.19\pm0.06$ | $0.80\pm0.03$ |
| 5 | $C_4$ | | | | | [6, 1, 7, 9, 10] | $2.40\pm0.05$ | $1.02\pm0.03$ | $0.99\pm0.03$ |
| 6 | $C_6$ | regular | $\rho_{\mathrm{reg}}$ | ELU | $G$-pooling | [8] | $2.08\pm0.03$ | $0.89\pm0.03$ | $0.84\pm0.02$ |
| 7 | $C_8$ | | | | | [7, 9] | $1.96\pm0.04$ | $0.84\pm0.02$ | $0.89\pm0.03$ |
| 8 | $C_{12}$ | | | | | [7] | $1.95\pm0.07$ | $0.80\pm0.03$ | $0.89\pm0.03$ |
| 9 | $C_{16}$ | | | | | [7, 9] | $1.93\pm0.04$ | $0.82\pm0.02$ | $0.95\pm0.04$ |
| 10 | $C_{20}$ | | | | | [7] | $1.95\pm0.05$ | $0.83\pm0.05$ | $0.94\pm0.06$ |
| 11 | $C_4$ | | $5\rho_{\mathrm{reg}}\oplus2\rho_{\mathrm{quot}}^{C_4/C_2}\oplus2\psi_0$ | | | [1] | $2.43\pm0.05$ | $1.03\pm0.05$ | $1.01\pm0.03$ |
| 12 | $C_8$ | | $5\rho_{\mathrm{reg}}\oplus2\rho_{\mathrm{quot}}^{C_8/C_2}\oplus2\rho_{\mathrm{quot}}^{C_8/C_4}\oplus2\psi_0$ | | | - | $2.03\pm0.05$ | $0.84\pm0.05$ | $0.91\pm0.02$ |
| 13 | $C_{12}$ | quotient | $5\rho_{\mathrm{reg}}\oplus2\rho_{\mathrm{quot}}^{C_{12}/C_2}\oplus2\rho_{\mathrm{quot}}^{C_{12}/C_4}\oplus3\psi_0$ | | | - | $2.04\pm0.04$ | $0.81\pm0.02$ | $0.95\pm0.02$ |
| 14 | $C_{16}$ | | $5\rho_{\mathrm{reg}}\oplus2\rho_{\mathrm{quot}}^{C_{16}/C_2}\oplus2\rho_{\mathrm{quot}}^{C_{16}/C_4}\oplus4\psi_0$ | | | - | $2.00\pm0.01$ | $0.86\pm0.04$ | $0.98\pm0.04$ |
| 15 | $C_{20}$ | | $5\rho_{\mathrm{reg}}\oplus2\rho_{\mathrm{quot}}^{C_{20}/C_2}\oplus2\rho_{\mathrm{quot}}^{C_{20}/C_4}\oplus5\psi_0$ | | | - | $2.01\pm0.05$ | $0.83\pm0.03$ | $0.96\pm0.04$ |
| 16 | | regular/scalar | $\psi_0 \xrightarrow{\mathrm{conv}} \rho_{\mathrm{reg}} \xrightarrow{G\text{-pool}} \psi_0$ | ELU, $G$-pooling | | [6, 38] | $2.02\pm0.02$ | $0.90\pm0.03$ | $0.93\pm0.04$ |
| 17 | $C_{16}$ | regular/vector | $\psi_1 \xrightarrow{\mathrm{conv}} \rho_{\mathrm{reg}} \xrightarrow{\text{vector pool}} \psi_1$ | vector field | | [13, 39] | $2.12\pm0.02$ | $1.07\pm0.03$ | $0.78\pm0.03$ |
| 18 | | mixed vector | $\rho_{\mathrm{reg}}\oplus\psi_1 \xrightarrow{\mathrm{conv}} 2\rho_{\mathrm{reg}} \xrightarrow[\text{pool}]{\text{vector}} \rho_{\mathrm{reg}}\oplus\psi_1$ | ELU, vector field | | - | $1.87\pm0.03$ | $0.83\pm0.02$ | $0.63\pm0.02$ |
| 19 | $D_1$ | | | | | - | $3.40\pm0.07$ | $3.44\pm0.10$ | $0.98\pm0.03$ |
| 20 | $D_2$ | | | | | - | $2.42\pm0.07$ | $2.39\pm0.04$ | $1.05\pm0.03$ |
| 21 | $D_3$ | | | | | - | $2.17\pm0.06$ | $2.15\pm0.05$ | $0.94\pm0.02$ |
| 22 | $D_4$ | | | | | [6, 1, 40] | $1.88\pm0.04$ | $1.87\pm0.04$ | $1.69\pm0.03$ |
| 23 | $D_6$ | regular | $\rho_{\mathrm{reg}}$ | ELU | $G$-pooling | [8] | $1.77\pm0.06$ | $1.77\pm0.04$ | $1.00\pm0.03$ |
| 24 | $D_8$ | | | | | - | $1.68\pm0.06$ | $1.73\pm0.03$ | $1.64\pm0.02$ |
| 25 | $D_{12}$ | | | | | - | $1.66\pm0.05$ | $1.65\pm0.05$ | $1.67\pm0.01$ |
| 26 | $D_{16}$ | | | | | - | $1.62\pm0.04$ | $1.65\pm0.02$ | $1.68\pm0.04$ |
| 27 | $D_{20}$ | | | | | - | $1.64\pm0.06$ | $1.62\pm0.05$ | $1.69\pm0.03$ |
| 28 | $D_{16}$ | regular/scalar | $\psi_{0,0} \xrightarrow{\mathrm{conv}} \rho_{\mathrm{reg}} \xrightarrow{G\text{-pool}} \psi_{0,0}$ | ELU, $G$-pooling | | - | $1.92\pm0.03$ | $1.88\pm0.07$ | $1.74\pm0.04$ |
| 29 | | irreps $\leq 1$ | $\bigoplus_{i=0}^1 \psi_i$ | | | - | $2.98\pm0.04$ | $1.38\pm0.09$ | $1.29\pm0.05$ |
| 30 | | irreps $\leq 3$ | $\bigoplus_{i=0}^3 \psi_i$ | | | - | $3.02\pm0.18$ | $1.38\pm0.09$ | $1.27\pm0.03$ |
| 31 | | irreps $\leq 5$ | $\bigoplus_{i=0}^5 \psi_i$ | | | - | $3.24\pm0.05$ | $1.44\pm0.10$ | $1.36\pm0.04$ |
| 32 | | irreps $\leq 7$ | $\bigoplus_{i=0}^7 \psi_i$ | ELU, norm-ReLU | conv2triv | - | $3.30\pm0.11$ | $1.51\pm0.10$ | $1.40\pm0.07$ |
| 33 | | $\mathbb{C}$-irreps $\leq 1$ | $\bigoplus_{i=0}^1 \psi_i^{\mathbb{C}}$ | | | [12] | $3.39\pm0.10$ | $1.47\pm0.06$ | $1.42\pm0.04$ |
| 34 | | $\mathbb{C}$-irreps $\leq 3$ | $\bigoplus_{i=0}^3 \psi_i^{\mathbb{C}}$ | | | [12] | $3.48\pm0.16$ | $1.51\pm0.05$ | $1.53\pm0.07$ |
| 35 | | $\mathbb{C}$-irreps $\leq 5$ | $\bigoplus_{i=0}^5 \psi_i^{\mathbb{C}}$ | | | - | $3.59\pm0.08$ | $1.59\pm0.05$ | $1.55\pm0.06$ |
| 36 | $SO(2)$ | $\mathbb{C}$-irreps $\leq 7$ | $\bigoplus_{i=0}^7 \psi_i^{\mathbb{C}}$ | | | - | $3.64\pm0.12$ | $1.61\pm0.06$ | $1.62\pm0.03$ |
| 37 | | | | ELU, squash | | - | $3.10\pm0.09$ | $1.41\pm0.04$ | $1.46\pm0.05$ |
| 38 | | | | ELU, norm-ReLU | | - | $3.23\pm0.08$ | $1.38\pm0.08$ | $1.33\pm0.03$ |
| 39 | | | | ELU, shared norm-ReLU | norm | - | $2.88\pm0.11$ | $1.15\pm0.06$ | $1.18\pm0.03$ |
| 40 | | irreps $\leq 3$ | $\bigoplus_{i=0}^3 \psi_i$ | shared norm-ReLU | | - | $3.61\pm0.09$ | $1.57\pm0.05$ | $1.88\pm0.05$ |
| 41 | | | | ELU, gate | conv2triv | - | $2.37\pm0.06$ | $1.09\pm0.03$ | $1.10\pm0.02$ |
| 42 | | | | ELU, shared gate | | - | $2.33\pm0.06$ | $1.11\pm0.03$ | $1.12\pm0.04$ |
| 43 | | | | ELU, gate | norm | - | $2.23\pm0.09$ | $1.04\pm0.04$ | $1.05\pm0.06$ |
| 44 | | | | ELU, shared gate | | - | $2.20\pm0.06$ | $1.01\pm0.03$ | $1.03\pm0.03$ |
| 45 | | irreps $= 0$ | $\psi_{0,0}$ | ELU | - | - | $5.46\pm0.46$ | $5.21\pm0.29$ | $3.98\pm0.04$ |
| 46 | | irreps $\leq 1$ | $\psi_{0,0}\oplus\psi_{1,0}\oplus2\psi_{1,1}$ | | | - | $3.31\pm0.17$ | $3.37\pm0.18$ | $3.05\pm0.09$ |
| 47 | | irreps $\leq 3$ | $\psi_{0,0}\oplus\psi_{1,0}\bigoplus_{i=1}^3 2\psi_{1,i}$ | ELU, norm-ReLU | $O(2)$-conv2triv | - | $3.42\pm0.03$ | $3.41\pm0.10$ | $3.86\pm0.09$ |
| 48 | | irreps $\leq 5$ | $\psi_{0,0}\oplus\psi_{1,0}\bigoplus_{i=1}^5 2\psi_{1,i}$ | | | - | $3.59\pm0.13$ | $3.78\pm0.31$ | $4.17\pm0.15$ |
| 49 | | irreps $\leq 7$ | $\psi_{0,0}\oplus\psi_{1,0}\bigoplus_{i=1}^7 2\psi_{1,i}$ | | | - | $3.84\pm0.25$ | $3.90\pm0.18$ | $4.57\pm0.27$ |
| 50 | | Ind-irreps $\leq 1$ | $\mathrm{Ind}\,\psi_0^{SO(2)} \oplus \mathrm{Ind}\,\psi_1^{SO(2)}$ | | | - | $2.72\pm0.05$ | $2.70\pm0.11$ | $2.39\pm0.07$ |
| 51 | $O(2)$ | Ind-irreps $\leq 3$ | $\mathrm{Ind}\,\psi_0^{SO(2)} \bigoplus_{i=1}^3 \mathrm{Ind}\,\psi_i^{SO(2)}$ | ELU, Ind norm-ReLU | Ind-conv2triv | - | $2.66\pm0.07$ | $2.65\pm0.12$ | $2.25\pm0.06$ |
| 52 | | Ind-irreps $\leq 5$ | $\mathrm{Ind}\,\psi_0^{SO(2)} \bigoplus_{i=1}^5 \mathrm{Ind}\,\psi_i^{SO(2)}$ | | | - | $2.71\pm0.11$ | $2.84\pm0.10$ | $2.39\pm0.09$ |
| 53 | | Ind-irreps $\leq 7$ | $\mathrm{Ind}\,\psi_0^{SO(2)} \bigoplus_{i=1}^7 \mathrm{Ind}\,\psi_i^{SO(2)}$ | | | - | $2.80\pm0.12$ | $2.85\pm0.06$ | $2.25\pm0.08$ |
| 54 | | irreps $\leq 3$ | $\psi_{0,0}\oplus\psi_{1,0}\bigoplus_{i=1}^3 2\psi_{1,i}$ | ELU, gate | $O(2)$-conv2triv | - | $2.39\pm0.05$ | $2.38\pm0.07$ | $2.28\pm0.07$ |
| 55 | | | | | norm | - | $2.21\pm0.09$ | $2.24\pm0.06$ | $2.15\pm0.03$ |
| 56 | | Ind-irreps $\leq 3$ | $\mathrm{Ind}\,\psi_0^{SO(2)} \bigoplus_{i=1}^3 \mathrm{Ind}\,\psi_i^{SO(2)}$ | ELU, Ind gate | Ind-conv2triv | - | $2.13\pm0.04$ | $2.09\pm0.05$ | $2.05\pm0.05$ |
| 57 | | | | | Ind-norm | - | $1.96\pm0.06$ | $1.95\pm0.05$ | $1.85\pm0.07$ |

Table 7: Extensive comparison of $G$-steerable CNNs for different choices of groups $G$, representations, nonlinearities and final $G$-invariant maps on three transformed MNIST datasets. Multiplicities of representations are reported in relative terms; the actual multiplicities are integer multiples with a depth dependent factor. All models apply a $G$-invariant map after the convolutions to guarantee an invariant prediction. Citations give credit to the works which proposed the corresponding model design. For reference see Sections 2.6, D.1, F and L.

**SO(2) irrep models:** The feature fields of all $\mathrm{SO}(2)$-equivariant models which we consider are defined to transform under irreducible representations; see Appendix E and I.2. Note that this covers scalar fields and vector fields which transform under $\psi_0^{\mathrm{SO}(2)}$ and $\psi_1^{\mathrm{SO}(2)}$, respectively. Overall these models are not competitive compared to the regular steerable CNNs. This result is particularly important for $\mathrm{SE}(3) \cong (\mathbb{R}^3, +) \rtimes \mathrm{SO}(3)$-equivariant CNNs whose feature fields are often transforming under the irreps of $\mathrm{SO}(3)$ [35, 2, 36, 15, 37].

The models in rows 29-32 are inspired by Harmonic Networks [12] and consist of irrep fields with the same multiplicity up to a certain threshold. All models apply ELUs on scalar fields and norm-ReLUs (see Section 2.6) on higher order fields. The projection to invariant features is done via a convolution to scalar features (conv2triv) in the last convolutional layer. We find that irrep fields up to order 1 and 3 perform equally well while higher thresholds yield worse results. The original implementation of Harmonic Networks considered complex irreps of $\mathrm{SO}(2)$ which results in a lower dimensional steerable kernel basis as discussed in Appendix I.5. We reimplemented these models and found that their reduced kernel space leads to consistently worse results (rows 33-36).

For the model containing irreps up to order 3 we implemented some alternative variants. For instance, the model in row 38 does not *convolve* to trivial features in the last layer but computes these by taking the norms of all non-scalar fields. This does not lead to significantly different results. Appendix L discusses all variations in detail.

By far the best results are achieved by the models in rows 41-44, which replace the norm-ReLUs with gated nonlinearities, see Section 2.6. This observation is in line with the results presented in [2], where gated nonlinearities were proposed.

**O(2) models:** As for $\mathrm{SO}(2)$, we are investigating $\mathrm{O}(2)$-equivariant models whose features transform under irreps up to a certain order and apply norm-ReLUs (rows 46-49). In this case we choose twice the multiplicity of 2-dimensional fields than scalar fields, which reflects the multiplicity of irreps contained in the regular representation of $\mathrm{O}(2)$. Invariant predictions are computed by convolving in equal proportion to fields which transform under trivial irreps $\psi_{0,0}^{\mathrm{O}(2)}$ and sign-flip irreps $\psi_{1,0}^{\mathrm{O}(2)}$ (see Appendix I.2), followed by taking the absolute value of the latter ($\mathrm{O}(2)$-conv2triv). We again find that higher irrep thresholds yield worse results, this time already starting from order 1. In particular, these models perform worse than their $\mathrm{SO}(2)$-equivariant counterparts even on MNIST $\mathrm{O}(2)$. This suggests that the kernel constraint for this particular choice of representations is too restrictive.

If only scalar fields, corresponding to the trivial irrep $\psi_{0,0}^{\mathrm{O}(2)}$, are chosen, the kernel constraint becomes $k(gx) = k(x) \ \forall g \in \mathrm{O}(2)$ and therefore allows for isotropic kernels only. This limits the expressivity of the model so severely that it performs even worse than a conventional CNN on MNIST rot and MNIST 12k while being on par for MNIST $\mathrm{O}(2)$, see row 45. Note that isotropic kernels correspond to vanilla graph convolutional networks (cf. the results and discussion in [5]).

In order to improve the performance of $\mathrm{O}(2)$-steerable CNNs, we propose to use representations $\mathrm{Ind}_{\mathrm{SO}(2)}^{\mathrm{O}(2)} \psi_k^{\mathrm{SO}(2)}$, which are induced from the irreps of $\mathrm{SO}(2)$ (see Appendix E for more details on induction). By the definition of induction, this leads to pairs of fields which transform according to $\psi_k^{\mathrm{SO}(2)}$ under rotations but permute under reflections. The multiplicity of the irreps of $\mathrm{O}(2)$ contained in this induced representation coincides with the multiplicities chosen in the pure $\mathrm{O}(2)$ irrep models. However, the change of basis, relating both representations, does not commute with the nonlinearities, such that the networks behave differently. We apply Ind norm-ReLU nonlinearities to the induced $\mathrm{O}(2)$ models which compute the norm of each of the permuting subfields individually but share the norm-ReLU parameters (the bias) to guarantee equivariance. In order to project to final, invariant features, we first apply a convolution producing $\mathrm{Ind}_{\mathrm{SO}(2)}^{\mathrm{O}(2)} \psi_0^{\mathrm{SO}(2)}$ fields (Ind-conv2triv). Since these transform like the regular representation of $(\{\pm 1\}, *) \cong \mathrm{O}(2)/\mathrm{SO}(2)$, we can simply apply $G$-pooling over the two reflections. The results, given in rows 50-53, show that these models perform significantly better than the $\mathrm{O}(2)$ irreps models and outperform the $\mathrm{SO}(2)$ irrep models on MNIST $\mathrm{O}(2)$. More specific details on all induced $\mathrm{O}(2)$ model operations are given in Appendix L.

We again build models which apply gated nonlinearities. As for $\mathrm{SO}(2)$, this leads to a greatly improved performance of the pure irrep models, see rows 54-55. In addition we adapt the gated nonlinearity to the *induced* irrep models (rows 56-57). Here we apply an independent gate to each of the two permuting sub-fields (Ind gate). In order to be equivariant, the gates need to permute under

reflections as well, which is easily achieved by deriving them from $\mathrm{Ind}_{\mathrm{SO}(2)}^{\mathrm{O}(2)} \psi_0^{\mathrm{SO}(2)}$ fields instead of scalar fields. The gated induced irrep model achieves the best results among all $\mathrm{O}(2)$-steerable networks, however, it is still not competitive compared to the $\mathrm{D}_N$ models with large $N$.

## D.2 On the convergence of Steerable CNNs

Figure 4: Validation errors and losses during the training of a conventional CNN and $\mathrm{C}_N$-equivariant models on MNIST rot. Networks with higher levels of equivariance converge significantly faster.

As shown in Figure 4, steerable CNNs converge significantly faster than non-equivariant CNNs. This faster convergence rate is explained by the fact that equivariant models generalize over transformed samples by design. Mathematically, $G$-steerable CNNs classify *equivalence classes* of images defined by the equivalence relation $f \sim f' \Leftrightarrow \exists\, tg \in (\mathbb{R}^2, +) \rtimes G$ s.t. $f(x) = f\big(g^{-1}(x-t)\big)$. Instead, MLPs learn to classify each image individually and conventional CNNs classify equivalence classes defined by translations, i.e. above equivalence classes for $G = \{e\}$. For more details see Section 2 of [7].

## D.3 STL-10 data ablation study

Figure 5 reports the results of a data ablation study which investigates the performance of the $\mathrm{D}_8\, \mathrm{D}_4\, \mathrm{D}_1$ models for smaller training set sizes. We use the same models and training procedure as in the main experiment on the full STL-10 dataset. For every single run, we generate new datasets by mixing the original training, validation and test set and sample reduced datasets such that all classes are balanced. The results are averaged over 4 runs on each of the considered training set sizes of 250, 500, 1000, 2000 or 4000. The validation and test sets contain 1000 and 8000 images, which are resampled in each run as well. The results validate that the gains from incorporating equivariance are consistent over all training sets.

Figure 5: Data ablation study on STL-10.

# E A short primer on group representation theory

Linear group representations model abstract algebraic group elements via their action on some vector space, that is, by representing them as linear transformations (matrices) on that space. Representation theory forms the backbone of Steerable CNNs since it describes the transformation law of their feature spaces. It is furthermore widely used to describe fields and their transformation behavior in physics.

Formally, a linear representation $\rho$ of a group $G$ on a vector space (representation space) $\mathbb{R}^n$ is a group homomorphism from $G$ to the general linear group $\mathrm{GL}(\mathbb{R}^n)$ (the group of invertible $n \times n$ matrices), i.e. it is a map

$$\rho : G \to \mathrm{GL}(\mathbb{R}^n) \quad \text{such that} \quad \rho(g_1 g_2) = \rho(g_1)\rho(g_2) \quad \forall g_1, g_2 \in G\,.$$

The requirement to be a homomorphism, i.e. to satisfy $\rho(g_1 g_2) = \rho(g_1)\rho(g_2)$, ensures the compatibility of the matrix multiplication $\rho(g_1)\rho(g_2)$ with the group composition $g_1 g_2$ which is necessary for a well defined group action. Note that group representations do not need to model the group *faithfully* (which would be the case for an isomorphism instead of a homomorphism).

A simple example is the *trivial representation* $\rho : G \to \mathrm{GL}(\mathbb{R})$ which maps any group element to the identity, i.e. $\forall g \in G \quad \rho(g) = 1$. The 2-dimensional rotation matrices $\psi(\theta) = \begin{bmatrix} \cos(\theta) & -\sin(\theta) \\ \sin(\theta) & \cos(\theta) \end{bmatrix}$ are an example of a representation of $\mathrm{SO}(2)$ (whose elements are identified by a rotation angle $\theta$).

**Equivalent representations** Two representations $\rho$ and $\rho'$ on $\mathbb{R}^n$ are called *equivalent* iff they are related by a change of basis $Q \in \mathrm{GL}(\mathbb{R}^n)$, i.e. $\rho'(g) = Q\rho(g)Q^{-1}$ for each $g \in G$. Equivalent representations behave similarly since their composition is basis independent as seen by $\rho'(g_1)\rho'(g_2) = Q\rho(g_1)Q^{-1}Q\rho(g_2)Q^{-1} = Q\rho(g_1)\rho(g_2)Q^{-1}$.

**Direct sums** Two representations can be combined by taking their *direct sum*. Given representations $\rho_1 : G \to \mathrm{GL}(\mathbb{R}^n)$ and $\rho_2 : G \to \mathrm{GL}(\mathbb{R}^m)$, their direct sum $\rho_1 \oplus \rho_2 : G \to \mathrm{GL}(\mathbb{R}^{n+m})$ is defined as

$$(\rho_1 \oplus \rho_2)(g) = \begin{bmatrix} \rho_1(g) & 0 \\ 0 & \rho_2(g) \end{bmatrix},$$

i.e. as the direct sum of the corresponding matrices. Its action is therefore given by the independent actions of $\rho_1$ and $\rho_2$ on the orthogonal subspaces $\mathbb{R}^n$ and $\mathbb{R}^m$ in $\mathbb{R}^{n+m}$. The direct sum admits an obvious generalization to an arbitrary number of representations $\rho_i$:

$$\bigoplus_i \rho_i(g) = \rho_1(g) \oplus \rho_2(g) \oplus \dots$$

**Irreducible representations** The action of a representation might leave a subspace of the representation space invariant. If this is the case there exists a change of basis to an equivalent representation which is decomposed into the direct sum of two independent representations on the invariant subspace and its orthogonal complement. A representation is called *irreducible* if no non-trivial invariant subspace exists.

Any representation $\rho : G \to \mathbb{R}^n$ of a compact group $G$ can therefore be decomposed as

$$\rho(g) = Q \left[ \bigoplus_{i \in I} \psi_i(g) \right] Q^{-1}$$

where $I$ is an index set specifying the irreducible representations $\psi_i$ contained in $\rho$ and $Q$ is a change of basis. In proofs it is therefore often sufficient to consider irreducible representations which we use in Section 2.4 to solve the kernel constraint.

**Regular and quotient representations** A commonly used representation in equivariant deep learning is the *regular representation*. The regular representation of a finite group $G$ acts on a vector space $\mathbb{R}^{|G|}$ by permuting its axes. Specifically, associating each axis $e_g$ of $\mathbb{R}^{|G|}$ to an element $g \in G$, the representation of an element $\tilde{g} \in G$ is a permutation matrix which maps $e_g$ to $e_{\tilde{g}g}$. For instance, the regular representation of the group $\mathrm{C}_4$ with elements $\{p\frac{\pi}{2}|p = 0, \dots, 3\}$ is instantiated by:

| $\phi$ | $0$ | $\frac{\pi}{2}$ | $\pi$ | $\frac{3\pi}{2}$ |
|---|---|---|---|---|
| $\rho_{\mathrm{reg}}^{\mathrm{C}_4}(\phi)$ | $\begin{bmatrix} 1 & 0 & 0 & 0 \\ 0 & 1 & 0 & 0 \\ 0 & 0 & 1 & 0 \\ 0 & 0 & 0 & 1 \end{bmatrix}$ | $\begin{bmatrix} 0 & 0 & 0 & 1 \\ 1 & 0 & 0 & 0 \\ 0 & 1 & 0 & 0 \\ 0 & 0 & 1 & 0 \end{bmatrix}$ | $\begin{bmatrix} 0 & 0 & 1 & 0 \\ 0 & 0 & 0 & 1 \\ 1 & 0 & 0 & 0 \\ 0 & 1 & 0 & 0 \end{bmatrix}$ | $\begin{bmatrix} 0 & 1 & 0 & 0 \\ 0 & 0 & 1 & 0 \\ 0 & 0 & 0 & 1 \\ 1 & 0 & 0 & 0 \end{bmatrix}$ |

A vector $v = \sum_g v_g e_g$ in $\mathbb{R}^{|G|}$ can be interpreted as a scalar function $v : G \to \mathbb{R}, g \mapsto v_g$ on $G$. Since $\rho(h)v = \sum_g v_g e_{hg} = \sum_{\tilde{g}} v_{h^{-1}\tilde{g}} e_{\tilde{g}}$ the regular representation corresponds to a left translation $[\rho(h)v](g) = v_{h^{-1}g}$ of such functions.

Very similarly, the quotient representation $\rho_{\mathrm{quot}}^{G/H}$ of $G$ w.r.t. a subgroup $H$ acts on $\mathbb{R}^{|G|/|H|}$ by permuting its axes. Labeling the axes by the cosets $gH$ in the quotient space $G/H$, it can be defined via its action $\rho_{\mathrm{quot}}^{G/H}(\tilde{g})e_{gH} = e_{\tilde{g}gH}$. An intuitive explanation of quotient representations is given in Appendix F.

Regular and trivial representations are two specific cases of quotient representations obtained by choosing $H = \{e\}$ or $H = G$, respectively. Vectors in the representation space $\mathbb{R}^{|G|/|H|}$ can be viewed as scalar functions on the quotient space $G/H$. The action of the quotient representations on $v$ then corresponds to a left translation of these functions on $G/H$.

**Restricted representations**  Any representation $\rho : G \to \mathrm{GL}(\mathbb{R}^n)$ can be uniquely restricted to a representation of a subgroup $H$ of $G$ by restricting its domain of definition:

$$\mathrm{Res}_H^G(\rho) : H \to \mathrm{GL}(\mathbb{R}^n), \ h \mapsto \rho\big|_H(h)$$

**Induced Representations**  Instead of restricting a representation from a group $G$ to a subgroup $H \leq G$, it is also possible to *induce* a representation of $H$ to a representation of $G$. In order to keep the presentation accessible we will first only consider the case of finite groups $G$ and $H$.

Let $\rho : H \to \mathrm{GL}(\mathbb{R}^n)$ be any representation of a subgroup $H$ of $G$. The induced representation $\mathrm{Ind}_H^G(\rho)$ is then defined on the representation space $\mathbb{R}^{n|G|/|H|}$ which can be seen as one copy of $\mathbb{R}^n$ for each of the $|G|/|H|$ cosets $gH$ in the quotient set $G/H$. For the definition of the induced representation it is customary to view this space as the tensor product $\mathbb{R}^{|G|/|H|} \otimes \mathbb{R}^n$ and to write vectors in this space as[9]

$$w = \sum_{gH} e_{gH} \otimes w_{gH} \ \in \mathbb{R}^{n\frac{|G|}{|H|}} \ , \tag{6}$$

where $e_{gH}$ is a basis vector of $\mathbb{R}^{|G|/|H|}$, associated to the coset $gH$, and $w_{gH}$ is some vector in the representation space $\mathbb{R}^n$ of $\rho$. Intuitively, $\mathrm{Ind}_H^G(\rho)$ acts on $\mathbb{R}^{n|G|/|H|}$ by $i)$ permuting the $|G|/|H|$ subspaces associated to the cosets $gH$ and $ii)$ acting on each of these subspaces via $\rho$.

To formalize this intuition, note that any element $g \in G$ can be identified by the coset $gH$ to which it belongs and an element $\mathrm{h}(g) \in H$ which specifies its position within this coset. Hereby $\mathrm{h} : G \to H$ expresses $g$ relative to an arbitrary *representative*[10] $\mathcal{R}(gH) \in G$ of $gH$ and is defined as $\mathrm{h}(g) := \mathcal{R}(gH)^{-1}g$ from which it immediately follows that $g$ is decomposed relative to $\mathcal{R}$ as

$$g = \mathcal{R}(gH)\mathrm{h}(g) \ . \tag{7}$$

The action of an element $\tilde{g} \in G$ on a coset $gH \in G/H$ is naturally given by $\tilde{g}gH \in G/H$. This action defines the aforementioned permutation of the $n$-dimensional subspaces in $\mathbb{R}^{n|G|/|H|}$ by sending $e_{gH}$ in Eq. (6) to $e_{\tilde{g}gH}$. Each of the $n$-dimensional, translated subspaces $\tilde{g}gH$, is in addition transformed by the action of $\rho\big(\mathrm{h}(\tilde{g}\mathcal{R}(gH))\big)$. This $H$-component $\mathrm{h}(\tilde{g}\mathcal{R}(gH)) = \mathcal{R}(\tilde{g}gH)^{-1}\tilde{g}\mathcal{R}(gH)$ of the $\tilde{g}$ action within the cosets accounts for the relative choice of representatives $\mathcal{R}(\tilde{g}gH)$ and $\mathcal{R}(gH)$. Overall, the action of $\mathrm{Ind}_H^G(\rho(\tilde{g}))$ is given by

$$\Big[\mathrm{Ind}_H^G \rho\Big](\tilde{g}) \sum_{gH} e_{gH} \otimes w_{gH} \ := \ \sum_{gH} e_{\tilde{g}gH} \otimes \rho\big(\mathrm{h}(\tilde{g}\mathcal{R}(gH))\big) w_{gH} \ , \tag{8}$$

which can be visualized as:

$$\text{Ind}_H^G \rho(\tilde{g}) \cdot \begin{bmatrix} \vdots \\ \hline w_{gH} \\ \vdots \\ \hline \vdots \\ \vdots \end{bmatrix} = \begin{bmatrix} \vdots \\ \hline \vdots \\ \hline \vdots \\ \hline \rho(\text{h}(\tilde{g}\mathcal{R}(gH)))w_{gH} \\ \vdots \end{bmatrix} \begin{matrix} \} \, gH \\ \\ \} \, \tilde{g}gH = \tilde{g}\mathcal{R}(gH)H \end{matrix}$$

Both quotient representations and regular representations can be viewed as being induced from trivial representations of a subgroup. Specifically, let $\rho_{\text{triv}}^{\{e\}} : \{e\} \to \text{GL}(R) = \{(+1)\}$ be the trivial representation of the the trivial subgroup. Then $\text{Ind}_{\{e\}}^G \rho_{\text{triv}}^{\{e\}} : G \to \text{GL}(\mathbb{R}^{|G|})$ is the regular representation which permutes the cosets $g\{e\}$ of $G/\{e\} \cong G$ which are in one to one relation to the group elements themself. For $\rho_{\text{triv}}^H : H \to \text{GL}(\mathbb{R}) = \{(+1)\}$ being the trivial representation of an arbitrary subgroup $H$ of $G$, the induced representation $\text{Ind}_H^G \rho_{\text{triv}}^H : G \to \text{GL}(\mathbb{R}^{|G|/|H|})$ permutes the cosets $gH$ of $H$ and thus coincides with the quotient representation $\rho_{\text{quot}}^{G/H}$.

Note that a vector in $\mathbb{R}^{|G|/|H|} \otimes \mathbb{R}^n$ is in one-to-one correspondence to a function $f : G/H \to \mathbb{R}^n$. The induced representation can therefore equivalently be defined as acting on the space of such functions as[11]

$$[\text{Ind}_H^G \rho(\tilde{g}) \cdot f](gH) = \rho(\text{h}(\tilde{g}\mathcal{R}(\tilde{g}^{-1}gH)))f(\tilde{g}^{-1}gH) \,. \tag{9}$$

This definition generalizes to non-finite groups where the quotient space $G/H$ is not necessarily finite anymore.

For the special case of semidirect product groups $G = N \rtimes H$ it is possible to choose representatives of the cosets $gH$ such that the elements $\text{h}(\tilde{g}\mathcal{R}(g'H)) = \text{h}(\tilde{g})$ become independent of the cosets [3]. This simplifies the action of the induced representation to

$$[\text{Ind}_H^G \rho(\tilde{g}) \cdot f](gH) = \rho(\text{h}(\tilde{g})) f(\tilde{g}^{-1}gH) \tag{10}$$

which corresponds to Eq. (1) for the group $G = \text{E}(2) = (\mathbb{R}^2, +) \rtimes \text{O}(2)$, subgroup $H = \text{O}(2)$ and quotient space $G/H = \text{E}(2)/\text{O}(2) = \mathbb{R}^2$.

## F  An intuition for quotient representation fields

The quotient representations of $\text{C}_N$ in rows 11-15 of Table 7 and in Table 3 are all of the form $\rho_{\text{quot}}^{\text{C}_N/\text{C}_M}$ with $\text{C}_M \leq \text{C}_N$. By the definition of quotient representations, given in Section 2.6, this implies features which are *invariant* under the action of $\text{C}_M$. For instance, $\rho_{\text{quot}}^{\text{C}_N/\text{C}_2}$-fields encode features like lines, which are invariant under rotations by $\pi$. Similarly, $\rho_{\text{quot}}^{\text{C}_N/\text{C}_4}$ features are invariant under rotations by $\pi/2$, and therefore describe features like a cross. The $N/M$ channels of a $\rho_{\text{quot}}^{\text{C}_N/\text{C}_M}$-field respond to different orientations of these patterns, e.g. to $+$ and $\times$ for the two channels of $\rho_{\text{quot}}^{\text{C}_8/\text{C}_4}$. A few more examples are given by the $16/2 = 8$ channels of $\rho_{\text{quot}}^{\text{C}_{16}/\text{C}_2}$, which respond to the patterns

$$-, \diagup\!\!\!, \diagup, /, |, \backslash, \diagdown \text{ and } \diagdown\!\!\!,$$

respectively, or the $16/4 = 4$ channels of $\rho_{\text{quot}}^{\text{C}_{16}/\text{C}_4}$, which respond to

$$+, \times\!\!\!, \times \text{ and } \times\!\!\!.$$

In principle, each of these patterns[12] could be encoded by a *regular* feature field of $\text{C}_N$. A regular field of type $\rho_{\text{reg}}^{\text{C}_N}$ comprises $N$ instead of $N/M$ channels, which detect arbitrary patterns in $N$ orientations, for instance,

$$\rtimes, \curlyvee, \times, \curlyvee, \ltimes, \curlyvee, \times \text{ and } \curlyvee$$

for $N = 8$. In the case of $C_M$-*symmetric* patterns, e.g. crosses for $M = 4$, this becomes

$$\times, +, \times, +, \times, +, \times \text{ and } + \,.$$

As evident from this example, the repetition after $N/M$ orientations (here $8/4 = 2$), introduces a redundancy in the responses of the regular feature fields. A quotient representation $\rho_{\text{quot}}^{C_N/C_M}$ addresses this redundancy by a-priori assuming the $C_M$ symmetry to be present and storing only the $N/M$ non-redundant responses. If symmetric patterns are important for the learning task, a quotient representation can therefore save computational, memory and parameter cost.

In our experiments we mostly used quotients by $C_2$ and $C_4$ since we assumed the corresponding symmetric patterns ($|$ and $+$) to be most frequent in MNIST. As hypothesized, our model, which uses the representations $5\rho_{\text{reg}} \oplus 2\rho_{\text{quot}}^{C_{16}/C_2} \oplus 2\rho_{\text{quot}}^{C_{16}/C_4} \oplus 4\psi_0$ of $C_{16}$, improves slightly upon a purely regular model with the same number of parameters, see Table 3. By mixing regular, quotient and trivial[13] representations, our model keeps a certain level of expressiveness in its feature fields but incorporates a-priori known symmetries and compresses the model.

We want to emphasize that quotient representations are expected to severely harm the model performance if the assumed symmetry does not actually exist in the data or is unimportant for the inference. Since the space of possible quotient representations and their multiplicities is very large, it might be necessary to apply some form of *neural architecture search* to find beneficial combinations. As a default choice we recommend the user to work with regular representations.

Further, note that the intuition given above is specific for the case of quotient representations $\rho_{\text{quot}}^{G/N}$ where $N \trianglelefteq G$ is a *normal* subgroup (which is always the case for $C_N$). Since normal subgroups imply $gN = Ng \;\; \forall g \in G$ by definition, the action of the quotient representation by any element $n \in N$ is given by $\rho_{\text{quot}}^{G/N}(n)e_{gN} = e_{ngN} = e_{nNg} = e_{Ng} = e_{gN}$, that it, it describes $N$-*invariant* feature fields. The quotient representations $\rho_{\text{quot}}^{G/H}$ for general, potentially non-normal subgroups $H \leq G$ also imply certain symmetries in the feature fields but are not necessarily $H$-invariant. For instance, the quotient representation $\rho_{\text{quot}}^{D_N/C_N}$ is invariant under rotations since $C_N$ is a normal subgroup of $D_N \cong C_N \rtimes (\{\pm 1\}, *)$, while the quotient representation $\rho_{\text{quot}}^{D_N/(\{\pm 1\}, *)}$ is not invariant since $(\{\pm 1\}, *)$ is not a normal subgroup of $D_N$. In the latter case one has instead

$$\rho_{\text{quot}}^{D_N/(\{\pm 1\}, *)}(s)e_{r(\{\pm 1\}, *)} = e_{sr(\{\pm 1\}, *)} = \begin{cases} e_{r(\{\pm 1\}, *)} & \text{for } s = +1 \\ e_{r^{-1}s(\{\pm 1\}, *)} = e_{r^{-1}(\{\pm 1\}, *)} & \text{for } s = -1 \end{cases}$$

for all $s \in (\{\pm 1\}, *)$ and representatives $r \in C_N$. The feature fields are therefore not invariant under the action of $(\{\pm 1\}, *)$ but become reversed.

## G   Equivariance of E(2) - steerable CNNs

### G.1   Equivariance of E(2) - steerable convolutions

Assume two feature fields $f_{\text{in}} : \mathbb{R}^2 \to \mathbb{R}^{c_{\text{in}}}$ of type $\rho_{\text{in}}$ and $f_{\text{out}} : \mathbb{R}^2 \to \mathbb{R}^{c_{\text{out}}}$ of type $\rho_{\text{out}}$ to be given. Under actions of the Euclidean group these fields transform as

$$f_{\text{in}}(x) \mapsto \left( \left[ \text{Ind}_G^{(\mathbb{R}^2, +) \rtimes G} \rho_{\text{in}} \right] (gt) f_{\text{in}} \right) (x) := \rho_{\text{in}}(g) f_{\text{in}} \left( g^{-1}(x - t) \right)$$

$$f_{\text{out}}(x) \mapsto \left( \left[ \text{Ind}_G^{(\mathbb{R}^2, +) \rtimes G} \rho_{\text{out}} \right] (gt) f_{\text{out}} \right) (x) := \rho_{\text{out}}(g) f_{\text{out}} \left( g^{-1}(x - t) \right) \,.$$

Here we show that the $G$-steerability (2) of convolution kernels is *sufficient* to guarantee the equivariance of the mapping. We therefore define the convolution (or correlation) operation of a feature field with a $G$-steerable kernel $k : \mathbb{R}^2 \to \mathbb{R}^{c_{\text{out}} \times c_{\text{in}}}$ as usual by

$$f_{\text{out}}(x) := (k * f_{\text{in}})(x) = \int_{\mathbb{R}^2} k(y) f_{\text{in}}(x + y) \, \mathrm{d}y \,.$$

The convolution with a transformed input field then gives

$$
\int_{\mathbb{R}^2} \mathrm{d}y \, k(y) \left( \left[ \mathrm{Ind}_G^{(\mathbb{R}^2,+) \rtimes G} \, \rho_{\mathrm{in}} \right] (gt) f_{\mathrm{in}} \right) (x+y)
$$

$$
= \int_{\mathbb{R}^2} \mathrm{d}y \, k(y) \rho_{\mathrm{in}}(g) f_{\mathrm{in}} \left( g^{-1}(x+y-t) \right)
$$

$$
= \int_{\mathbb{R}^2} \mathrm{d}y \, \rho_{\mathrm{out}}(g) k(g^{-1}y) \rho_{\mathrm{in}}(g)^{-1} \, \rho_{\mathrm{in}}(g) f_{\mathrm{in}} \left( g^{-1}(x+y-t) \right)
$$

$$
= \rho_{\mathrm{out}}(g) \int_{\mathbb{R}^2} \mathrm{d}\tilde{y} \, k(\tilde{y}) f_{\mathrm{in}} \left( g^{-1}(x-t) + \tilde{y} \right)
$$

$$
= \rho_{\mathrm{out}}(g) f_{\mathrm{out}} \left( g^{-1}(x-t) \right)
$$

$$
= \left( \left[ \mathrm{Ind}_G^{(\mathbb{R}^2,+) \rtimes G} \, \rho_{\mathrm{out}} \right] (gt) f_{\mathrm{out}} \right) (x) \,,
$$

i.e. it satisfies the desired equivariance condition

$$
k * \left( \left[ \mathrm{Ind}_G^{(\mathbb{R}^2,+) \rtimes G} \, \rho_{\mathrm{in}} \right] (gt) f_{\mathrm{in}} \right) \;=\; \left[ \mathrm{Ind}_G^{(\mathbb{R}^2,+) \rtimes G} \, \rho_{\mathrm{out}} \right] (gt) \left( k * f_{\mathrm{in}} \right) .
$$

We used the kernel steerability (2) in the second step to identify $k(x)$ with $\rho_{\mathrm{out}}(g) k(g^{-1}x) \rho_{\mathrm{in}}(g^{-1})$. In the third step we substituted $\tilde{y} = g^{-1}y$ which does not affect the integral measure since $\left| \det \left( \frac{\partial y}{\partial \tilde{y}} \right) \right| = |\det(g)| = 1$ for an orthogonal transformation $g \in G$.

A proof showing the $G$-steerability of the kernel to not only be sufficient but necessary is given in [2].

## G.2 Equivariance of spatially localized nonlinearities

We consider nonlinearities of the form

$$
\sigma : \mathbb{R}^{c_{\mathrm{in}}} \to \mathbb{R}^{c_{\mathrm{out}}}, \; f(x) \mapsto \sigma\big(f(x)\big) \,,
$$

which act spatially localized on feature vectors $f(x) \in \mathbb{R}^{c_{\mathrm{in}}}$. These localized nonlinearities are used to define nonlinearities $\bar{\sigma}$ acting on entire feature fields $f : \mathbb{R}^2 \to \mathbb{R}^{c_{\mathrm{in}}}$ by mapping each feature vector individually, that is,

$$
\bar{\sigma} : \; f \mapsto \bar{\sigma}(f) \quad \text{such that} \quad \bar{\sigma}(f)(x) := \sigma\big(f(x)\big) \,.
$$

In order for $\bar{\sigma}$ to be equivariant under the action of induced representations it is sufficient to require

$$
\sigma \circ \rho_{\mathrm{in}}(g) \;=\; \rho_{\mathrm{out}}(g) \circ \sigma \qquad \forall g \in G
$$

since then

$$
\bar{\sigma} \left( \left[ \mathrm{Ind}_G^{(\mathbb{R}^2,+) \rtimes G} \, \rho_{\mathrm{in}} \right] (gt) f \right) (x) \; = \sigma \left( \rho_{\mathrm{in}}(g) f(g^{-1}(x-t)) \right)
$$

$$
= \rho_{\mathrm{out}}(g) \sigma \left( f(g^{-1}(x-t)) \right)
$$

$$
= \rho_{\mathrm{out}}(g) \bar{\sigma}(f)(g^{-1}(x-t))
$$

$$
= \left[ \mathrm{Ind}_G^{(\mathbb{R}^2,+) \rtimes G} \, \rho_{\mathrm{out}} \right] \bar{\sigma}(f)(x) \,.
$$

### G.2.1 Equivariance of individual subspace nonlinearities w.r.t. direct sum representations

The feature spaces of steerable CNNs comprise multiple feature fields $f_i : \mathbb{R}^2 \to \mathbb{R}^{c_{\mathrm{in},i}}$ which are concatenated into one big feature field $f : \mathbb{R}^2 \to \mathbb{R}^{\sum_i c_{\mathrm{in},i}}$ defined by $f := \bigoplus_i f_i$. By definition $f(x)$ transforms under $\rho_{\mathrm{in}} = \bigoplus_i \rho_{\mathrm{in},i}$ if each $f_i(x)$ transforms under $\rho_{\mathrm{in},i}$. If $\sigma_i : \mathbb{R}^{c_{\mathrm{in},i}} \to \mathbb{R}^{c_{\mathrm{out},i}}$ is an equivariant nonlinearity satisfying $\sigma_i \circ \rho_{\mathrm{in},i}(g) = \rho_{\mathrm{out},i}(g) \circ \sigma_i$ for all $g \in G$, then

$$
\left( \bigoplus_i \sigma_i \right) \circ \left( \bigoplus_i \rho_{\mathrm{in},i}(g) \right) = \left( \bigoplus_i \rho_{\mathrm{out},i}(g) \right) \circ \left( \bigoplus_i \sigma_i \right) \quad \forall g \in G,
$$

i.e. the concatenation of feature fields respects the equivariance of the individual nonlinearities. Here we defined $\bigoplus_i \sigma_i : \mathbb{R}^{\sum_i c_{\text{in},i}} \to \mathbb{R}^{\sum_i c_{\text{out},i}}$ as acting individually on the corresponding $f_i(x)$ in $f(x)$, that is, $\left(\bigoplus_i \sigma_i\right)\left(\bigoplus_i f_i(x)\right) = \bigoplus_i \left(\sigma_i \circ \text{proj}_i\right)(f(x))$.

To proof this statement consider without loss of generality the case of two feature fields $f_1$ and $f_2$ with corresponding representations $\rho_1$ and $\rho_2$ and equivariant nonlinearities $\sigma_1$ and $\sigma_2$. Then it follows for all $g \in G$ that

$$
\begin{aligned}
&\left(\sigma_1 \oplus \sigma_2\right)\left((\rho_{1,\text{in}} \oplus \rho_{2,\text{in}})(g)\right)\left(f_1 \oplus f_2\right) \\
&= \sigma_1\left(\rho_1(g)\,f_1\right) \oplus \sigma_2\left(\rho_2(g)\,f_2\right) \\
&= \rho_1(g)\,\sigma_1(f_1) \,\oplus\, \rho_2(g)\,\sigma_2(f_2) \\
&= \left((\rho_{1,\text{out}} \oplus \rho_{2,\text{out}})(g)\right)\left(\sigma_1 \oplus \sigma_2\right)\left(f_1 \oplus f_2\right).
\end{aligned}
$$

The general case follows by induction.

### G.2.2 Equivariance of norm nonlinearities w.r.t. unitary representations

We define unitary representations to preserve the norm of feature vectors, i.e. $\left|\rho_{\text{iso}}(g)f(x)\right| = \left|f(x)\right| \,\forall g \in G$. Norm nonlinearities are functions of the type $\sigma_{\text{norm}}\left(f(x)\right) := \eta\left(|f(x)|\right)\frac{f(x)}{|f(x)|}$, where $\eta : \mathbb{R}_{\geq 0} \to \mathbb{R}$ is some nonlinearity acting on norm of a feature vector. Since norm nonlinearities preserve the orientation of feature vectors they are equivariant under the action of unitary representations:

$$
\begin{aligned}
\sigma_{\text{norm}}\left(\rho_{\text{iso}}(g)f(x)\right) &= \eta\left(\left|\rho_{\text{iso}}(g)f(x)\right|\right)\frac{\rho_{\text{iso}}(g)f(x)}{\left|\rho_{\text{iso}}(g)f(x)\right|} \\
&= \eta\left(|f(x)|\right)\frac{\rho_{\text{iso}}(g)f(x)}{|f(x)|} \\
&= \rho_{\text{iso}}(g)\sigma_{\text{norm}}\left(f(x)\right).
\end{aligned}
$$

### G.2.3 Equivariance of pointwise nonlinearities w.r.t. regular and quotient representations

Quotient representations act on feature vectors by permuting their entries according to the group composition as defined by $\rho_{\text{quot}}^{G/H}(\tilde{g})e_{gH} := e_{\tilde{g}gH}$. The permutation of vector entries commutes with pointwise nonlinearities $\sigma : \mathbb{R} \to \mathbb{R}$ which are being applied to each entry of a feature vector individually:

$$
\begin{aligned}
\sigma_{\text{pt}}\left(\rho_{\text{quot}}^{G/H}(\tilde{g})\,f(x)\right) &= \sigma_{\text{pt}}\left(\rho_{\text{quot}}^{G/H}(\tilde{g})\sum_{gH \in G/H} f_{gH}(x)\,e_{gH}\right) \\
&= \sigma_{\text{pt}}\left(\sum_{gH \in G/H} f_{gH}(x)\,e_{\tilde{g}gH}\right) \\
&= \sum_{gH \in G/H} \sigma_{\text{pt}}\left(f_{gH}(x)\right)e_{\tilde{g}gH} \\
&= \rho_{\text{quot}}^{G/H}(\tilde{g})\sum_{gH \in G/H} \sigma_{\text{pt}}\left(f_{gH}(x)\right)e_{gH} \\
&= \rho_{\text{quot}}^{G/H}(\tilde{g})\sigma_{\text{pt}}\left(f(x)\right).
\end{aligned}
$$

Any pointwise nonlinearity is therefore equivariant under the action of quotient representations. The same holds true for regular representations $\rho_{\text{reg}}^{G} = \rho_{\text{quot}}^{G/\{e\}}$ which are a special case of quotient representations for the choice $H = \{e\}$

### G.2.4 Equivariance of vector field nonlinearities w.r.t. regular and standard representations

Vector field nonlinearities map an $N$-dimensional feature field which is transforming under the regular representation $\rho_{\text{reg}}^{C_N}$ of $C_N$ to a vector field. As the elements of $C_N$ correspond to rotations by angles

$\theta_p \in \left\{ p \frac{2\pi}{N} \right\}_{p=0}^{N-1}$ we can write the action of the cyclic group in this specific case as $\rho_{\mathrm{reg}}^{\mathrm{C}_N}(\tilde{\theta}) e_\theta := e_{\tilde{\theta}+\theta}$ and the feature vector as $f(x) = \sum_{\theta \in \mathrm{C}_N} f_\theta(x) e_\theta$. In this convention vector field nonlinearities are defined as

$$\sigma_{\mathrm{vec}}(f(x)) := \max(f(x)) \begin{pmatrix} \cos(\arg\max f(x)) \\ \sin(\arg\max f(x)) \end{pmatrix}.$$

The maximum operation $\max : \mathbb{R}^N \to \mathbb{R}$ thereby returns the maximal field value which is invariant under transformations of the regular input field. Observe that

$$\begin{aligned} \arg\max\left( \rho_{\mathrm{reg}}^{\mathrm{C}_N}(\tilde{\theta}) f(x) \right) &= \arg\max\left( \rho_{\mathrm{reg}}^{\mathrm{C}_N}(\tilde{\theta}) \sum_{\theta \in \mathrm{C}_N} f_\theta(x) e_\theta \right) \\ &= \arg\max\left( \sum_{\theta \in \mathrm{C}_N} f_\theta(x) e_{\tilde{\theta}+\theta} \right) \\ &= \tilde{\theta} + \arg\max(f(x)) \end{aligned}$$

such that

$$\begin{aligned} \sigma_{\mathrm{vec}}\left( \rho_{\mathrm{reg}}^{\mathrm{C}_N}(\tilde{\theta}) f(x) \right) &= \max(f(x)) \begin{pmatrix} \cos(\tilde{\theta} + \arg\max f(x)) \\ \sin(\tilde{\theta} + \arg\max f(x)) \end{pmatrix} \\ &= \max(f(x)) \begin{pmatrix} \cos(\tilde{\theta}) & -\sin(\tilde{\theta}) \\ \sin(\tilde{\theta}) & \cos(\tilde{\theta}) \end{pmatrix} \begin{pmatrix} \cos(\arg\max f(x)) \\ \sin(\arg\max f(x)) \end{pmatrix} \end{aligned}$$

transforms under the standard representation $\rho(\tilde{\theta}) = \begin{pmatrix} \cos(\tilde{\theta}) & -\sin(\tilde{\theta}) \\ \sin(\tilde{\theta}) & \cos(\tilde{\theta}) \end{pmatrix}$ of $\mathrm{C}_N$. This proofs that the resulting feature field indeed transforms like a vector field.

The original paper [13] used a different convention $\arg\max : \mathbb{R}^N \to \{0, \dots, N-1\}$, returning the integer index of the maximal vector entry. This leads to a corresponding rotation angle $\theta = \frac{2\pi}{N} \arg\max(f(x)) \in \mathrm{C}_N$ in terms of which the vector field nonlinearity reads $\sigma_{\mathrm{vec}}(f(x)) = \max(f(x)) \begin{pmatrix} \cos(\theta) \\ \sin(\theta) \end{pmatrix}$

### G.2.5 Equivariance of nonlinearities w.r.t. induced representations

Consider a group $H < G$ with two representations $\rho_{\mathrm{in}} : H \to \mathrm{GL}(\mathbb{R}^{c_{\mathrm{in}}})$ and $\rho_{\mathrm{out}} : H \to \mathrm{GL}(\mathbb{R}^{c_{\mathrm{out}}})$. Suppose we are given an equivariant non-linearity $\sigma : \mathbb{R}^{c_{\mathrm{in}}} \to \mathbb{R}^{c_{\mathrm{out}}}$ with respect to the actions of $\rho_{\mathrm{in}}$ and $\rho_{\mathrm{out}}$, that is, $\rho_{\mathrm{out}}(h) \circ \sigma = \sigma \circ \rho_{\mathrm{in}}(h) \quad \forall h \in H$. Then an *induced* non-linearity $\tilde{\sigma}$, equivariant w.r.t. the induced representations $\mathrm{Ind}_H^G \rho_{\mathrm{in}}$ and $\mathrm{Ind}_H^G \rho_{\mathrm{out}}$ of $G$, can be defined as applying $\sigma$ independently to each of the $|G : H|$ different $c_{\mathrm{in}}$-dimensional subspaces of the representation space which are being permuted by the action of $\mathrm{Ind}_H^G \rho_{\mathrm{in}}$, see Appendix E. The permutation of the subspaces commutes with the individual action of the nonlinearity $\tilde{\sigma}$ on the subspaces, while the non-linearity $\sigma$ itself commutes with the transformation within the subspaces through $\rho$ by assumption.

Expressing the feature vector as $f(x) = \sum_{gH} e_{gH} \otimes f_{gH}(x)$ this is seen by:

$$\tilde{\sigma}\left(\mathrm{Ind}_H^G \rho_{\mathrm{in}}(\tilde{g})\, f(x)\right) = \tilde{\sigma}\left(\mathrm{Ind}_H^G \rho_{\mathrm{in}}(\tilde{g}) \sum_{gH \in G/H} e_{gH} \otimes f_{gH}(x)\right)$$

$$= \tilde{\sigma}\left(\sum_{gH \in G/H} \mathrm{Ind}_H^G \rho_{\mathrm{in}}(\tilde{g})\, (e_{gH} \otimes f_{gH}(x))\right)$$

$$= \tilde{\sigma}\left(\sum_{gH \in G/H} e_{\tilde{g}gH} \otimes \rho_{\mathrm{in}}(\mathrm{h}(\tilde{g}r(gH)))f_{gH}(x)\right)$$

$$= \sum_{gH \in G/H} e_{\tilde{g}gH} \otimes \sigma\big(\rho_{\mathrm{in}}(\mathrm{h}(\tilde{g}r(gH)))f_{gH}(x)\big)$$

$$= \sum_{gH \in G/H} e_{\tilde{g}gH} \otimes \rho_{\mathrm{out}}(\mathrm{h}(\tilde{g}r(gH)))\sigma\big(f_{gH}(x)\big)$$

$$= \mathrm{Ind}_H^G \rho_{\mathrm{out}}(\tilde{g}) \sum_{gH \in G/H} e_{gH} \otimes \sigma\left(f_{gH}(x)\right)$$

$$= \mathrm{Ind}_H^G \rho_{\mathrm{out}}(\tilde{g})\, \tilde{\sigma}\big(f(x)\big)$$

## H   Visualizations of the irrep kernel constraint

The irrep kernel constraint

$$\kappa(gx) = \left[\bigoplus_{i \in I_{\mathrm{out}}} \psi_i(g)\right] \kappa(x) \left[\bigoplus_{j \in I_{\mathrm{in}}} \psi_j^{-1}(g)\right] \qquad \forall g \in G,\ x \in \mathbb{R}^2$$

decomposes into independent constraints

$$\kappa^{ij}(gx) = \psi_i(g)\, \kappa^{ij}(x)\, \psi_j^{-1}(g) \qquad \forall g \in G,\ x \in \mathbb{R}^2 \quad \text{where } i \in I_{\mathrm{out}},\ j \in I_{\mathrm{in}},$$

on invariant subspaces corresponding to blocks $\kappa^{ij}$ of $\kappa$. This is the case since the direct sums of irreps on the right hand side are block diagonal:

$$\underbrace{\begin{pmatrix} \kappa^{i_1 j_1}(gx) & \kappa^{i_1 j_2}(gx) & \cdots \\ \hline \kappa^{i_2 j_1}(gx) & \kappa^{i_2 j_2}(gx) & \cdots \\ \hline \vdots & \vdots & \ddots \end{pmatrix}}_{\kappa(gx)} = \underbrace{\begin{pmatrix} \psi_{i_1}(g) & & \\ \hline & \psi_{i_2}(g) & \\ \hline & & \ddots \end{pmatrix}}_{\bigoplus_{i \in I_{\mathrm{out}}} \psi_i(g)} \cdot \underbrace{\begin{pmatrix} \kappa^{i_1 j_1}(x) & \kappa^{i_1 j_2}(x) & \cdots \\ \hline \kappa^{i_2 j_1}(x) & \kappa^{i_2 j_2}(x) & \cdots \\ \hline \vdots & \vdots & \ddots \end{pmatrix}}_{\kappa(x)} \cdot \underbrace{\begin{pmatrix} \psi_{j_1}^{-1}(g) & & \\ \hline & \psi_{j_2}^{-1}(g) & \\ \hline & & \ddots \end{pmatrix}}_{\bigoplus_{j \in I_{\mathrm{in}}} \psi_j^{-1}(g)}$$

A basis $\left\{\kappa_1^{ij}, \cdots, \kappa_{d_{ij}}^{ij}\right\}$ for the space of $G$-steerable kernels satisfying the independent constraints (3) on $\kappa^{ij}$ contributes to a part of the full basis

$$\{k_1, \cdots, k_d\} := \bigcup_{i \in I_{\mathrm{out}}} \bigcup_{j \in I_{\mathrm{in}}} \left\{Q_{\mathrm{out}}^{-1} \overline{\kappa}_1^{ij} Q_{\mathrm{in}}, \cdots, Q_{\mathrm{out}}^{-1} \overline{\kappa}_{d_{ij}}^{ij} Q_{\mathrm{in}}\right\}. \qquad (11)$$

of $G$-steerable kernels satisfying the original constraint (2). Here we defined a zero-padded block

$$\overline{\kappa}^{ij} := \begin{pmatrix} 0 & 0 & 0 \\ \hline 0 & \kappa^{ij} & 0 \\ \hline 0 & 0 & 0 \end{pmatrix}.$$

## I   Solutions of the kernel constraints for irreducible representations

In this section we are deriving analytical solutions of the kernel constraints

$$\kappa^{ij}(gx) = \psi_i(g)\, \kappa^{ij}(x)\, \psi_j^{-1}(g) \qquad \forall g \in G,\ x \in \mathbb{R}^2 \qquad (12)$$

for irreducible representations $\psi_i$ of O(2) and its subgroups. The linearity of the constraint implies that the solution space of G-steerable kernels forms a linear subspace of the unrestricted kernel space $k \in L^2(\mathbb{R}^2)^{c_{\text{out}} \times c_{\text{in}}}$ of square integrable functions $k : \mathbb{R}^2 \to \mathbb{R}^{c_{\text{out}} \times c_{\text{in}}}$.

Section I.1 introduces the conventions, notation and basic properties used in the derivations. Since our numerical implementation is on the real field we are considering real-valued irreps. It is in general possible to derive all solutions considering complex valued irreps of $G \leq O(2)$. While this approach would simplify some steps it comes with an overhead of relating the final results back to the real field which leads to further complications, see Appendix I.5. An overview over the real-valued irreps of $G \leq O(2)$ and their properties is given in Section I.2.

We present the analytical solutions of the irrep kernel constraints for all possible pairs of irreps in Section I.3. Specifically, the solutions for SO(2) are given in Table 8 while the solutions for O(2), $(\{\pm 1\}, *)$, $C_N$ and $D_N$ are given in Table 9, Table 10, Table 11 and Table 12 respectively.

Our derivation of the irrep kernel bases is motivated by the observation that the irreps of O(2) and subgroups are harmonics, that is, they are associated to one particular angular frequency. This suggests that the kernel constraint (12) decouples into simpler constraints on individual Fourier modes. In the derivations, presented in Section I.4, we are therefore defining the kernels in polar coordinates $x = x(r, \phi)$ and expand them in terms of an orthogonal, angular, Fourier-like basis. A projection on this orthogonal basis then yields constraints on the expansion coefficients. Only specific coefficients are allowed to be non-zero; these coefficients parameterize the complete space of $G$-steerable kernels satisfying the irrep constraint (12). The completeness of the solution follows from the completeness of the orthogonal basis.

We start with deriving the bases for the simplest cases SO(2) and $(\{\pm 1\}, *)$ in sections I.4.1 and I.4.2. The $G$-steerable kernel basis for O(2) forms a subspace of the kernel basis for SO(2) such that it can be easily derived from this solution by adding the additional constraint coming from the reflectional symmetries in $(\{\pm 1\}, *) \cong O(2)/SO(2)$. This additional constraint is imposed in Section I.4.3. Since $C_N$ is a subgroup of discrete rotations in SO(2) their derivation is mostly similar. However, the discreteness of rotation angles leads to $N$ systems of linear congruences modulo $N$ in the final step. This system of equations is solved in Section I.4.4. Similar to how we derived the kernel basis for O(2) from SO(2), we derive the basis for $D_N$ from $C_N$ by adding reflectional constraints from $(\{\pm 1\}, *) \cong D_N / C_N$ in Section I.4.5.

## I.1 Conventions, notation and basic properties

Throughout this section we denote rotations in SO(2) and $C_N$ by $r_\theta$ with $\theta \in [0, 2\pi)$ and $\theta \in \left\{ p\frac{2\pi}{N} \right\}_{p=0}^{N-1}$ respectively. Since $O(2) \cong SO(2) \rtimes (\{\pm 1\}, *)$ can be seen as a semidirect product of rotations and reflections we decompose orthogonal group elements into a unique product $g = r_\theta s \in O(2)$ where $s \in (\{\pm 1\}, *)$ is a reflection and $r_\theta \in SO(2)$. Similarly, we write $g = r_\theta s \in D_N$ for the dihedral group $D_N \cong C_N \rtimes (\{\pm 1\}, *)$, in this case with $r_\theta \in C_N$.

The action of a rotation $r_\theta$ on $\mathbb{R}^2$ in polar coordinates $x(r, \phi)$ is given by $r_\theta.x(r, \phi) = x(r, r_\theta.\phi) = x(r, \phi + \theta)$. An element $g = r_\theta s$ of O(2) or $D_N$ acts on $\mathbb{R}^2$ as $g.x(r, \phi) = x(r, r_\theta s.\phi) = x(r, s\phi + \theta)$ where the symbol $s$ denotes both group elements in $(\{\pm 1\}, *)$ and numbers in $\{\pm 1\}$.

We denote a $2 \times 2$ orthonormal matrix with positive determinant, i.e. rotation matrix for an angle $\theta$, by:

$$\psi(\theta) = \begin{bmatrix} \cos(\theta) & -\sin(\theta) \\ \sin(\theta) & \cos(\theta) \end{bmatrix}$$

We define the orthonormal matrix with negative determinant corresponding to a reflection with respect to the horizontal axis as:

$$\xi(s = \text{-}1) = \begin{bmatrix} 1 & 0 \\ 0 & \text{-}1 \end{bmatrix}$$

and a general orthonormal matrix with negative determinant, i.e. reflection with respect to the axis $2\theta$, as:

$$\begin{bmatrix} \cos(\theta) & \sin(\theta) \\ \sin(\theta) & -\cos(\theta) \end{bmatrix} = \begin{bmatrix} \cos(\theta) & -\sin(\theta) \\ \sin(\theta) & \cos(\theta) \end{bmatrix} \begin{bmatrix} 1 & 0 \\ 0 & -1 \end{bmatrix}$$

Hence, we can express any orthonormal matrix in the form:

$$\begin{bmatrix} \cos{(\theta)} & -\sin{(\theta)} \\ \sin{(\theta)} & \cos{(\theta)} \end{bmatrix} \begin{bmatrix} 1 & 0 \\ 0 & s \end{bmatrix} = \psi(\theta)\xi(s)$$

where $\xi(s) = \begin{bmatrix} 1 & 0 \\ 0 & s \end{bmatrix}$ and $s \in (\{\pm 1\}, *)$.

Moreover, these properties will be useful later:

$$\psi(\theta)\xi(s) = \xi(s)\psi(s\theta) \tag{13}$$

$$\xi(s)^{-1} = \xi(s)^T = \xi(s) \tag{14}$$

$$\psi(\theta)^{-1} = \psi(\theta)^T = \psi(-\theta) \tag{15}$$

$$\psi(\theta_1)\psi(\theta_2) = \psi(\theta_1 + \theta_2) = \psi(\theta_2)\psi(\theta_1) \tag{16}$$

$$\mathrm{tr}(\psi(\theta)\xi(-1)) = \mathrm{tr}\begin{bmatrix} \cos{(\theta)} & \sin{(\theta)} \\ \sin{(\theta)} & -\cos{(\theta)} \end{bmatrix} = 0 \tag{17}$$

$$\mathrm{tr}(\psi(\theta)) = \mathrm{tr}\begin{bmatrix} \cos{(\theta)} & -\sin{(\theta)} \\ \sin{(\theta)} & \cos{(\theta)} \end{bmatrix} = 2\cos(\theta) \tag{18}$$

$$w_1\cos(\alpha) + w_2\sin(\alpha) = w_1\cos(\beta) + w_2\sin(\beta) \ \ \forall w_1, w_2 \in \mathbb{R}$$
$$\Leftrightarrow \quad \exists t \in \mathbb{Z} \text{ s.t. } \alpha = \beta + 2t\pi \tag{19}$$

## I.2 Irreducible representations of G≤O(2)

**Special Orthogonal group SO(2):**  SO(2) irreps would decompose into complex irreps of U(1) on the complex field but since we are implementing the theory with real-valued variables we will not consider these. Except for the trivial representation $\psi_0$, all the other irreps are 2-dimensional rotation matrices with frequencies $k \in \mathbb{N}^+$.

– $\psi_0^{\mathrm{SO}(2)}(r_\theta) = 1$

– $\psi_k^{\mathrm{SO}(2)}(r_\theta) = \begin{bmatrix} \cos{(k\theta)} & -\sin{(k\theta)} \\ \sin{(k\theta)} & \cos{(k\theta)} \end{bmatrix} = \psi(k\theta), \quad k \in \mathbb{N}^+$

**Orthogonal group O(2):**  O(2) has two 1-dimensional irreps: the trivial representation $\psi_{0,0}$ and a representation $\psi_{1,0}$ which assigns $\pm 1$ to reflections. The other representations are rotation matrices precomposed by a reflection.

– $\psi_{0,0}^{\mathrm{O}(2)}(r_\theta s) = 1$
– $\psi_{1,0}^{\mathrm{O}(2)}(r_\theta s) = s$
– $\psi_{1,k}^{\mathrm{O}(2)}(r_\theta s) = \begin{bmatrix} \cos{(k\theta)} & -\sin{(k\theta)} \\ \sin{(k\theta)} & \cos{(k\theta)} \end{bmatrix} \begin{bmatrix} 1 & 0 \\ 0 & s \end{bmatrix} = \psi(k\theta)\xi(s), \quad k \in \mathbb{N}^+$

**Cyclic groups C_N:**  The irreps of $C_N$ are identical to the irreps of SO(2) up to frequency $\lfloor N/2 \rfloor$. Due to the discreteness of rotation angles, higher frequencies would be aliased.

– $\psi_0^{C_N}(r_\theta) = 1$
– $\psi_k^{C_N}(r_\theta) = \begin{bmatrix} \cos{(k\theta)} & -\sin{(k\theta)} \\ \sin{(k\theta)} & \cos{(k\theta)} \end{bmatrix} = \psi(k\theta), \quad k \in \{1, \dots, \lfloor \frac{N-1}{2} \rfloor\}$

If $N$ is even, there is an additional 1-dimensional irrep corresponding to frequency $\lfloor \frac{N}{2} \rfloor = \frac{N}{2}$:

$$- \ \psi_{N/2}^{C_N}(r_\theta) = \cos\left(\tfrac{N}{2}\theta\right) \ \in \ \{\pm 1\} \text{ since } \theta \in \{p\tfrac{2\pi}{N}\}_{p=0}^{N-1}$$

**Dihedral groups $D_N$:** Similarly, $D_N$ consists of irreps of $O(2)$ up to frequency $\lfloor N/2 \rfloor$.

$$- \ \psi_{0,0}^{D_N}(r_\theta s) = 1$$

$$- \ \psi_{1,0}^{D_N}(r_\theta s) = s$$

$$- \ \psi_{1,k}^{D_N}(r_\theta s) = \begin{bmatrix} \cos(k\theta) & -\sin(k\theta) \\ \sin(k\theta) & \cos(k\theta) \end{bmatrix} \begin{bmatrix} 1 & 0 \\ 0 & s \end{bmatrix} = \psi(k\theta)\xi(s), \quad k \in \{1, \ldots, \lfloor \tfrac{N-1}{2} \rfloor\}$$

If $N$ is even, there are two 1-dimensional irreps:

$$- \ \psi_{0,N/2}^{D_N}(r_\theta s) = \ \ \cos\left(\tfrac{N}{2}\theta\right) \ \in \{\pm 1\} \quad \text{since } \theta \in \{p\tfrac{2\pi}{N}\}_{p=0}^{N-1}$$

$$- \ \psi_{1,N/2}^{D_N}(r_\theta s) = s\cos\left(\tfrac{N}{2}\theta\right) \ \in \{\pm 1\} \quad \text{since } \theta \in \{p\tfrac{2\pi}{N}\}_{p=0}^{N-1}$$

**Reflection group $(\{\pm 1\}, *) \cong D_1 \cong C_2$:** The reflection group $(\{\pm 1\}, *)$ is isomorphic to $D_1$ and $C_2$. For this reason, it has the same irreps of these two groups:

$$- \ \psi_0^{(\{\pm 1\}, *)}(s) = 1$$

$$- \ \psi_1^{(\{\pm 1\}, *)}(s) = s$$

## I.3 Analytical solutions of the irrep kernel constraints

### Special Orthogonal Group SO(2)

| $\psi_m \backslash \psi_n$ | $\psi_0$ | $\psi_n, n \in \mathbb{N}^+$ |
|---|---|---|
| $\psi_0$ | $\begin{bmatrix}1\end{bmatrix}$ | $\begin{bmatrix}\cos(n\phi) & \sin(n\phi)\end{bmatrix}, \begin{bmatrix}-\sin(n\phi) & \cos(n\phi)\end{bmatrix}$ |
| $\psi_m,$ $m \in \mathbb{N}^+$ | $\begin{bmatrix}\cos(m\phi)\\\sin(m\phi)\end{bmatrix},$ $\begin{bmatrix}-\sin(m\phi)\\\cos(m\phi)\end{bmatrix}$ | $\begin{bmatrix}\cos((m-n)\phi) & -\sin((m-n)\phi)\\\sin((m-n)\phi) & \cos((m-n)\phi)\end{bmatrix}, \begin{bmatrix}-\sin((m-n)\phi) & -\cos((m-n)\phi)\\\cos((m-n)\phi) & -\sin((m-n)\phi)\end{bmatrix},$ $\begin{bmatrix}\cos((m+n)\phi) & \sin((m+n)\phi)\\\sin((m+n)\phi) & -\cos((m+n)\phi)\end{bmatrix}, \begin{bmatrix}-\sin((m+n)\phi) & \cos((m+n)\phi)\\\cos((m+n)\phi) & \sin((m+n)\phi)\end{bmatrix}$ |

Table 8: Bases for the angular parts of SO(2)-steerable kernels satisfying the irrep constraint (3) for different pairs of input field irreps $\psi_n$ and output field irreps $\psi_m$. The different types of irreps are explained in I.2.

### Orthogonal Group O(2)

| $\psi_{i,m} \backslash \psi_{j,n}$ | $\psi_{0,0}$ | $\psi_{1,0}$ | $\psi_{1,n}, n \in \mathbb{N}^+$ |
|---|---|---|---|
| $\psi_{0,0}$ | $\begin{bmatrix}1\end{bmatrix}$ | $\varnothing$ | $\begin{bmatrix}-\sin(n\phi) & \cos(n\phi)\end{bmatrix}$ |
| $\psi_{1,0}$ | $\varnothing$ | $\begin{bmatrix}1\end{bmatrix}$ | $\begin{bmatrix}\cos(n\phi) & \sin(n\phi)\end{bmatrix}$ |
| $\psi_{1,m},$ $m \in \mathbb{N}^+$ | $\begin{bmatrix}-\sin(m\phi)\\\cos(m\phi)\end{bmatrix}$ | $\begin{bmatrix}\cos(m\phi)\\\sin(m\phi)\end{bmatrix}$ | $\begin{bmatrix}\cos((m-n)\phi) & -\sin((m-n)\phi)\\\sin((m-n)\phi) & \cos((m-n)\phi)\end{bmatrix}, \begin{bmatrix}\cos((m+n)\phi) & \sin((m+n)\phi)\\\sin((m+n)\phi) & -\cos((m+n)\phi)\end{bmatrix}$ |

Table 9: Bases for the angular parts of O(2)-steerable kernels satisfying the irrep constraint (3) for different pairs of input field irreps $\psi_{j,n}$ and output field irreps $\psi_{i,m}$. The different types of irreps are explained in I.2.

### Reflection group $(\{\pm 1\}, *)$

| $\psi_i \backslash \psi_j$ | $\psi_0$ | $\psi_1$ |
|---|---|---|
| $\psi_0$ | $\begin{bmatrix}\cos\left(\mu(\phi-\beta)\right)\end{bmatrix}$ | $\begin{bmatrix}\sin\left(\mu(\phi-\beta)\right)\end{bmatrix}$ |
| $\psi_1$ | $\begin{bmatrix}\sin\left(\mu(\phi-\beta)\right)\end{bmatrix}$ | $\begin{bmatrix}\cos\left(\mu(\phi-\beta)\right)\end{bmatrix}$ |

Table 10: Bases for the angular parts of $(\{\pm 1\}, *)$-steerable kernels satisfying the irrep constraint (3) for different pairs of input field irreps $\psi_j$ and output field irreps $\psi_i$ for $i, j \in \{0, 1\}$. The different types of irreps are explained in I.2. The group is assumed to act by reflecting over an axis defined by the angle $\beta$. Note that the bases shown here are a special case of the bases shown in Table 12 since $(\{\pm 1\}, *) \cong D_1$.

## Cyclic groups $C_N$

| $\psi_m$ \ $\psi_n$ | $\psi_0$ | $\psi_{N/2}$ (if $N$ even) | $\psi_n$ with $n \in \mathbb{N}^+$ and $1 \leq n < N/2$ | |
|---|---|---|---|---|
| $\psi_0$ | $\begin{bmatrix}\cos(\hat{t}N\phi)\end{bmatrix},$ $\begin{bmatrix}\sin(\hat{t}N\phi)\end{bmatrix}$ | $\begin{bmatrix}\cos\left(\frac{\hat{t}+1}{2}N\phi\right)\end{bmatrix},$ $\begin{bmatrix}\sin\left(\frac{\hat{t}+1}{2}N\phi\right)\end{bmatrix}$ | $\begin{bmatrix}-\sin((n+tN)\phi) & \cos((n+tN)\phi)\end{bmatrix},$ $\begin{bmatrix}\cos((n+tN)\phi) & \sin((n+tN)\phi)\end{bmatrix}$ | |
| $\psi_{N/2}$ ($N$ even) | $\begin{bmatrix}\cos\left(\frac{\hat{t}+1}{2}N\phi\right)\end{bmatrix},$ $\begin{bmatrix}\sin\left(\frac{\hat{t}+1}{2}N\phi\right)\end{bmatrix}$ | $\begin{bmatrix}\cos(\hat{t}N\phi)\end{bmatrix},$ $\begin{bmatrix}\sin(\hat{t}N\phi)\end{bmatrix}$ | $\begin{bmatrix}-\sin\left(\left(n+\frac{t+1}{2}N\right)\phi\right) & \cos\left(\left(n+\frac{t+1}{2}N\right)\phi\right)\end{bmatrix},$ $\begin{bmatrix}\cos\left(\left(n+\frac{t+1}{2}N\right)\phi\right) & \sin\left(\left(n+\frac{t+1}{2}N\right)\phi\right)\end{bmatrix}$ | |
| $\psi_m,$ $m \in \mathbb{N}^+$ $1 \leq m < N/2$ | $\begin{bmatrix}-\sin((m+tN)\phi)\\\cos((m+tN)\phi)\end{bmatrix},$ $\begin{bmatrix}\cos((m+tN)\phi)\\\sin((m+tN)\phi)\end{bmatrix}$ | $\begin{bmatrix}-\sin\left(\left(m+\frac{t+1}{2}N\right)\phi\right)\\\cos\left(\left(m+\frac{t+1}{2}N\right)\phi\right)\end{bmatrix},$ $\begin{bmatrix}\cos\left(\left(m+\frac{t+1}{2}N\right)\phi\right)\\\sin\left(\left(m+\frac{t+1}{2}N\right)\phi\right)\end{bmatrix}$ | $\begin{bmatrix}\cos((m-n+tN)\phi) & -\sin((m-n+tN)\phi)\\\sin((m-n+tN)\phi) & \cos((m-n+tN)\phi)\\\cos((m+n+tN)\phi) & \sin((m+n+tN)\phi)\\\sin((m+n+tN)\phi) & -\cos((m+n+tN)\phi)\end{bmatrix},$ | $\begin{bmatrix}-\sin((m-n+tN)\phi) & -\cos((m-n+tN)\phi)\\\cos((m-n+tN)\phi) & -\sin((m-n+tN)\phi)\\-\sin((m+n+tN)\phi) & -\cos((m+n+tN)\phi)\\\cos((m+n+tN)\phi) & -\sin((m+n+tN)\phi)\end{bmatrix}$ |

Table 11: Bases for the angular parts of $C_N$-steerable kernels for different pairs of input and output fields irreps $\psi_n$ and $\psi_m$. The full basis is found by instantiating these solutions for each $t \in \mathbb{Z}$ or $\hat{t} \in \mathbb{N}$. The different types of irreps are explained in Appendix I.2.

## Dihedral groups $D_N$

| $\psi_{i,m}$ \ $\psi_{j,n}$ | $\psi_{0,0}$ | $\psi_{1,0}$ | $\psi_{0,N/2}$ (if $N$ even) | $\psi_{1,N/2}$ (if $N$ even) | $\psi_{1,n}$ with $n \in \mathbb{N}^+$ and $1 \leq n < N/2$ |
|---|---|---|---|---|---|
| $\psi_{0,0}$ | $\begin{bmatrix}\cos(\hat{t}N\phi)\end{bmatrix}$ | $\begin{bmatrix}\sin(\hat{t}N\phi)\end{bmatrix}$ | $\begin{bmatrix}\cos\left(\frac{\hat{t}+1}{2}N\phi\right)\end{bmatrix}$ | $\begin{bmatrix}\sin\left(\frac{\hat{t}+1}{2}N\phi\right)\end{bmatrix}$ | $\begin{bmatrix}-\sin((n+tN)\phi) & \cos((n+tN)\phi)\end{bmatrix}$ |
| $\psi_{1,0}$ | $\begin{bmatrix}\sin(\hat{t}N\phi)\end{bmatrix}$ | $\begin{bmatrix}\cos(\hat{t}N\phi)\end{bmatrix}$ | $\begin{bmatrix}\sin\left(\frac{\hat{t}+1}{2}N\phi\right)\end{bmatrix}$ | $\begin{bmatrix}\cos\left(\frac{\hat{t}+1}{2}N\phi\right)\end{bmatrix}$ | $\begin{bmatrix}\cos((n+tN)\phi) & \sin((n+tN)\phi)\end{bmatrix}$ |
| $\psi_{0,N/2}$ ($N$ even) | $\begin{bmatrix}\cos\left(\frac{\hat{t}+1}{2}N\phi\right)\end{bmatrix}$ | $\begin{bmatrix}\sin\left(\frac{\hat{t}+1}{2}N\phi\right)\end{bmatrix}$ | $\begin{bmatrix}\cos(\hat{t}N\phi)\end{bmatrix}$ | $\begin{bmatrix}\sin(\hat{t}N\phi)\end{bmatrix}$ | $\begin{bmatrix}-\sin\left(\left(n+\frac{t+1}{2}N\right)\phi\right) & \cos\left(\left(n+\frac{t+1}{2}N\right)\phi\right)\end{bmatrix}$ |
| $\psi_{1,N/2}$ ($N$ even) | $\begin{bmatrix}\sin\left(\frac{\hat{t}+1}{2}N\phi\right)\end{bmatrix}$ | $\begin{bmatrix}\cos\left(\frac{\hat{t}+1}{2}N\phi\right)\end{bmatrix}$ | $\begin{bmatrix}\sin(\hat{t}N\phi)\end{bmatrix}$ | $\begin{bmatrix}\cos(\hat{t}N\phi)\end{bmatrix}$ | $\begin{bmatrix}\cos\left(\left(n+\frac{t+1}{2}N\right)\phi\right) & \sin\left(\left(n+\frac{t+1}{2}N\right)\phi\right)\end{bmatrix}$ |
| $\psi_{1,m},$ $m \in \mathbb{N}^+$ $1 \leq m < N/2$ | $\begin{bmatrix}-\sin((m+tN)\phi)\\\cos((m+tN)\phi)\end{bmatrix}$ | $\begin{bmatrix}\cos((m+tN)\phi)\\\sin((m+tN)\phi)\end{bmatrix}$ | $\begin{bmatrix}-\sin\left(\left(m+\frac{t+1}{2}N\right)\phi\right)\\\cos\left(\left(m+\frac{t+1}{2}N\right)\phi\right)\end{bmatrix}$ | $\begin{bmatrix}\cos\left(\left(m+\frac{t+1}{2}N\right)\phi\right)\\\sin\left(\left(m+\frac{t+1}{2}N\right)\phi\right)\end{bmatrix}$ | $\begin{bmatrix}\cos((m-n+tN)\phi) & -\sin((m-n+tN)\phi)\\\sin((m-n+tN)\phi) & \cos((m-n+tN)\phi)\end{bmatrix},$ $\begin{bmatrix}\cos((m+n+tN)\phi) & \sin((m+n+tN)\phi)\\\sin((m+n+tN)\phi) & -\cos((m+n+tN)\phi)\end{bmatrix}$ |

Table 12: Bases for the angular parts of $D_N$-steerable kernels for different pairs of input and output fields irreps $\psi_{j,n}$ and $\psi_{i,m}$. The full basis is found by instantiating these solutions for each $t \in \mathbb{Z}$ or $\hat{t} \in \mathbb{N}$. The different types of irreps are explained in Appendix I.2. The solutions here shown are for a group action where the reflection is defined around the horizontal axis ($\beta = 0$). For different axes $\beta \neq 0$ substitute $\phi$ with $\phi - \beta$.

## I.4 Derivations of the kernel constraints

Here we solve the kernel constraints for the irreducible representations of $G \leq O(2)$. Since the irreps of $G$ are either 1- or 2-dimensional, we distinguish between mappings between 2-dimensional irreps, mappings from 2- to 1-dimensional and 1- to 2-dimensional irreps and mappings between 1-dimensional irreps. We are first exclusively considering positive radial parts $r > 0$ in the following sections. The constraint at the origin $r = 0$ requires some additional considerations which we postpone to Section I.4.6.

### I.4.1 Derivation for SO(2)

**2-dimensional irreps:**

We first consider the case of 2-dimensional irreps both in the input and in output, that is, $\rho_{\text{out}} = \psi_m^{\text{SO}(2)}$ and $\rho_{\text{in}} = \psi_n^{\text{SO}(2)}$, where $\psi_k^{\text{SO}(2)}(\theta) = \begin{bmatrix} \cos(k\theta) & -\sin(k\theta) \\ \sin(k\theta) & \cos(k\theta) \end{bmatrix}$. This means that the kernel has the form $\kappa^{ij} : \mathbb{R}^2 \to \mathbb{R}^{2\times 2}$. To reduce clutter we will from now on suppress the indices $ij$ corresponding to the input and output irreps in the input and output fields.

We expand each entry of the kernel $\kappa$ in terms of an (angular) Fourier series[14]

$$\kappa(r, \phi) = \sum_{\mu=0}^{\infty} A_{00,\mu}(r) \begin{bmatrix} \cos(\mu\phi) & 0 \\ 0 & 0 \end{bmatrix} + B_{00,\mu}(r) \begin{bmatrix} \sin(\mu\phi) & 0 \\ 0 & 0 \end{bmatrix}$$

$$+ A_{01,\mu}(r) \begin{bmatrix} 0 & \cos(\mu\phi) \\ 0 & 0 \end{bmatrix} + B_{01,\mu}(r) \begin{bmatrix} 0 & \sin(\mu\phi) \\ 0 & 0 \end{bmatrix}$$

$$+ A_{10,\mu}(r) \begin{bmatrix} 0 & 0 \\ \cos(\mu\phi) & 0 \end{bmatrix} + B_{10,\mu}(r) \begin{bmatrix} 0 & 0 \\ \sin(\mu\phi) & 0 \end{bmatrix}$$

$$+ A_{11,\mu}(r) \begin{bmatrix} 0 & 0 \\ 0 & \cos(\mu\phi) \end{bmatrix} + B_{11,\mu}(r) \begin{bmatrix} 0 & 0 \\ 0 & \sin(\mu\phi) \end{bmatrix}$$

and, for convenience, perform a change of basis to a different, non-sparse, orthogonal basis

$$\kappa(r, \phi) = \sum_{\mu=0}^{\infty} w_{0,0,\mu}(r) \begin{bmatrix} \cos(\mu\phi) & -\sin(\mu\phi) \\ \sin(\mu\phi) & \cos(\mu\phi) \end{bmatrix} + w_{0,1,\mu}(r) \begin{bmatrix} \cos(\mu\phi + \frac{\pi}{2}) & -\sin(\mu\phi + \frac{\pi}{2}) \\ \sin(\mu\phi + \frac{\pi}{2}) & \cos(\mu\phi + \frac{\pi}{2}) \end{bmatrix}$$

$$+ w_{1,0,\mu}(r) \begin{bmatrix} \cos(\mu\phi) & \sin(\mu\phi) \\ \sin(\mu\phi) & -\cos(\mu\phi) \end{bmatrix} + w_{1,1,\mu}(r) \begin{bmatrix} \cos(\mu\phi + \frac{\pi}{2}) & \sin(\mu\phi + \frac{\pi}{2}) \\ \sin(\mu\phi + \frac{\pi}{2}) & -\cos(\mu\phi + \frac{\pi}{2}) \end{bmatrix}$$

$$+ w_{2,0,\mu}(r) \begin{bmatrix} \cos(-\mu\phi) & -\sin(-\mu\phi) \\ \sin(-\mu\phi) & \cos(-\mu\phi) \end{bmatrix} + w_{2,1,\mu}(r) \begin{bmatrix} \cos(-\mu\phi + \frac{\pi}{2}) & -\sin(-\mu\phi + \frac{\pi}{2}) \\ \sin(-\mu\phi + \frac{\pi}{2}) & \cos(-\mu\phi + \frac{\pi}{2}) \end{bmatrix}$$

$$+ w_{3,0,\mu}(r) \begin{bmatrix} \cos(-\mu\phi) & \sin(-\mu\phi) \\ \sin(-\mu\phi) & -\cos(-\mu\phi) \end{bmatrix} + w_{3,1,\mu}(r) \begin{bmatrix} \cos(-\mu\phi + \frac{\pi}{2}) & \sin(-\mu\phi + \frac{\pi}{2}) \\ \sin(-\mu\phi + \frac{\pi}{2}) & -\cos(-\mu\phi + \frac{\pi}{2}) \end{bmatrix}.$$

The last four matrices are equal to the first four, except for their opposite frequency. Moreover, the second matrices of each row are equal to the first matrices, with a phase shift of $\frac{\pi}{2}$ added. Therefore, we can as well write:

$$\kappa(r,\phi) = \sum_{\mu=-\infty}^{\infty} \sum_{\gamma \in \{0, \frac{\pi}{2}\}} w_{0,\gamma,\mu}(r) \begin{bmatrix} \cos(\mu\phi+\gamma) & -\sin(\mu\phi+\gamma) \\ \sin(\mu\phi+\gamma) & \cos(\mu\phi+\gamma) \end{bmatrix} + w_{1,\gamma,\mu}(r) \begin{bmatrix} \cos(\mu\phi+\gamma) & \sin(\mu\phi+\gamma) \\ \sin(\mu\phi+\gamma) & -\cos(\mu\phi+\gamma) \end{bmatrix}$$

Notice that the first matrix evaluates to $\psi(\mu\phi + \gamma)\xi(1) = \psi(\mu\phi + \gamma)$ while the second evaluates to $\psi(\mu\phi + \gamma)\xi(-1)$. Hence, for $s \in \{\pm 1\}$ we can compactly write:

$$\kappa(r, \phi) = \sum_{\mu=-\infty}^{\infty} \sum_{\gamma \in \{0, \frac{\pi}{2}\}} \sum_{s \in \{\pm 1\}} w_{s,\gamma,\mu}(r)\psi(\mu\phi + \gamma)\xi(s)$$

As already shown in Section 2.5, we can w.l.o.g. consider the kernels as being defined only on the angular component $\phi \in [0, 2\pi) = S^1$ by solving only for a specific radial component $r$. As a result, we consider the basis

$$\left\{ b_{\mu,\gamma,s}(\phi) = \psi(\mu\phi + \gamma)\xi(s) \,\middle|\, \mu \in \mathbb{Z}, \, \gamma \in \left\{0, \frac{\pi}{2}\right\}, \, s \in (\{\pm 1\}, *) \right\} \tag{20}$$

of the unrestricted kernel space which we will constrain in the following by demanding

$$\kappa(\phi + \theta) = \psi_m^{SO(2)}(r_\theta)\kappa(\phi)\psi_n^{SO(2)}(r_\theta)^{-1} \quad \forall \phi, \theta \in [0, 2\pi), \tag{21}$$

where we dropped the unrestricted radial part.

We solve for a basis of the subspace satisfying this constraint by projecting both sides on the basis elements defined above. The inner product on $L^2\left(S^1\right)^{2\times 2}$ is hereby defined as

$$\langle k_1, k_2 \rangle = \frac{1}{4\pi} \int d\phi \langle k_1(\phi), k_2(\phi) \rangle_F = \frac{1}{4\pi} \int d\phi \; \mathrm{tr}\left(k_1(\phi)^T k_2(\phi)\right),$$

where $\langle \cdot, \cdot \rangle_F$ denotes the *Frobenius* inner product between 2 matrices.

First consider the projection of the *lhs* of the kernel constraint (21) on a generic basis element $b_{\mu',\gamma',s'}(\phi) = \psi(\mu'\phi + \gamma')\xi(s')$. Defining the operator $R_\theta$ by $(R_\theta \kappa)(\phi) := \kappa(\phi + \theta)$, the projection gives:

$$\langle b_{\mu',\gamma',s'}, R_\theta \kappa \rangle = \frac{1}{4\pi} \int d\phi \; \mathrm{tr}\left(b_{\mu',\gamma',s'}(\phi)^T (R_\theta \kappa)(\phi)\right)$$

$$= \frac{1}{4\pi} \int d\phi \; \mathrm{tr}\left(b_{\mu',\gamma',s'}(\phi)^T \kappa(\phi + \theta)\right).$$

By expanding the kernel in the linear combination of the basis we further obtain

$$= \frac{1}{4\pi} \int d\phi \; \mathrm{tr}\left(b_{\mu',\gamma',s'}(\phi)^T \left(\sum_\mu \sum_\gamma \sum_s w_{s,\gamma,\mu}\psi\left(\mu(\phi + \theta) + \gamma\right)\xi(s)\right)\right),$$

which, observing that the trace, sums and integral commute, results in:

$$= \frac{1}{4\pi} \sum_\mu \sum_\gamma \sum_s w_{s,\gamma,\mu} \, \mathrm{tr}\left(\int d\phi \; b_{\mu',\gamma',s'}(\phi)^T \psi\left(\mu(\phi + \theta) + \gamma\right)\xi(s)\right)$$

$$= \frac{1}{4\pi} \sum_\mu \sum_\gamma \sum_s w_{s,\gamma,\mu} \, \mathrm{tr}\left(\int d\phi \; (\psi(\mu'\phi + \gamma')\xi(s'))^T \psi\left(\mu(\phi + \theta) + \gamma\right)\xi(s)\right)$$

$$= \frac{1}{4\pi} \sum_\mu \sum_\gamma \sum_s w_{s,\gamma,\mu} \, \mathrm{tr}\left(\int d\phi \; \xi(s')^T \psi(\mu'\phi + \gamma')^T \psi\left(\mu(\phi + \theta) + \gamma\right)\xi(s)\right)$$

Using the properties in Eq. (14) and (15) then yields:

$$= \frac{1}{4\pi} \sum_\mu \sum_\gamma \sum_s w_{s,\gamma,\mu} \, \mathrm{tr}\left(\int d\phi \; \xi(s')\psi(-\mu'\phi - \gamma')\psi\left(\mu(\phi + \theta) + \gamma\right)\xi(s)\right)$$

$$= \frac{1}{2} \sum_\mu \sum_\gamma \sum_s w_{s,\gamma,\mu} \, \mathrm{tr}\left(\xi(s')\psi(\gamma - \gamma')\left(\frac{1}{2\pi}\int d\phi \; \psi((\mu - \mu')\phi)\right)\psi(\mu\theta)\xi(s)\right)$$

In the integral, each cell of the matrix $\psi((\mu - \mu')\phi)$ contains either a sine or cosine. As a result, if $\mu - \mu' \neq 0$, all these integrals evaluate to 0. Otherwise, the cosines on the diagonal evaluate to 1, while the sines integrate to 0. The whole integral evaluates to $\delta_{\mu,\mu'} \, \mathrm{id}_{2\times 2}$, such that

$$= \frac{1}{2} \sum_\gamma \sum_s w_{s,\gamma,\mu'} \, \mathrm{tr}\left(\xi(s')\psi(\gamma - \gamma')\psi(\mu'\theta)\xi(s)\right),$$

which, using the property in Eq. (13) leads to

$$= \frac{1}{2} \sum_\gamma \sum_s w_{s,\gamma,\mu'} \, \mathrm{tr}\left(\psi(s'(\gamma - \gamma' + \mu'\theta))\xi(s' * s)\right).$$

Recall the propeties of the trace in Eq. (17), (18). If $s' * s = -1$, i.e. $s' \neq s$, the matrix has a trace of 0:

$$= \frac{1}{2} \sum_{\gamma} \sum_{s} w_{s,\gamma,\mu'} \delta_{s',s} 2 \cos(s'(\gamma - \gamma' + \mu'\theta))$$

Since $\cos(-\alpha) = \cos(\alpha)$ and $s' \in \{\pm 1\}$:

$$= \frac{1}{2} \sum_{\gamma} \sum_{s} w_{s,\gamma,\mu'} \delta_{s',s} 2 \cos(\gamma - \gamma' + \mu'\theta)$$

$$= \sum_{\gamma} w_{s',\gamma,\mu'} \cos((\gamma - \gamma') + \mu'\theta)$$

Next consider the projection of the *rhs* of Eq. (21):

$$\langle b_{\mu',\gamma',s'}, \ \psi_m^{SO(2)}(r_\theta)\kappa(\cdot)\psi_n^{SO(2)}(r_\theta)^{-1}\rangle$$

$$= \frac{1}{4\pi} \int d\phi \ \mathrm{tr} \left( b_{\mu',\gamma',s'}(\phi)^T \psi_m^{SO(2)}(r_\theta)\kappa(\phi)\psi_n^{SO(2)}(r_\theta)^{-1} \right)$$

$$= \frac{1}{4\pi} \int d\phi \ \mathrm{tr} \left( b_{\mu',\gamma',s'}(\phi)^T \psi(m\theta)\kappa(\phi)\psi(-n\theta) \right)$$

An expansion of the kernel in the linear combination of the basis yields:

$$= \frac{1}{4\pi} \int d\phi \ \mathrm{tr} \left( b_{\mu',\gamma',s'}(\phi)^T \psi(m\theta) \left( \sum_{\mu} \sum_{\gamma} \sum_{s} w_{s,\gamma,\mu}\psi(\mu\phi + \gamma)\xi(s) \right) \psi(-n\theta) \right)$$

$$= \frac{1}{4\pi} \sum_{\mu} \sum_{\gamma} \sum_{s} w_{s,\gamma,\mu} \ \mathrm{tr} \left( \int d\phi \ b_{\mu',\gamma',s'}(\phi)^T \psi(m\theta)\psi(\mu\phi + \gamma)\xi(s)\psi(-n\theta) \right)$$

$$= \frac{1}{4\pi} \sum_{\mu} \sum_{\gamma} \sum_{s} w_{s,\gamma,\mu} \ \mathrm{tr} \left( \int d\phi \ \xi(s')\psi(-\mu'\phi - \gamma')\psi(m\theta)\psi(\mu\phi + \gamma)\xi(s)\psi(-n\theta) \right)$$

$$= \frac{1}{2} \sum_{\mu} \sum_{\gamma} \sum_{s} w_{s,\gamma,\mu} \ \mathrm{tr} \left( \xi(s')\psi(\gamma - \gamma')\psi(m\theta) \left( \frac{1}{2\pi} \int d\phi \ \psi((\mu - \mu')\phi) \right) \xi(s)\psi(-n\theta) \right)$$

Again, the integral evaluates to $\delta_{\mu,\mu'} \mathrm{id}_{2\times 2}$:

$$= \frac{1}{2} \sum_{\mu} \sum_{\gamma} \sum_{s} w_{s,\gamma,\mu}\delta_{\mu,\mu'} \ \mathrm{tr} \left( \xi(s')\psi(\gamma - \gamma')\psi(m\theta)\xi(s)\psi(-n\theta) \right)$$

$$= \frac{1}{2} \sum_{\gamma} \sum_{s} w_{s,\gamma,\mu'} \ \mathrm{tr} \left( \xi(s')\psi(\gamma - \gamma')\psi(m\theta)\xi(s)\psi(-n\theta) \right)$$

$$= \frac{1}{2} \sum_{\gamma} \sum_{s} w_{s,\gamma,\mu'} \ \mathrm{tr} \left( \psi(s'(\gamma - \gamma' + m\theta - ns\theta))\xi(s' * s) \right)$$

For the same reason as before, the trace is not zero if and only if $s' = s$:

$$= \frac{1}{2} \sum_{\gamma} \sum_{s} w_{s,\gamma,\mu'}\delta_{s',s} 2 \cos(s'(\gamma - \gamma' + m\theta - ns\theta))$$

Since $\cos(-\alpha) = \cos(\alpha)$ and $s' \in \{\pm 1\}$:

$$= \sum_{\gamma} w_{s',\gamma,\mu'} \cos(\gamma - \gamma' + m\theta - ns'\theta)$$

$$= \sum_{\gamma} w_{s',\gamma,\mu'} \cos((\gamma - \gamma') + (m - ns')\theta)$$

Finally, we require the two projections to be equal for all rotations in $SO(2)$, that is,

$$\sum_{\gamma} w_{s',\gamma,\mu'} \cos((\gamma - \gamma') + \mu'\theta) = \sum_{\gamma} w_{s',\gamma,\mu'} \cos((\gamma - \gamma') + (m - ns')\theta) \qquad \forall \theta \in [0, 2\pi),$$

or, explicitly, with $\gamma \in \{0, \frac{\pi}{2}\}$ and $\cos(\alpha + \frac{\pi}{2}) = -\sin(\alpha)$:

$$
\begin{aligned}
& w_{s',0,\mu'} \cos(\mu'\theta - \gamma') && - \; w_{s',\frac{\pi}{2},\mu'} \sin(\mu'\theta - \gamma') \\
&= w_{s',0,\mu'} \cos((m - ns')\theta - \gamma') &&- \; w_{s',\frac{\pi}{2},\mu'} \sin((m - ns')\theta - \gamma') && \forall \theta \in [0, 2\pi)
\end{aligned}
$$

Using the property in Eq. (19) then implies that for each $\theta$ in $[0, 2\pi)$ there exists a $t \in \mathbb{Z}$ such that:

$$
\begin{aligned}
\Leftrightarrow && \mu'\theta - \gamma' &= (m - ns')\theta - \gamma' + 2t\pi \\
\Leftrightarrow && (\mu' - (m - ns'))\theta &= 2t\pi && (22)
\end{aligned}
$$

Since the constraint needs to hold for any $\theta \in [0, 2\pi)$ this results in the condition $\mu' = m - sn'$ on the frequencies occurring in the SO(2)-steerable kernel basis. Both $\gamma$ and $s$ are left unrestricted such that we end up with the four-dimensional basis

$$
\mathcal{K}^{\mathrm{SO}(2)}_{\psi_m \leftarrow \psi_n} = \left\{ b_{\mu,\gamma,s}(\phi) = \psi(\mu\phi + \gamma)\xi(s) \,\middle|\, \mu = (m - sn),\ \gamma \in \left\{0, \frac{\pi}{2}\right\},\ s \in \{\pm 1\} \right\} \quad (23)
$$

for the angular parts of equivariant kernels for $m, n > 0$. This basis is explicitly written out in the lower right cell of Table 8.

**1-dimensional irreps:**

For the case of 1-dimensional irreps in both the input and output, i.e. $\rho_{\mathrm{out}} = \rho_{\mathrm{in}} = \psi_0^{\mathrm{SO}(2)}$ the kernel has the form $\kappa^{ij} : \mathbb{R}^2 \to \mathbb{R}^{1 \times 1}$. As a scalar function in $L^2(\mathbb{R}^2)$, it can be expressed by the Fourier decomposition of its angular part:

$$
\kappa(r, \phi) = w_{0,0} + \sum_{\mu=1}^{\infty} \sum_{\gamma \in \{0, \frac{\pi}{2}\}} w_{\mu,\gamma}(r) \cos(\mu\phi + \gamma)
$$

As before, we can w.l.o.g. drop the dependency on the radial part as it is not restricted by the constraint. We are therefore considering the basis

$$
\left\{ b_{\mu,\gamma}(\phi) = \cos(\mu\phi + \gamma) \,\middle|\, \mu \in \mathbb{N},\ \gamma \in \begin{cases} \{0\} & \text{if } \mu = 0 \\ \{0, \pi/2\} & \text{otherwise} \end{cases} \right\} \quad (24)
$$

of angular kernels in $L^2(S^1)^{1 \times 1}$. The kernel constraint in Eq. (3) then requires

$$
\begin{aligned}
\kappa(\phi + \theta) &= \psi_m^{\mathrm{SO}(2)}(r_\theta)\kappa(\phi)\psi_n^{\mathrm{SO}(2)}(r_\theta)^{-1} && \forall \theta, \phi \in [0, 2\pi) \\
\Leftrightarrow \quad \kappa(\phi + \theta) &= \kappa(\phi) && \forall \theta, \phi \in [0, 2\pi),
\end{aligned}
$$

i.e. the kernel has to be *invariant* to rotations.

Again, we find the space of all solutions by projecting both sides on the basis defined above. Here, the projection of two kernels is defined through the standard inner product $\langle k_1, k_2 \rangle = \frac{1}{2\pi} \int d\phi \, k_1(\phi) k_2(\phi)$ on $L^2(S^1)$.

We first consider the projection of the *lhs*:

$$
\begin{aligned}
\langle b_{\mu',\gamma'},\ R_\theta \kappa \rangle &= \frac{1}{2\pi} \int d\phi \, b_{\mu',\gamma'}(\phi) \, (R_\theta \kappa)\,(\phi) \\
&= \frac{1}{2\pi} \int d\phi \, b_{\mu',\gamma'}(\phi) \kappa(\phi + \theta)
\end{aligned}
$$

As before we expand the kernel in the linear combination of the basis:

$$
\begin{aligned}
&= \sum_{\mu,\gamma} w_{\mu,\gamma} \frac{1}{2\pi} \int d\phi \, b_{\mu',\gamma'}(\phi) \cos(\mu\phi + \mu\theta + \gamma) \\
&= \sum_{\mu,\gamma} w_{\mu,\gamma} \frac{1}{2\pi} \int d\phi \, \cos(\mu'\phi + \gamma') \cos(\mu\phi + \mu\theta + \gamma)
\end{aligned}
$$

With $\cos(\alpha)\cos(\beta) = \frac{1}{2}\left(\cos(\alpha - \beta) + \cos(\alpha + \beta)\right)$ this results in:

$$= \sum_{\mu,\gamma} w_{\mu,\gamma}\frac{1}{2\pi}\int d\phi\,\frac{1}{2}\Big(\cos(\mu'\phi + \gamma' - \mu\phi - \mu\theta - \gamma)$$

$$+ \cos(\mu'\phi + \gamma' + \mu\phi + \mu\theta + \gamma)\Big)$$

$$= \sum_{\mu,\gamma} w_{\mu,\gamma}\frac{1}{2}\Big(\frac{1}{2\pi}\int d\phi\,\cos((\mu' - \mu)\phi + (\gamma' - \gamma) - \mu\theta)$$

$$+ \frac{1}{2\pi}\int d\phi\,\cos((\mu' + \mu)\phi + (\gamma' + \gamma) + \mu\theta)\Big)$$

$$= \sum_{\mu,\gamma} w_{\mu,\gamma}\frac{1}{2}\left(\delta_{\mu,\mu'}\cos((\gamma' - \gamma) - \mu\theta) + \delta_{\mu,-\mu'}\cos((\gamma' + \gamma) + \mu\theta)\right)$$

Since $\mu, \mu' \geq 0$ and $\mu = -\mu'$ imply $\mu = \mu' = 0$ this simplifies further to

$$= \frac{1}{2}\sum_{\gamma} w_{\mu',\gamma}\left(\cos((\gamma' - \gamma) - \mu'\theta) + \delta_{\mu',0}\cos(\gamma' + \gamma)\right).$$

A projection of the *rhs* yields:

$$\langle b_{\mu',\gamma'},\,\kappa\rangle = \frac{1}{2\pi}\int d\phi\, b_{\mu',\gamma'}(\phi)\kappa(\phi)$$

$$= \sum_{\mu,\gamma} w_{\mu,\gamma}\frac{1}{2\pi}\int d\phi\, b_{\mu',\gamma'}(\phi)\cos(\mu\phi + \gamma)$$

$$= \sum_{\mu,\gamma} w_{\mu,\gamma}\frac{1}{2\pi}\int d\phi\,\cos(\mu'\phi + \gamma')\cos(\mu\phi + \gamma)$$

$$= \frac{1}{2}\sum_{\gamma} w_{\mu',\gamma}\left(\cos(\gamma' - \gamma) + \delta_{\mu',0}\cos((\gamma' + \gamma))\right)$$

The projections are required to coincide for all rotations:

$$\langle b_{\mu',\gamma'}, R_\theta\kappa\rangle = \langle b_{\mu',\gamma'},\kappa\rangle \qquad\qquad \forall\theta \in [0,2\pi)$$

$$\sum_{\gamma} w_{\mu',\gamma}\left(\cos((\gamma' - \gamma) - \mu'\theta) + \delta_{\mu',0}\cos((\gamma' + \gamma))\right) = \sum_{\gamma} w_{\mu',\gamma}\left(\cos(\gamma' - \gamma) + \delta_{\mu',0}\cos((\gamma' + \gamma))\right)$$

$$\forall\theta \in [0,2\pi)$$

We consider two cases:

- $\mu' = 0$  In this case, the basis in Eq.(24) is restricted to the single case $\gamma' = 0$ (as $\gamma' = \frac{\pi}{2}$ and $\mu' = 0$ together lead to a null basis element). Then:

$$\sum_{\gamma} w_{0,\gamma}\left(\cos(-\gamma) + \cos(\gamma)\right) = \sum_{\gamma} w_{0,\gamma}\left(\cos(-\gamma) + \cos(\gamma)\right)$$

  As $\gamma \in \{0, \frac{\pi}{2}\}$ and $\cos(\pm\frac{\pi}{2}) = 0$:

$$\Leftrightarrow \qquad\qquad w_{0,0}\left(\cos(0) + \cos(0)\right) = w_{0,0}\left(\cos(0) + \cos(0)\right)$$

$$\Leftrightarrow \qquad\qquad w_{0,0} = w_{0,0}$$

  which is always true.

- $\mu' > 0$  Here:

$$\sum_{\gamma} w_{\mu',\gamma}\cos((\gamma' - \gamma) - \mu'\theta) = \sum_{\gamma} w_{\mu',\gamma}\cos(\gamma' - \gamma) \qquad \forall\theta \in [0,2\pi)$$

$$\Leftrightarrow \quad w_{\mu',0}\cos(\gamma' - \mu'\theta) + w_{\mu',\frac{\pi}{2}}\sin(\gamma' - \mu'\theta) = w_{\mu',0}\cos(\gamma') + w_{\mu',\frac{\pi}{2}}\sin(\gamma') \quad \forall\theta \in [0,2\pi)$$

$$\Leftrightarrow \qquad\qquad -\mu'\theta = 2t\pi \qquad\qquad \forall\theta \in [0,2\pi),$$

  where Eq. (19) was used in the last step. From the last equation one can see that $\mu'$ must be zero. Since this contradicts the assumption that $\mu' \geq 0$, no solution exists.

This results in a one dimensional basis of isotropic (rotation invariant) kernels

$$\mathcal{K}^{\text{SO}(2)}_{\psi_m \leftarrow \psi_n} = \{b_{0,0}(\phi) = 1\} \tag{25}$$

for $m = n = 0$, i.e. trivial representations. The basis is presented in the upper left cell of Table 8.

**1 and 2-dimensional irreps:**

Finally, consider the case of a 1-dimensional irrep in the input and a 2-dimensional irrep in the output, that is, $\rho_{\text{out}} = \psi_m^{\text{SO}(2)}$ and $\rho_{\text{in}} = \psi_0^{\text{SO}(2)}$. The corresponding kernel $\kappa^{ij} : \mathbb{R}^2 \to \mathbb{R}^{2\times1}$ can be expanded in the following generalized Fourier series on $L^2(\mathbb{R}^2)^{2\times1}$:

$$\kappa(r, \phi) = \sum_{\mu=0}^{\infty} A_{0,\mu}(r) \begin{bmatrix} \cos(\mu\phi) \\ 0 \end{bmatrix} + B_{0,\mu}(r) \begin{bmatrix} \sin(\mu\phi) \\ 0 \end{bmatrix}$$

$$+ A_{1,\mu}(r) \begin{bmatrix} 0 \\ \cos(\mu\phi) \end{bmatrix} + B_{1,\mu}(r) \begin{bmatrix} 0 \\ \sin(\mu\phi) \end{bmatrix}$$

As before, we perform a change of basis to produce a non-sparse basis

$$\kappa(r, \phi) = \sum_{\mu=-\infty}^{\infty} \sum_{\gamma \in \{0, \frac{\pi}{2}\}} w_{\gamma,\mu}(r) \begin{bmatrix} \cos(\mu\phi + \gamma) \\ \sin(\mu\phi + \gamma) \end{bmatrix}.$$

Dropping the radial parts as usual, this corresponds to the complete basis :

$$\left\{ b_{\mu,\gamma}(\phi) = \begin{bmatrix} \cos(\mu\phi + \gamma) \\ \sin(\mu\phi + \gamma) \end{bmatrix} \;\middle|\; \mu \in \mathbb{Z}, \; \gamma \in \left\{0, \frac{\pi}{2}\right\} \right\} \tag{26}$$

of angular kernels on $L^2(S^1)^{2\times1}$.

The constraint in Eq. (3) requires the kernel space to satisfy

$$\kappa(\phi + \theta) = \psi_m^{\text{SO}(2)}(r_\theta)\kappa(\phi)\psi_0^{\text{SO}(2)}(r_\theta)^{-1} \qquad \forall \theta, \phi \in [0, 2\pi)$$

$$\Leftrightarrow \quad \kappa(\phi + \theta) = \psi_m^{\text{SO}(2)}(r_\theta)\kappa(\phi) \qquad \qquad \forall \theta, \phi \in [0, 2\pi).$$

We again project both sides of this equation on the basis elements defined above where the projection on $L^2(S^1)^{2\times1}$ is defined by $\langle k_1, k_2 \rangle = \frac{1}{2\pi} \int d\phi \, k_1(\phi)^T k_2(\phi)$.

Consider first the projection of the *lhs*

$$\langle b_{\mu',\gamma'}, \, R_\theta \kappa \rangle = \frac{1}{2\pi} \int d\phi \, b_{\mu',\gamma'}(\phi)^T \, (R_\theta \kappa) \, (\phi)$$

$$= \frac{1}{2\pi} \int d\phi \, b_{\mu',\gamma'}(\phi)^T \kappa(\phi + \theta) \,,$$

which, after expanding the kernel in terms of the basis reads:

$$= \sum_{\mu,\gamma} w_{\mu,\gamma} \frac{1}{2\pi} \int d\phi \, b_{\mu',\gamma'}(\phi)^T \begin{bmatrix} \cos(\mu(\phi + \theta) + \gamma) \\ \sin(\mu(\phi + \theta) + \gamma) \end{bmatrix}$$

$$= \sum_{\mu,\gamma} w_{\mu,\gamma} \frac{1}{2\pi} \int d\phi \, [\cos(\mu'\phi + \gamma') \quad \sin(\mu'\phi + \gamma')] \begin{bmatrix} \cos(\mu(\phi + \theta) + \gamma) \\ \sin(\mu(\phi + \theta) + \gamma) \end{bmatrix}$$

$$= \sum_{\mu,\gamma} w_{\mu,\gamma} \frac{1}{2\pi} \int d\phi \, \cos((\mu' - \mu)\phi + (\gamma' - \gamma) - \mu\theta).$$

As before, the integral is non-zero only if the frequency is 0, i.e. iff $\mu' - \mu = 0$ and thus:

$$= \sum_{\gamma} w_{\mu',\gamma} \cos((\gamma' - \gamma) - \mu'\theta)$$

For the *rhs* we obtain:

$$
\langle b_{\mu',\gamma'},\ \psi_m^{\mathrm{SO}(2)}(r_\theta)\kappa(\cdot)\rangle
$$

$$
= \frac{1}{2\pi}\int d\phi\, b_{\mu',\gamma'}(\phi)^T \psi_m(r_\theta)\kappa(\phi)
$$

$$
= \sum_{\mu,\gamma} w_{\mu,\gamma}\frac{1}{2\pi}\int d\phi\, b_{\mu',\gamma'}(\phi)^T \psi_m(r_\theta)\begin{bmatrix}\cos(\mu\phi+\gamma)\\\sin(\mu\phi+\gamma)\end{bmatrix}
$$

$$
= \sum_{\mu,\gamma} w_{\mu,\gamma}\frac{1}{2\pi}\int d\phi\, [\cos(\mu'\phi+\gamma')\quad \sin(\mu'\phi+\gamma')]\,\psi_m(r_\theta)\begin{bmatrix}\cos(\mu\phi+\gamma)\\\sin(\mu\phi+\gamma)\end{bmatrix}
$$

$$
= \sum_{\mu,\gamma} w_{\mu,\gamma}\frac{1}{2\pi}\int d\phi\, \cos(\mu'\phi+\gamma'-\mu\phi-\gamma-m\theta)
$$

$$
= \sum_{\mu,\gamma} w_{\mu,\gamma}\frac{1}{2\pi}\int d\phi\, \cos((\mu'-\mu)\phi+\gamma'-\gamma-m\theta)\,.
$$

The integral is non-zero only if the frequency is 0, i.e. $\mu'-\mu=0$:

$$
= \sum_{\gamma} w_{\mu',\gamma}\cos(\gamma'-\gamma-m\theta)
$$

Requiring the projections to be equal implies

$$
\langle b_{\mu',\gamma'}, R_\theta\kappa\rangle = \langle b_{\mu',\gamma'},\psi_m(r_\theta)\kappa(\cdot)\rangle \qquad\qquad \forall\theta\in[0,2\pi)
$$

$$
\Leftrightarrow \qquad \sum_{\gamma} w_{\mu',\gamma}\cos(\gamma'-\gamma-\mu'\theta) = \sum_{\gamma} w_{\mu',\gamma}\cos(\gamma'-\gamma-m\theta) \qquad \forall\theta\in[0,2\pi)
$$

$$
\Leftrightarrow \quad w_{\mu',0}\cos(\gamma'-\mu'\theta) + w_{\mu',\frac{\pi}{2}}\sin(\gamma'-\mu'\theta) = w_{\mu',0}\cos(\gamma'-m\theta) + w_{\mu',\frac{\pi}{2}}\sin(\gamma'-m\theta)
$$

$$
\forall\theta\in[0,2\pi)
$$

$$
\Leftrightarrow \qquad\qquad\qquad \gamma'-\mu'\theta = \gamma'-m\theta+2t\pi \qquad\qquad\qquad \forall\theta\in[0,2\pi)
$$

$$
\Leftrightarrow \qquad\qquad\qquad\qquad \mu'\theta = m\theta+2t\pi \qquad\qquad\qquad\qquad \forall\theta\in[0,2\pi),
$$

where we made use of Eq. (19) once again. It follows that $\mu'=m$, resulting in the two-dimensional basis

$$
\mathcal{K}_{\psi_m\leftarrow\psi_n}^{\mathrm{SO}(2)} = \left\{ b_{m,\gamma}(\phi) = \begin{bmatrix}\cos(m\phi+\gamma)\\\sin(m\phi+\gamma)\end{bmatrix}\ \middle|\ \gamma\in\left\{0,\frac{\pi}{2}\right\}\right\} \tag{27}
$$

of equivariant kernels for $m>0$ and $n=0$. This basis is explicitly given in the lower left cell of Table 8.

## 2 and 1-dimensional irreps:

The case for 2-dimensional input and 1-dimensional output representations, i.e. $\rho_{\mathrm{in}}=\psi_n^{\mathrm{SO}(2)}$ and $\rho_{\mathrm{out}}=\psi_0^{\mathrm{SO}(2)}$, is identical to the previous one up to a transpose. The final two-dimensional basis for $m=0$ and $n>0$ is therefore given by

$$
\mathcal{K}_{\psi_m\leftarrow\psi_n}^{\mathrm{SO}(2)} = \left\{ b_{n,\gamma}(\phi) = [\cos(n\phi+\gamma)\quad \sin(n\phi+\gamma)]\ \middle|\ \gamma\in\left\{0,\frac{\pi}{2}\right\}\right\} \tag{28}
$$

as shown in the upper right cell of Table 8.

### I.4.2 Derivation for the reflection group

The action of the reflection group $(\{\pm 1\}, *)$ on $\mathbb{R}^2$ depends on a choice of reflection axis, which we specify by an angle $\beta$. More precisely, the element $s \in (\{\pm 1\}, *)$ acts on $x = (r, \phi) \in \mathbb{R}^2$ as

$$s.x(r, \phi) := x(r, s.\phi) := x(r, 2\beta\delta_{s,-1} + s\phi) = \begin{cases} x(r, \phi) & \text{if } s = 1 \\ x(r, 2\beta - \phi) & \text{if } s = \text{-1} . \end{cases}$$

The kernel constraint for the reflection group is therefore being made explicit by

$$\begin{aligned} & \kappa(r, s.\phi) = \rho_{\text{out}}(s)\kappa(r, \phi)\rho_{\text{in}}(s)^{-1} && \forall s \in (\{\pm 1\}, *), \phi \in [0, 2\pi) \\ \Leftrightarrow \quad & \kappa(r, \delta_{s,-1}2\beta + s\phi) = \rho_{\text{out}}(s)\kappa(r, \phi)\rho_{\text{in}}(s)^{-1} && \forall s \in (\{\pm 1\}, *), \phi \in [0, 2\pi) \\ \Leftrightarrow \quad & \kappa(r, \delta_{s,-1}2\beta + s\phi) = \rho_{\text{out}}(s)\kappa(r, \phi)\rho_{\text{in}}(s) && \forall s \in (\{\pm 1\}, *), \phi \in [0, 2\pi) , \end{aligned}$$

where we used the identity $s^{-1} = s$. For $s = +1$ the constraint is trivially true. We will thus in the following consider the case $s = -1$, that is,

$$\kappa(r, 2\beta - \phi) = \rho_{\text{out}}(-1)\kappa(r, \phi)\rho_{\text{in}}(-1) \qquad \forall \phi \in [0, 2\pi) .$$

In order to simplify this constraint further we define a transformed kernel $\kappa'(r, \phi) := \kappa(r, \phi + \beta)$ which is oriented relative to the reflection axis. The transformed kernel is then required to satisfy

$$\kappa'(r, \beta - \phi) = \rho_{\text{out}}(-1)\kappa'(r, \phi - \beta)\rho_{\text{in}}(-1) \qquad \forall \phi \in [0, 2\pi) ,$$

which, with the change of variables $\phi' = \phi - \beta$, reduces to the constraint for equivariance under reflections around the x-axis, i.e. the case for $\beta = 0$:

$$\kappa'(r, -\phi') = \rho_{\text{out}}(-1)\kappa'(r, \phi')\rho_{\text{in}}(-1) \qquad \forall \phi' \in [0, 2\pi) .$$

As a consequence we can retrieve kernels equivariant under reflections around the $\beta$-axis through

$$\kappa(r, \phi) := \kappa'(r, \phi - \beta) .$$

We will therefore without loss of generality consider the case $\beta = 0$ only in the following.

**1-dimensional irreps:**

The reflection group $(\{\pm 1\}, *)$ has only two irreps, namely the trivial representation $\psi_0^{(\{\pm 1\},*)}(s) = 1$ and the sign-flip representation $\psi_1^{(\{\pm 1\},*)}(s) = s$. Therefore only the 1-dimensional case with a kernel of form $\kappa : \mathbb{R}^2 \to \mathbb{R}^{1 \times 1}$ exists. Note that we can write the irreps out as $\psi_f^{(\{\pm 1\},*)}(s) = s^f$, in particular $\psi_f^{(\{\pm 1\},*)}(-1) = (-1)^f$.

Consider the output and input irreps $\rho_{\text{out}} = \psi_i^{(\{\pm 1\},*)}$ and $\rho_{\text{in}} = \psi_j^{(\{\pm 1\},*)}$ (with $i, j \in \{0, 1\}$) and the usual 1-dimensional Fourier basis for scalar functions in $L^2(S^1)$ as before:

$$\left\{ b_{\mu,\gamma}(\phi) = \cos(\mu\phi + \gamma) \,\middle|\, \mu \in \mathbb{N}, \gamma \in \begin{cases} \{0\} & \text{if } \mu = 0 \\ \{0, \pi/2\} & \text{otherwise} \end{cases} \right\} \tag{29}$$

Defining the reflection operator $S$ by its action $(S\kappa)(\phi) := \kappa(-\phi)$, we require the projections of both sides of the kernel constraint on the same basis element to be equal as usual. Specifically, for a particular basis $b_{\mu',\gamma'}$:

$$\langle b_{\mu',\gamma'}, S\kappa \rangle = \langle b_{\mu',\gamma'}, \psi_i^{(\{\pm 1\},*)}(-1)\kappa(\cdot)\psi_j^{(\{\pm 1\},*)}(-1) \rangle$$

The *lhs* implies

$$\langle b_{\mu',\gamma'}, S\,\kappa\rangle = \sum_{\mu,\gamma} w_{\mu,\gamma}\frac{1}{2\pi}\int d\phi \; b_{\mu',\gamma'}(\phi)b_{\mu,\gamma}(-\phi)$$

$$= \sum_{\mu,\gamma} w_{\mu,\gamma}\frac{1}{2\pi}\int d\phi \; \cos(\mu'\phi+\gamma')\cos(-\mu\phi+\gamma)$$

$$= \sum_{\mu,\gamma} w_{\mu,\gamma}\frac{1}{2\pi}\int d\phi \; \frac{1}{2}\left(\cos((\mu'+\mu)\phi+(\gamma'-\gamma))+\cos((\mu'-\mu)\phi+(\gamma'+\gamma))\right)$$

$$= \sum_{\gamma} w_{\mu',\gamma}\frac{1}{2}\left(\cos(\gamma'+\gamma)+\delta_{\mu',0}\cos(\gamma'-\gamma)\right)$$

while the *rhs* leads to

$$\langle b_{\mu',\gamma'}, \psi_m^{(\{\pm1\},*)}(-1)\kappa(\cdot)\psi_n^{(\{\pm1\},*)}(-1)\rangle$$

$$= \sum_{\mu,\gamma} w_{\mu,\gamma}\frac{1}{2\pi}\int d\phi \; b_{\mu',\gamma'}(\phi)\psi_i^{(\{\pm1\},*)}(-1)b_{\mu,\gamma}(\phi)\psi_j^{(\{\pm1\},*)}(-1)$$

$$= \sum_{\mu,\gamma} w_{\mu,\gamma}\frac{1}{2\pi}\int d\phi \; \cos(\mu'\phi+\gamma')(-1)^i\cos(\mu\phi+\gamma)(-1)^j$$

$$= (-1)^{i+j}\sum_{\mu,\gamma} w_{\mu,\gamma}\frac{1}{2\pi}\int d\phi \; \cos(\mu'\phi+\gamma')\cos(\mu\phi+\gamma)$$

$$= (-1)^{i+j}\sum_{\gamma} w_{\mu',\gamma}\frac{1}{2}\left(\cos(\gamma'-\gamma)+\delta_{\mu',0}\cos(\gamma'+\gamma)\right)\;.$$

Now, we require both sides to be equal, that is,

$$\sum_{\gamma} w_{\mu',\gamma}\frac{1}{2}\left(\cos(\gamma'+\gamma)+\delta_{\mu',0}\cos(\gamma'\text{ - }\gamma)\right) = (\text{-}1)^{i+j}\sum_{\gamma} w_{\mu',\gamma}\frac{1}{2}\left(\cos(\gamma'\text{ - }\gamma)+\delta_{\mu',0}\cos(\gamma'+\gamma)\right)$$

and again consider two cases for $\mu'$:

• $\mu'=0$  The basis in Eq.(29) is restricted to the single case $\gamma'=0$. Hence:

$$\sum_{\gamma} w_{0,\gamma}\frac{1}{2}\left(\cos(\gamma)+\cos(-\gamma)\right) = (-1)^{i+j}\sum_{\gamma} w_{0,\gamma}\frac{1}{2}\left(\cos(-\gamma)+\cos(\gamma)\right)$$

As $\gamma\in\{0,\frac{\pi}{2}\}$ and $\cos(\pm\frac{\pi}{2})=0$:

$$\Leftrightarrow \qquad w_{0,0}\frac{1}{2}\left(\cos(0)+\cos(0)\right) = (-1)^{i+j}w_{0,0}\frac{1}{2}\left(\cos(-0)+\cos(0)\right)$$

$$\Leftrightarrow \qquad w_{0,0} = (-1)^{i+j}w_{0,0}$$

Which is always true when $i=j$, while it enforces $w_{0,0}=0$ when $i\neq j$.

• $\mu'>0$  In this case we get:

$$\sum_{\gamma} w_{\mu',\gamma}\frac{1}{2}\cos(\gamma'+\gamma) = (-1)^{i+j}\sum_{\gamma} w_{\mu',\gamma}\frac{1}{2}\cos(\gamma'-\gamma)$$

$$\Leftrightarrow \qquad (1-(-1)^{i+j})w_{\mu',0}\cos(\gamma') = (1+(-1)^{i+j})w_{\mu,\frac{\pi}{2}}\sin(\gamma')$$

If $i+j\equiv 0 \mod 2$, the equation becomes $\sin(\gamma')=0$ and, so, $\gamma'=0$. Otherwise, it becomes $\cos(\gamma')=0$, which means $\gamma'=\frac{\pi}{2}$. Shortly, $\gamma'=(i+j \mod 2)\frac{\pi}{2}$.

As a result, only half of the basis for $\beta=0$ is preserved:

$$\mathcal{K}_{\psi_i\leftarrow\psi_j}^{(\{\pm1\},*),\beta=0} = \left\{ b_{\mu,\gamma}(\phi)=\cos(\mu\phi+\gamma) \;\middle|\; \mu\in\mathbb{N},\; \gamma=(i+j \mod 2)\frac{\pi}{2},\; \mu>0\vee\gamma=0\right\}$$

$$\tag{30}$$

The solution for a general reflection axis $\beta$ is therefore given by

$$\mathcal{K}_{\psi_i \leftarrow \psi_j}^{(\{\pm 1\},*),\beta} = \left\{ b_{\mu,\gamma}(\phi) = \cos(\mu(\phi - \beta) + \gamma) \,\middle|\, \mu \in \mathbb{N}, \gamma = (i + j \mod 2)\frac{\pi}{2}, \mu > 0 \vee \gamma = 0 \right\}$$

(31)

which is visualized in Table 10 for the different cases of irreps for $i, j \in \{0, 1\}$.

### I.4.3 Derivation for O(2)

The orthogonal group $\mathrm{O}(2)$ is the semi-direct product between the rotation group $\mathrm{SO}(2)$ and the reflection group $(\{\pm 1\}, *)$, i.e. $\mathrm{O}(2) \cong \mathrm{SO}(2) \rtimes (\{\pm 1\}, *)$. This justifies a decomposition of the constraint on $\mathrm{O}(2)$-equivariant kernels as the union of the constraints for rotations and reflections. Consequently, the space of $\mathrm{O}(2)$-equivariant kernels is the intersection between the spaces of $\mathrm{SO}(2)$- and reflection-equivariant kernels.

**Proof**

*Sufficiency*:

Assume a rotation- and reflection-equivariant kernel, i.e. a kernel which for all $r \in \mathbb{R}_0^+$ and $\phi \in [0, 2\pi)$ satisfies

$$\kappa(r, r_\theta \phi) = \left(\mathrm{Res}_{\mathrm{SO}(2)}^{\mathrm{O}(2)} \rho_{\mathrm{out}}\right)(r_\theta)\, \kappa(r, \phi) \left(\mathrm{Res}_{\mathrm{SO}(2)}^{\mathrm{O}(2)} \rho_{\mathrm{in}}\right)^{-1}(r_\theta) \qquad \forall\, r_\theta \in \mathrm{SO}(2)$$

$$= \rho_{\mathrm{out}}(r_\theta)\, \kappa(r, \phi)\, \rho_{\mathrm{in}}^{-1}(r_\theta)$$

and

$$\kappa(r, s\phi) = \left(\mathrm{Res}_{(\{\pm 1\},*)}^{\mathrm{O}(2)} \rho_{\mathrm{out}}\right)(s)\, \kappa(r, \phi) \left(\mathrm{Res}_{(\{\pm 1\},*)}^{\mathrm{O}(2)} \rho_{\mathrm{in}}\right)^{-1}(s) \qquad \forall\, s \in (\{\pm 1\}, *)$$

$$= \rho_{\mathrm{out}}(s)\, \kappa(r, \phi)\, \rho_{\mathrm{in}}^{-1}(s)\,.$$

Then, for any $g = r_\theta s \in \mathrm{O}(2)$, the kernel constraint becomes:

$$\kappa(r, g\phi) = \rho_{\mathrm{out}}(g)\, \kappa(r, \phi)\, \rho_{\mathrm{in}}^{-1}(g)$$

$$\Leftrightarrow \quad \kappa(r, r_\theta s\phi) = \rho_{\mathrm{out}}(r_\theta s)\, \kappa(r, \phi)\, \rho_{\mathrm{in}}^{-1}(r_\theta s)$$

$$\Leftrightarrow \quad \kappa(r, r_\theta s\phi) = \rho_{\mathrm{out}}(r_\theta)\rho_{\mathrm{out}}(s)\, \kappa(r, \phi)\, \rho_{\mathrm{in}}^{-1}(s)\rho_{\mathrm{in}}^{-1}(r_\theta)\,.$$

Applying reflection-equivariance this equation simplifies to

$$\Leftrightarrow \quad \kappa(r, r_\theta s\phi) = \rho_{\mathrm{out}}(r_\theta)\, \kappa(r, s\phi)\, \rho_{\mathrm{in}}^{-1}(r_\theta)\,,$$

which, applying rotation-equivariance yields

$$\Leftrightarrow \quad \kappa(r, r_\theta s\phi) = \kappa(r, r_\theta s\phi)\,.$$

Hence any kernel satisfying both $\mathrm{SO}(2)$ and reflection constraints is also $\mathrm{O}(2)$ equivariant.

*Necessity*:

Trivially, $\mathrm{O}(2)$ equivariance implies equivariance under $\mathrm{SO}(2)$ and reflections. Specifically, for any $r \in \mathbb{R}_0^+$ and $\phi \in [0, 2\pi)$, the equation

$$\kappa(r, g\phi) = \rho_{\mathrm{out}}(g)\, \kappa(r, \phi)\, \rho_{\mathrm{in}}^{-1}(g) \qquad \forall\, g = r_\theta s \in \mathrm{O}(2)$$

implies

$$\kappa(r, r_\theta \phi) = \rho_{\mathrm{out}}(r_\theta)\, \kappa(r, \phi)\, \rho_{\mathrm{in}}^{-1}(r_\theta)$$

$$= \left(\mathrm{Res}_{\mathrm{SO}(2)}^{\mathrm{O}(2)} \rho_{\mathrm{out}}\right)(r_\theta)\, \kappa(r, \phi) \left(\mathrm{Res}_{\mathrm{SO}(2)}^{\mathrm{O}(2)} \rho_{\mathrm{in}}\right)^{-1}(r_\theta) \qquad \forall\, r_\theta \in \mathrm{SO}(2)$$

and

$$\kappa(r, s\phi) = \rho_{\text{out}}(s)\, \kappa(r, \phi)\, \rho_{\text{in}}^{-1}(s)$$

$$= \left(\text{Res}_{(\{\pm 1\},*)}^{O(2)} \rho_{\text{out}}\right)(s)\, \kappa(r, \phi) \left(\text{Res}_{(\{\pm 1\},*)}^{O(2)} \rho_{\text{in}}\right)^{-1}(s) \quad \forall\, s \in (\{\pm 1\}, *).$$

This observation allows us to derive the kernel space for $O(2)$ by intersecting the previously derived kernel space of $SO(2)$ with the kernel space of the reflection group:

$$\mathcal{K}_{\rho_{\text{out}} \leftarrow \rho_{\text{in}}}^{O(2)} = \left\{\kappa \mid \kappa(r, g.\phi) = \rho_{\text{out}}(g)\kappa(r, \phi)\rho_{\text{in}}(g)^{-1} \ \forall\, g \in O(2)\right\}$$

$$= \left\{\kappa \mid \kappa(r, r_\theta.\phi) = \rho_{\text{out}}(r_\theta)\kappa(r, \phi)\rho_{\text{in}}(r_\theta)^{-1} \ \forall\, r_\theta \in SO(2)\right\}$$

$$\cap \left\{\kappa \mid \kappa(r, \ s.\phi) = \ \rho_{\text{out}}(s)\kappa(r, \phi)\rho_{\text{in}}(s)^{-1} \ \ \forall\, s \in (\{\pm 1\}, *)\right\}$$

As $O(2)$ contains all rotations, it does also contain all reflection axes. Without loss of generality, we define $s \in O(2)$ as the reflection along the $x$-axis. A reflection along any other axis $\beta$ is associated with the group element $r_{2\beta}s \in O(2)$, i.e. the combination of a reflection with a rotation of $2\beta$. As a result, we consider the basis for reflection equivariant kernels derived for $\beta = 0$ in Eq. (30).

Therefore, to derive a basis associated to a pair of input and output representations $\rho_{\text{in}}$ and $\rho_{\text{out}}$, we restrict the representations to $SO(2)$ and the reflection group, compute the two bases using the results found in Appendix I.4.1 and in Appendix I.4.2, and, finally, take their intersection.

**2-dimensional irreps:**

The restriction of any 2-dimensional irrep $\psi_{1,n}^{O(2)}$ of $O(2)$ to the reflection group decomposes into the direct sum of the two 1-dimensional irreps of the reflection group, i.e. into the diagonal matrix

$$\text{Res}_{(\{\pm 1\},*)}^{O(2)} \psi_{1,n}(s) = \left(\psi_0^{(\{\pm 1\},*)} \oplus \psi_1^{(\{\pm 1\},*)}\right)(s) = \begin{bmatrix} \psi_0^{(\{\pm 1\},*)}(s) & 0 \\ 0 & \psi_1^{(\{\pm 1\},*)}(s) \end{bmatrix} = \begin{bmatrix} 1 & 0 \\ 0 & s \end{bmatrix}.$$

It follows that the restricted kernel space constraint decomposes into independent constraints on each entry of the original kernel. Specifically, for output and input representations $\rho_{\text{out}} = \psi_{1,m}^{O(2)}$ and $\rho_{\text{in}} = \psi_{1,n}^{O(2)}$, the constraint becomes

$$\kappa(s.x) = \underbrace{\left(\begin{array}{c|c} \psi_0^{(\{\pm 1\},*)}(s) & \\ \hline & \psi_1^{(\{\pm 1\},*)}(s) \end{array}\right)}_{\text{Res}_{(\{\pm 1\},*)}^{O(2)} \rho_{\text{out}}(s)} \cdot \underbrace{\left(\begin{array}{c|c} \kappa^{00} & \kappa^{01} \\ \hline \kappa^{10} & \kappa^{11} \end{array}\right)}_{\kappa(x)} \cdot \underbrace{\left(\begin{array}{c|c} \psi_0^{(\{\pm 1\},*)}(s)^{-1} & \\ \hline & \psi_1^{(\{\pm 1\},*)}(s)^{-1} \end{array}\right)}_{\text{Res}_{(\{\pm 1\},*)}^{O(2)} \rho_{\text{in}}(s)}$$

We can therefore solve for a basis for each entry individually following Appendix I.4.2 to obtain the complete basis

$$\left\{b_{\mu,0}^{00}\ (\phi) = \begin{bmatrix} \cos(\mu\phi) & 0 \\ 0 & 0 \end{bmatrix} \middle| \mu \in \mathbb{N}\ \right\} \cup \left\{b_{\mu,\frac{\pi}{2}}^{01}\ (\phi) = \begin{bmatrix} 0 & \sin(\mu\phi) \\ 0 & 0 \end{bmatrix} \middle| \mu \in \mathbb{N}^+\right\} \cup$$

$$\left\{b_{\mu,\frac{\pi}{2}}^{10}\ (\phi) = \begin{bmatrix} 0 & 0 \\ \sin(\mu\phi) & 0 \end{bmatrix} \middle| \mu \in \mathbb{N}^+\right\} \cup \left\{b_{\mu,0}^{11}\ (\phi) = \begin{bmatrix} 0 & 0 \\ 0 & \cos(\mu\phi) \end{bmatrix} \middle| \mu \in \mathbb{N}\ \right\}.$$

Through the same change of basis applied in the first paragraph of Appendix I.4.1, we get the following equivalent basis for the same space:

$$\left\{b_{\mu,s}(\phi) = \begin{bmatrix} \cos(\mu\phi) & -\sin(\mu\phi) \\ \sin(\mu\phi) & \cos(\mu\phi) \end{bmatrix} \begin{bmatrix} 1 & 0 \\ 0 & s \end{bmatrix}\right\}_{\mu \in \mathbb{Z}, s \in \{\pm 1\}}$$

$$= \left\{b_{\mu,s}(\phi) = \psi(\mu\phi)\xi(s)\right\}_{\mu \in \mathbb{Z}, s \in \{\pm 1\}}. \tag{32}$$

On the other hand, 2-dimensional $O(2)$ representations restrict to the $SO(2)$ irreps of the corresponding frequency, i.e.

$$\text{Res}_{SO(2)}^{O(2)} \rho_{\text{in}} = \text{Res}_{SO(2)}^{O(2)} \psi_{1,n}^{O(2)}(r_\theta) = \psi_n^{SO(2)}(r_\theta)$$

and

$$\text{Res}^{\text{O}(2)}_{\text{SO}(2)} \rho_{\text{out}} = \text{Res}^{\text{O}(2)}_{\text{SO}(2)} \psi^{\text{O}(2)}_{1,m}(r_\theta) = \psi^{\text{SO}(2)}_m(r_\theta).$$

In Appendix I.4.1, a basis for $\text{SO}(2)$-equivariant kernels with respect to a $\psi^{\text{SO}(2)}_n$ input field and $\psi^{\text{SO}(2)}_m$ output field was derived starting from the basis in Eq. (20). Notice that the basis of reflection-equivariant kernels in Eq. (32) contains exactly half of the elements in Eq. (20), indexed by $\gamma = 0$. A basis for $\text{O}(2)$-equivariant kernels can be found by repeating the derivations in Appendix I.4.1 for $\text{SO}(2)$-equivariant kernels using only the subset in Eq. (32) of reflection-equivariant kernels. The resulting two-dimensional $\text{O}(2)$-equivariant basis, which includes the $\text{SO}(2)$-equivariance conditions ($\mu = m - sn$) and the reflection-equivariance conditions ($\gamma = 0$), is given by

$$\mathcal{K}^{\text{O}(2)}_{\psi_{i,m} \leftarrow \psi_{j,n}} = \left\{ b_{\mu,0,s}(\phi) = \psi(\mu\phi)\xi(s) \ \middle| \ \mu = m - sn, s \in \{\pm 1\} \right\}, \tag{33}$$

where $i = j = 1$ and $m, n > 0$. See the bottom right cell in Table 9.

**1-dimensional irreps:**

$\text{O}(2)$ has two 1-dimensional irreps $\psi^{\text{O}(2)}_{0,0}$ and $\psi^{\text{O}(2)}_{1,0}$ (see Appendix I.2). Both are trivial under rotations and each of them corresponds to one of the two reflection group's irreps, i.e.

$$\text{Res}^{\text{O}(2)}_{(\{\pm 1\},*)} \psi^{\text{O}(2)}_{i,0}(s) = \psi^{(\{\pm 1\},*)}_i(s) = s^i$$

and

$$\text{Res}^{\text{O}(2)}_{\text{SO}(2)} \psi^{\text{O}(2)}_{i,0}(r_\theta) = \psi^{\text{SO}(2)}_0(r_\theta) = 1.$$

Considering output and input representations $\rho_{\text{out}} = \psi^{\text{O}(2)}_{i,0}$ and $\rho_{\text{in}} = \psi^{\text{O}(2)}_{j,0}$, it follows that:

$$\text{Res}^{\text{O}(2)}_{(\{\pm 1\},*)} \rho_{\text{in}} = \text{Res}^{\text{O}(2)}_{(\{\pm 1\},*)} \psi^{\text{O}(2)}_{j,0} = \psi^{(\{\pm 1\},*)}_j$$
$$\text{Res}^{\text{O}(2)}_{(\{\pm 1\},*)} \rho_{\text{out}} = \text{Res}^{\text{O}(2)}_{(\{\pm 1\},*)} \psi^{\text{O}(2)}_{i,0} = \psi^{(\{\pm 1\},*)}_i$$
$$\text{Res}^{\text{O}(2)}_{\text{SO}(2)} \rho_{\text{in}} = \text{Res}^{\text{O}(2)}_{\text{SO}(2)} \psi^{\text{O}(2)}_{j,0} = \psi^{\text{SO}(2)}_0$$
$$\text{Res}^{\text{O}(2)}_{\text{SO}(2)} \rho_{\text{out}} = \text{Res}^{\text{O}(2)}_{\text{SO}(2)} \psi^{\text{O}(2)}_{i,0} = \psi^{\text{SO}(2)}_0$$

In order to solve the $\text{O}(2)$ kernel constraint consider again the reflectional constraint and the $\text{SO}(2)$ constraint. Bases for reflection-equivariant kernels with above representations were derived in Appendix I.4.2 and are shown in Eq. (30). These bases form a subset of the Fourier basis in Eq. (24) which is being indexed by $\gamma = (i + j \mod 2)\frac{\pi}{2}$. On the other hand, the full Fourier basis was restricted by the $\text{SO}(2)$ constraint to satisfy $\mu = 0$ and therefore $\gamma = 0$, see Eq. (25). Intersecting both constraints therefore implies $i = j$, resulting in the $\text{O}(2)$-equivariant basis

$$\mathcal{K}^{\text{O}(2)}_{\psi_{i,m} \leftarrow \psi_{j,n}} = \begin{cases} \{b_{0,0}(\phi) = 1\} & \text{if } i = j, \\ \emptyset & \text{else} \end{cases} \tag{34}$$

for $m, n = 0$ which is shown in the top left cell in Table 9.

**1 and 2-dimensional irreps:**

Now we consider the 2-dimensional output representation $\rho_{\text{out}} = \psi^{\text{O}(2)}_{1,m}$ and the 1-dimensional input representation $\rho_{\text{in}} = \psi^{\text{O}(2)}_{j,0}$.

Following the same strategy as before we find the reflectional constraints for these representations to be given by

$$\kappa(s.x) = \underbrace{\left( \frac{\psi_0^{(\{\pm 1\},*)}(s)}{\psi_1^{(\{\pm 1\},*)}(s)} \right)}_{\mathrm{Res}_{(\{\pm 1\},*)}^{O(2)} \rho_{\mathrm{out}}(s)} \cdot \underbrace{\left( \frac{\kappa^{00}}{\kappa^{10}} \right)}_{\kappa(x)} \cdot \underbrace{\left( \psi_j^{(\{\pm 1\},*)}(s)^{-1} \right)}_{\mathrm{Res}_{(\{\pm 1\},*)}^{O(2)} \rho_{\mathrm{in}}(s)},$$

and therefore to decompose into two independent constraints on the entries $\kappa^{00}$ and $\kappa^{10}$. Solving for a basis for each entry and taking their union as before we get[15]

$$\left\{ b_\mu^{00}(\phi) = \begin{bmatrix} \cos(\mu\phi + j\frac{\pi}{2}) \\ 0 \end{bmatrix} \right\}_{\mu \in \mathbb{N}} \cup \left\{ b_\mu^{10}(\phi) = \begin{bmatrix} 0 \\ \sin(\mu\phi - j\frac{\pi}{2}) \end{bmatrix} \right\}_{\mu \in \mathbb{N}},$$

which, through a change of basis, can be rewritten as

$$\left\{ b_{\mu, j\frac{\pi}{2}}(\phi) = \begin{bmatrix} \cos(\mu\phi + j\frac{\pi}{2}) \\ \sin(\mu\phi + j\frac{\pi}{2}) \end{bmatrix} \right\}_{\mu \in \mathbb{Z}}. \tag{35}$$

We intersect this basis with the basis of $SO(2)$ equivariant kernels with respect to a $\mathrm{Res}_{SO(2)}^{O(2)} \rho_{\mathrm{in}} = \psi_0^{SO(2)}$ input field and $\mathrm{Res}_{SO(2)}^{O(2)} \rho_{\mathrm{out}} = \psi_m^{SO(2)}$ output field as derived in Appendix I.4.1. Both constraints, that is, $\gamma = j\frac{\pi}{2}$ for the reflection group and $\mu = m$ for $SO(2)$ (see Eq. (26)), define the one-dimensional basis for $O(2)$-equivariant kernels for $n = 0$, $m > 0$ and $i = 1$ as

$$\boxed{\mathcal{K}_{\psi_{i,m} \leftarrow \psi_{j,n}}^{O(2)} = \left\{ b_{\mu, j\frac{\pi}{2}}(\phi) = \begin{bmatrix} \cos(\mu\phi + j\frac{\pi}{2}) \\ \sin(\mu\phi + j\frac{\pi}{2}) \end{bmatrix} \;\middle|\; \mu = m \right\},} \tag{36}$$

see the bottom left cell in Table 9.

**2 and 1-dimensional irreps:**

As already argued in the case for $SO(2)$, the basis for 2-dimensional input representations $\rho_{\mathrm{in}} = \psi_{1,n}^{O(2)}$ and 1-dimensional output representations $\rho_{\mathrm{out}} = \psi_{i,0}^{O(2)}$ is identical to the previous basis up to a transpose, i.e. it is given by

$$\boxed{\mathcal{K}_{\psi_{i,m} \leftarrow \psi_{j,n}}^{O(2)} = \left\{ b_{\mu, i\frac{\pi}{2}}(\phi) = \begin{bmatrix} \cos(\mu\phi + i\frac{\pi}{2}) & \sin(\mu\phi + i\frac{\pi}{2}) \end{bmatrix} \;\middle|\; \mu = n \right\},} \tag{37}$$

where $j = 1$, $n > 0$ and $m = 0$. This case is visualized in the top right cell of Table 9.

### I.4.4   Derivation for $C_N$

The derivations for $C_N$ coincide mostly with the derivations done for $SO(2)$ with the difference that the projected constraints need to hold for discrete angles $\theta \in \left\{ p\frac{2\pi}{N} \,|\, p = 0, \ldots, N - 1 \right\}$ only. Furthermore, $C_N$ has one additional 1-dimensional irrep of frequency $N/2$ if (and only if) $N$ is even.

**2-dimensional irreps:**

During the derivation of the solutions for $SO(2)$'s 2-dimensional irreps in Appendix I.4.1, we assumed

continuous angles only in the very last step. The constraint in Eq. (22) therefore holds for $C_N$ as well. Specifically, it demands that for each $\theta \in \{p\frac{2\pi}{N} \,|\, p = 0, \dots, N-1\}$ there exists a $t \in \mathbb{Z}$ such that:

$$(\mu' - (m - ns'))\theta = 2t\pi$$
$$\Leftrightarrow \qquad (\mu' - (m - ns'))p\frac{2\pi}{N} = 2t\pi$$
$$\Leftrightarrow \qquad (\mu' - (m - ns'))p = tN$$

The last result corresponds to a system of $N$ linear congruence equations modulo $N$ which require $N$ to divide $(\mu' - (m - ns'))p$ for each non-negative integer $p$ smaller than $N$. Note that solutions of the constraint for $p = 1$ already satisfy the constraints for $p \in 2, \dots, N-1$ such that it is sufficient to consider

$$(\mu' - (m - ns'))1 = tN$$
$$\Leftrightarrow \qquad \mu' = m - ns' + tN\,.$$

The resulting basis

$$\mathcal{K}^{C_N}_{\psi_m \leftarrow \psi_n} = \left\{ b_{\mu,\gamma,s}(\phi) = \psi(\mu\phi + \gamma)\xi(s) \,\middle|\, \mu = m - sn + tN, \gamma \in \left\{0, \frac{\pi}{2}\right\} s \in \{\pm 1\} \right\}_{t \in \mathbb{Z}} \qquad (38)$$

for $m, n > 0$ thus coincides mostly with the basis 23 for $SO(2)$ but contains solutions for aliased frequencies, defined by adding $tN$. The bottom right cell in Table 11 gives the explicit form of this basis.

**1-dimensional irreps:**

The same trick could be applied to solve the remaining three cases. However, since $C_N$ has an additional one dimensional irrep of frequency $N/2$ for even $N$ it is convenient to rederive all cases. We therefore consider $\rho_{\text{out}} = \psi_m^{C_N}$ and $\rho_{\text{in}} = \psi_n^{C_N}$, where $m, n \in \{0, N/2\}$. Note that $\psi_m^{C_N}(\theta), \psi_n^{C_N}(\theta) \in \{\pm 1\}$ for $\theta \in \{p\frac{2\pi}{N} \,|\, p = 0, \dots, N-1\}$.

We use the same Fourier basis

$$\left\{ b_{\mu,\gamma}(\phi) = \cos(\mu\phi + \gamma) \,\middle|\, \mu \in \mathbb{N},\ \gamma \in \begin{cases} \{0\} & \text{if } \mu = 0 \\ \{0, \pi/2\} & \text{otherwise} \end{cases} \right\} \qquad (39)$$

and the same projection operators as used for $SO(2)$.

Since the *lhs* of the kernel constraint does not depend on the representations considered its projection $\langle b_{\mu',\gamma'}, R_\theta \kappa \rangle$ is the same found for $SO(2)$:

$$\langle b_{\mu',\gamma'}, R_\theta \kappa \rangle = \frac{1}{2} \sum_\gamma w_{\mu',\gamma} \left( \cos((\gamma' - \gamma) - \mu'\theta) + \delta_{\mu',0}\cos(\gamma' + \gamma) \right)$$

For the *rhs* we find

$$\langle b_{\mu',\gamma'}, \psi_m^{C_N}(r_\theta)\kappa\, \psi_n^{C_N}(r_\theta)^{-1} \rangle$$
$$= \frac{1}{2\pi} \int d\phi\, b_{\mu',\gamma'}(\phi)\psi_m^{C_N}(r_\theta)\kappa(\phi)\psi_n^{C_N}(r_\theta)^{-1}\,,$$

which by expanding the kernel in the linear combination of the basis and writing the respresentations out yields:

$$= \sum_{\mu,\gamma} w_{\mu,\gamma}\frac{1}{2\pi} \int d\phi\, b_{\mu',\gamma'}(\phi)\cos(m\theta)b_{\mu,\gamma}(\phi)\cos(n\theta)^{-1}$$
$$= \sum_{\mu,\gamma} w_{\mu,\gamma}\frac{1}{2\pi} \int d\phi\, \cos(\mu'\phi + \gamma')\cos(m\theta)\cos(\mu\phi + \gamma)\cos(n\theta)^{-1}$$

Since $\cos(n\theta) \in \{\pm 1\}$ the inverses can be dropped and terms can be collected via trigonometric identities:

$$= \sum_{\mu,\gamma} w_{\mu,\gamma} \frac{1}{2\pi} \int d\phi \, \cos(\mu'\phi + \gamma') \cos(m\theta) \cos(\mu\phi + \gamma) \cos(n\theta)$$

$$= \sum_{\mu,\gamma} w_{\mu,\gamma} \cos(m\theta) \cos(n\theta) \frac{1}{2\pi} \int d\phi \, \cos(\mu'\phi + \gamma') \cos(\mu\phi + \gamma)$$

$$= \sum_{\mu,\gamma} w_{\mu,\gamma} \cos((\pm m \pm n)\theta) \frac{1}{4\pi} \int d\phi \Big( \cos((\mu' - \mu)\phi + \gamma' - \gamma) + \cos((\mu' + \mu)\phi + \gamma' + \gamma) \Big)$$

$$= \sum_{\mu,\gamma} w_{\mu,\gamma} \cos((\pm m \pm n)\theta) \frac{1}{2} \left( \delta_{\mu,\mu'} \cos(\gamma' - \gamma) + \delta_{\mu+\mu',0} \cos(\gamma' + \gamma) \right)$$

$$= \frac{1}{2} \sum_{\mu',\gamma} w_{\mu',\gamma} \cos((\pm m \pm n)\theta) \left( \cos(\gamma' - \gamma) + \delta_{\mu',0} \cos(\gamma' + \gamma) \right)$$

We require the projections to be equal for each $\theta = p\frac{2\pi}{N}$ with $p \in \{0, \ldots, N-1\}$:

$$\langle b_{\mu',\gamma'}, R_\theta \kappa \rangle = \left\langle b_{\mu',\gamma'}, \psi_m^{C_N}(r_\theta)\kappa\, \psi_n^{C_N}(r_\theta)^{-1} \right\rangle$$

$$\Leftrightarrow \quad \sum_\gamma w_{\mu',\gamma} \left( \cos((\gamma' - \gamma) - \mu'\theta) + \delta_{\mu',0} \cos(\gamma' + \gamma) \right) =$$

$$= \sum_{\mu',\gamma} w_{\mu',\gamma} \cos((\pm m \pm n)\theta) \Big( \cos(\gamma' - \gamma) + \delta_{\mu',0} \cos(\gamma' + \gamma) \Big)$$

Again, we consider two cases for $\mu'$:

- $\mu' = 0$ : The basis in Eq.(39) is restricted to the single case $\gamma' = 0$.

$$\sum_\gamma w_{0,\gamma}(\cos(-\gamma) + \cos(\gamma)) = \cos((\pm m \pm n)\theta) \sum_\gamma w_{0,\gamma} \Big( \cos(-\gamma) + \cos(\gamma) \Big)$$

$$\Leftrightarrow \quad w_{0,0} 2\cos(0) + w_{0,\frac{\pi}{2}} 0 = \cos((\pm m \pm n)\theta) \left( w_{0,0} 2\cos(0) + w_{0,\frac{\pi}{2}} 0 \right)$$

$$\Leftrightarrow \quad w_{0,0} = \cos((\pm m \pm n)\theta) w_{0,0}$$

If $\cos((\pm m \pm n)\theta) \neq 1$, the coefficient $w_{0,0}$ is forced to 0. Conversely:

$$\cos((\pm m \pm n)\theta) = 1$$

$$\Leftrightarrow \quad \exists t \in \mathbb{Z} \text{ s.t.} \quad (\pm m \pm n)\theta = 2t\pi$$

Using $\theta = p\frac{2\pi}{N}$:

$$\Leftrightarrow \quad \exists t \in \mathbb{Z} \text{ s.t.} \quad (\pm m \pm n)p\frac{2\pi}{N} = 2t\pi$$

$$\Leftrightarrow \quad \exists t \in \mathbb{Z} \text{ s.t.} \quad (\pm m \pm n)p = tN$$

- $\mu' > 0$ :

$$\sum_\gamma w_{\mu',\gamma} \cos(\gamma' - \gamma - \mu'\theta) = \cos((\pm m \pm n)\theta) \sum_\gamma w_{\mu',\gamma} \cos(\gamma' - \gamma)$$

$$\Leftrightarrow \quad w_{\mu',0} \cos(\gamma' - \mu'\theta) + w_{\mu',\frac{\pi}{2}} \sin(\gamma' - \mu'\theta) =$$

$$\cos((\pm m \pm n)\theta) \left( w_{\mu',0} \cos(\gamma') + w_{\mu',\frac{\pi}{2}} \sin(\gamma') \right)$$

Since $(\pm m \pm n)\theta \in \{-\pi, 0, \pi\}$ we have $\cos((\pm m \pm n)\theta) = \pm 1$, therefore:

$$\Leftrightarrow \quad w_{\mu',0} \cos(\gamma' - \mu'\theta) + w_{\mu',\frac{\pi}{2}} \sin(\gamma' - \mu'\theta) =$$

$$= w_{\mu',0} \cos(\gamma' + (\pm m \pm n)\theta) + w_{\mu',\frac{\pi}{2}} \sin(\gamma' + (\pm m \pm n)\theta)$$

Using the property in Eq. (19):

$$\Leftrightarrow \quad \exists t \in \mathbb{Z} \text{ s.t.} \qquad \gamma' - \mu'\theta = \gamma' + (\pm m \pm n)\theta + 2t\pi$$

$$\Leftrightarrow \quad \exists t \in \mathbb{Z} \text{ s.t.} \qquad \mu'\theta = (\pm m \pm n)\theta + 2t\pi$$

Using $\theta = p\frac{2\pi}{N}$:

$$\Leftrightarrow \quad \exists t \in \mathbb{Z} \text{ s.t.} \qquad \mu'p\frac{2\pi}{N} = (\pm m \pm n)p\frac{2\pi}{N} + 2t\pi$$

$$\Leftrightarrow \quad \exists t \in \mathbb{Z} \text{ s.t.} \qquad \mu'p = (\pm m \pm n)p + tN$$

$$\Leftrightarrow \quad \exists t \in \mathbb{Z} \text{ s.t.} \qquad (\pm m \pm n + \mu')p = tN$$

In both cases $\mu' = 0$ and $\mu' > 0$ we thus find the constraints

$$\forall p \in \{0, 1, \ldots, N-1\} \ \exists t \in \mathbb{Z} \text{ s.t.} \quad (\pm m \pm n + \mu')p = tN \,.$$

It is again sufficient to consider the constraint for $p = 1$ which results in solutions with frequencies $\mu' = \pm m \pm n + tN$. As $(\pm m \pm n) \in \{0, \pm\frac{N}{2}, \pm N\}$, all valid solutions are captured by $\mu' = (m+n \mod N) + tN$, resulting in the basis

$$\mathcal{K}^{C_N}_{\psi_m \leftarrow \psi_n} = \left\{ b_{\mu,\gamma}(\phi) = \cos(\mu\phi + \gamma) \,\middle|\, \mu = (m+n \mod N) + tN, \gamma \in \left\{0, \frac{\pi}{2}\right\}, \mu \neq 0 \vee \gamma = 0 \right\}_{t \in \mathbb{N}}$$

$$(40)$$

for $n, m \in \left\{0, \frac{N}{2}\right\}$. See the top left cells in Table 11.

**1 and 2-dimensional irreps** Next consider a 1-dimensional irrep $\rho_{\text{in}} = \psi_n^{C_N}$ with $n \in \{0, \frac{N}{2}\}$ in the input and a 2-dimensional irrep $\rho_{\text{out}} = \psi_m^{C_N}$ in the output. We derive the solutions by projecting the kernel constraint on the basis introduced in Eq. (26).

For the *lhs* the projection coincides with the result found for SO(2) as before:

$$\langle b_{\mu',\gamma'}, R_\theta \kappa \rangle = \sum_\gamma w_{\mu',\gamma} \cos((\gamma' - \gamma) - \mu'\theta)$$

An expansion and projection of *rhs* gives:

$$\langle b_{\mu',\gamma'}, \psi_m^{C_N}(r_\theta)\kappa(\cdot)\psi_n^{C_N}(r_\theta)^{-1} \rangle$$

$$= \frac{1}{2\pi} \int d\phi \, b_{\mu',\gamma'}(\phi)^T \psi_m^{C_N}(r_\theta) \kappa(\phi) \psi_n^{C_N}(r_\theta)^{-1}$$

$$= \sum_{\mu,\gamma} w_{\mu,\gamma} \frac{1}{2\pi} \int d\phi \, b_{\mu',\gamma'}(\phi)^T \psi_m^{C_N}(r_\theta) \begin{bmatrix} \cos(\mu\phi + \gamma) \\ \sin(\mu\phi + \gamma) \end{bmatrix} \psi_n^{C_N}(r_\theta)^{-1}$$

$$= \sum_{\mu,\gamma} w_{\mu,\gamma} \frac{1}{2\pi} \int d\phi \, [\cos(\mu'\phi + \gamma') \quad \sin(\mu'\phi + \gamma')] \, \psi_m^{C_N}(r_\theta) \begin{bmatrix} \cos(\mu\phi + \gamma) \\ \sin(\mu\phi + \gamma) \end{bmatrix} \psi_n^{C_N}(r_\theta)^{-1}$$

$$= \sum_{\mu,\gamma} w_{\mu,\gamma} \left( \frac{1}{2\pi} \int d\phi \, \cos(\mu'\phi + \gamma' - \mu\phi - \gamma - m\theta) \right) \psi_n^{C_N}(r_\theta)^{-1} \,.$$

The integral is non-zero only if the frequency is 0, i.e. iff $\mu' = \mu$:

$$= \sum_\gamma w_{\mu',\gamma} \cos(\gamma' - \gamma - m\theta) \psi_n^{C_N}(r_\theta)^{-1}$$

$$= \sum_\gamma w_{\mu',\gamma} \cos(\gamma' - \gamma - m\theta) \cos(\pm n\theta)$$

Since $\pm n\theta = p\pi$ for some $p \in \mathbb{N}$ one has $\sin(\pm n\theta) = 0$ which allows to add the following zero summand and simplify:

$$= \sum_{\gamma} w_{\mu',\gamma} \left( \cos(\gamma' - \gamma - m\theta)\cos(\pm n\theta) - \sin(\gamma' - \gamma - m\theta)\sin(\pm n\theta) \right)$$

$$= \sum_{\gamma} w_{\mu',\gamma} \cos(\gamma' - \gamma - (m \pm n)\theta)$$

Requiring the projections to be equal then yields:

$$\langle b_{\mu',\gamma'}, R_\theta \kappa \rangle = \langle b_{\mu',\gamma'}, \psi_m^{C_N}(r_\theta)\kappa(\cdot)\psi_n^{C_N}(r_\theta)^{-1} \rangle \qquad \forall \theta \in \left\{ p\frac{2\pi}{N} \right\}$$

$$\Leftrightarrow \qquad \sum_{\gamma} w_{\mu',\gamma} \cos(\gamma' - \gamma - \mu'\theta) = \sum_{\gamma} w_{\mu',\gamma} \cos(\gamma' - \gamma - (m \pm n)\theta) \quad \forall \theta \in \left\{ p\frac{2\pi}{N} \right\}$$

$$\Leftrightarrow \qquad w_{\mu',0}\cos(\gamma' - \mu'\theta) + w_{\mu',\frac{\pi}{2}}\sin(\gamma' - \mu'\theta) = w_{\mu',0}\cos(\gamma' - (m \pm n)\theta) + w_{\mu',\frac{\pi}{2}}\sin(\gamma' - (m \pm n)\theta)$$

$$\forall \theta \in \left\{ p\frac{2\pi}{N} \right\}$$

Using the property in Eq. (19), this requires that for each $\theta$ there exists a $t \in \mathbb{Z}$ such that:

$$\Leftrightarrow \qquad \gamma' - \mu'\theta = \gamma' - (m \pm n)\theta + 2t\pi \qquad \forall \theta \in \left\{ p\frac{2\pi}{N} \right\}$$

$$\Leftrightarrow \qquad \mu'\theta = (m \pm n)\theta + 2t\pi \qquad \forall \theta \in \left\{ p\frac{2\pi}{N} \right\}$$

Since $\theta = p\frac{2\pi}{N}$ with $p \in \{0, \ldots, N-1\}$ we find that

$$\Leftrightarrow \qquad \mu'p\frac{2\pi}{N} = (m \pm n)p\frac{2\pi}{N} + 2t\pi \qquad \forall p \in \{0, \ldots, N\text{-}1\}$$

$$\Leftrightarrow \qquad \mu'p = (m \pm n)p + tN \qquad \forall p \in \{0, \ldots, N\text{-}1\}$$

$$\Leftrightarrow \qquad \mu' = (m \pm n) + tN$$

$$\Leftrightarrow \qquad \mu' - (m \pm n) = tN \, ,$$

which implies that $N$ needs to divide $\mu' - (m \pm n)$. It follows that the condition holds also for any other $p$. This gives the basis

$$\mathcal{K}_{\psi_m \leftarrow \psi_n}^{C_N} = \left\{ b_{\mu,\gamma}(\phi) = \begin{bmatrix} \cos(\mu\phi + \gamma) \\ \sin(\mu\phi + \gamma) \end{bmatrix} \, \middle| \, \mu = (m \pm n) + tN, \, \gamma \in \left\{ 0, \frac{\pi}{2} \right\} \right\}_{t \in \mathbb{Z}} \qquad (41)$$

for $m > 0$ and $n \in \{0, \frac{N}{2}\}$; see the bottom left cells in Table 11.

**2 and 1-dimensional irreps:**

The basis for 2-dimensional input and 1-dimensional output representations, i.e. $\rho_{\text{in}} = \psi_n^{C_N}$ and $\rho_{\text{out}} = \psi_m^{C_N}$ with $n > 0$ and $m \in \{0, \frac{N}{2}\}$, is identical to the previous one up to a transpose:

$$\mathcal{K}_{\psi_m \leftarrow \psi_n}^{C_N} = \left\{ b_{\mu,\gamma}(\phi) = [\cos(\mu\phi + \gamma) \ \sin(\mu\phi + \gamma)] \, \middle| \, \mu = (\pm m + n) + tN, \, \gamma \in \left\{ 0, \frac{\pi}{2} \right\} \right\}_{t \in \mathbb{Z}}$$

$$(42)$$

for $n > 0$ and $m \in \{0, \frac{N}{2}\}$. See the top right cells in Table 11.

### I.4.5 Derivation for D$_N$

A solution for D$_N$ can easily be derived by repeating the process done for $O(2)$ in Appendix I.4.3 but starting from the bases derived for C$_N$ in Appendix I.4.4 instead of those for $SO(2)$.

In contrast to the case of $O(2)$-equivariant kernels, the choice of reflection axis $\beta$ is not irrelevant since D$_N$ does not act transitively on axes. More precisely, the action of D$_N$ defines equivalence classes $\beta \cong \beta' \Leftrightarrow \exists\, 0 \leq n < N\ :\ \beta = \beta' + n\frac{2\pi}{N}$ of axes which can be labeled by representatives $\beta \in [0, \frac{2\pi}{N})$. For the same argument considered in Appendix I.4.2 we can without loss of generality consider reflections along the axis $\beta = 0$ in our derivations and retrieve kernels $\kappa'$, equivariant to reflections along a general axis $\beta$, as $\kappa'(r, \phi) = \kappa(r, \phi - \beta)$.

**2-dimensional irreps:**

For 2-dimensional input and output representations $\rho_{\text{in}} = \psi_{1,n}^{\text{D}_N}$ and $\rho_{\text{out}} = \psi_{1,m}^{\text{D}_N}$, the final basis is

$$\mathcal{K}_{\psi_{i,m} \leftarrow \psi_{j,n}}^{\text{D}_N} = \left\{ b_{\mu,0,s}(\phi) = \psi(\mu\phi)\xi(s) \,\middle|\, \mu = m - sn + tN, s \in \{\pm 1\} \right\}_{t \in \mathbb{Z}} \tag{43}$$

where $i = j = 1$ and $m, n > 0$. These solutions are written out explicitly in the bottom right of Table 12.

**1-dimensional irreps:**

D$_N$ has 1-dimensional representations $\rho_{\text{in}} = \psi_{j,n}^{\text{D}_N}$ and $\rho_{\text{out}} = \psi_{i,m}^{\text{D}_N}$ for $m, n \in \{0, \frac{N}{2}\}$. In these cases we find the bases

$$\mathcal{K}_{\psi_{i,m} \leftarrow \psi_{j,n}}^{\text{D}_N} = \left\{ b_{\mu,\gamma}(\phi) = \cos(\mu\phi + \gamma) \,\middle|\, \mu = (m+n \mod N) + tN, \right.$$
$$\left. \gamma = (i+j \mod 2)\frac{\pi}{2},\ \mu \neq 0 \vee \gamma = 0 \right\}_{t \in \mathbb{N}} \tag{44}$$

which are shown in the top left cells of Table 12.

**1 and 2-dimensional irreps:**

For 1-dimensional input and 2-dimensional output representations, that is, $\rho_{\text{in}} = \psi_{j,n}^{\text{D}_N}$ and $\rho_{\text{out}} = \psi_{1,m}^{\text{D}_N}$ with $i = 1$, $m > 0$ and $n \in \{0, \frac{N}{2}\}$, the kernel basis is given by:

$$\mathcal{K}_{\psi_{i,m} \leftarrow \psi_{j,n}}^{\text{D}_N} = \left\{ b_{\mu,\gamma}(\phi) = \begin{bmatrix} \cos(\mu\phi + \gamma) \\ \sin(\mu\phi + \gamma) \end{bmatrix} \,\middle|\, \mu = (m \pm n) + tN,\ \gamma = j\frac{\pi}{2} \right\}_{t \in \mathbb{Z}} \tag{45}$$

See the bottom left of Table 12.

**2 and 1-dimensional irreps:**

Similarly, for 2-dimensional input and 1-dimensional output representations $\rho_{\text{in}} = \psi_{1,n}^{\text{D}_N}$ and $\rho_{\text{out}} = \psi_{i,m}^{\text{D}_N}$ with $j = 1$, $n > 0$ and $m \in \{0, \frac{N}{2}\}$, we find:

$$\mathcal{K}_{\psi_{i,m} \leftarrow \psi_{j,n}}^{\text{D}_N} = \left\{ b_{\mu,\gamma}(\phi) = [\cos(\mu\phi + \gamma)\ \ \sin(\mu\phi + \gamma)] \,\middle|\, \mu = (\pm m + n) + tN,\ \gamma = i\frac{\pi}{2} \right\}_{t \in \mathbb{Z}}$$

$$\tag{46}$$

Table 12 shows these solutions in its top right cells.

### I.4.6 Kernel constraints at the origin

Our derivations rely on the fact that the kernel constraints restrict only the angular parts of the unconstrained kernel space $L^2(\mathbb{R}^2)^{c_{\text{out}} \times c_{\text{in}}}$ which suggests an independent solution for each radius $r \in \mathbb{R}^+ \cup \{0\}$. Particular attention is required for kernels defined at the origin, i.e. when $r = 0$. The reason for this is that we are using polar coordinates $(r, \phi)$ which are ambiguous at the origin where the angle is not defined. In order to stay consistent with the solutions for $r > 0$ we still define the kernel at the origin as an element of $L^2(S^1)^{c_{\text{out}} \times c_{\text{in}}}$. However, since the coordinates $(0, \phi)$ map to the same point for all $\phi \in [0, 2\pi)$, we need to demand the kernels to be angularly constant, that is, $\kappa(\phi) = \kappa(0)$. This additional constraint restricts the angular Fourier bases used in the previous derivations to zero frequencies only. Apart from this, the kernel constraints are the same for $r = 0$ and $r > 0$ which implies that the G-steerable kernel bases at $r = 0$ are given by restricting the bases derived in I.4.1, I.4.2, I.4.3, I.4.4 and I.4.5 to the elements indexed by frequencies $\mu = 0$.

### I.5 Complex valued representations and Harmonic Networks

Instead of considering real (irreducible) representations we could have derived all results using complex representations, acting on complex feature maps. For the case of $O(2)$ and $D_N$ this would essentially not affect the derivations since their complex and real irreps are equivalent, that is, they coincide up to a change of basis. Conversely, all complex irreps of $SO(2)$ and $C_N$ are 1-dimensional which simplifies the derivations in complex space. However, the solution spaces of complex G-steerable kernels need to be translated back to a real valued implementation. This translation has some not immediately obvious pitfalls which can lead to an underparameterized implementation in real space. In particular, Harmonic Networks [12] were derived with a complete solution in complex space; however, their real valued implementation is using a $G$-steerable kernel space of half the dimensionality as ours. We will in the following explain why this is the case.

In the complex field, the irreps of $SO(2)$ are given by $\psi_k^{\mathbb{C}}(\theta) = e^{ik\theta} \in \mathbb{C}$ with frequencies $k \in \mathbb{Z}$. Notice that these complex irreps are indexed by positive and negative frequencies while their real counterparts, defined in Appendix I.2, only involve non-negative frequencies. As in [12] we consider complex feature fields $f^{\mathbb{C}} : \mathbb{R}^2 \to \mathbb{C}$ which are transforming according to complex irreps of $SO(2)$. A complex input field $f_{\text{in}}^{\mathbb{C}} : \mathbb{R}^2 \to \mathbb{C}$ of type $\psi_n^{\mathbb{C}}$ is mapped to a complex output field $f_{\text{out}}^{\mathbb{C}} : \mathbb{R}^2 \to \mathbb{C}$ of type $\psi_m^{\mathbb{C}}$ via the cross-correlation

$$f_{\text{out}}^{\mathbb{C}} = k^{\mathbb{C}} \star f_{\text{in}}^{\mathbb{C}} \ . \tag{47}$$

with a complex filter $k^{\mathbb{C}} : \mathbb{R}^2 \to \mathbb{C}$. The (angular part of the) complete space of equivariant kernels between $f_{\text{in}}^{\mathbb{C}}$ and $f_{\text{out}}^{\mathbb{C}}$ was in [12] proven to be parameterized by

$$k^{\mathbb{C}}(\phi) \ = \ w \, e^{i(m-n)\phi},$$

where $w \in \mathbb{C}$ is a complex weight which scales and phase-shifts the complex exponential. We want to point out that an equivalent parametrization is given in terms of the real and imaginary parts $w^{\text{Re}}$ and $w^{\text{Im}}$ of the weight $w$, i.e.

$$\begin{aligned} k^{\mathbb{C}}(\phi) \ &= \ w^{\text{Re}} e^{i(m-n)\phi} + i \, w^{\text{Im}} e^{i(m-n)\phi} \\ &= \ w^{\text{Re}} e^{i(m-n)\phi} + w^{\text{Im}} e^{i((m-n)\phi + \pi/2)} \ . \end{aligned} \tag{48}$$

The real valued implementation of Harmonic Networks models the complex feature fields $f^{\mathbb{C}}$ of type $\psi_k^{\mathbb{C}}(\theta)$ by splitting them in two real valued channels $f^{\mathbb{R}} := (f^{\text{Re}}, f^{\text{Im}})^T$ which contain their real and imaginary part. The action of the complex irrep $\psi_k^{\mathbb{C}}(\theta)$ is modeled accordingly by a rotation matrix of the same, potentially negative[16] frequency. A real valued implementation of the cross-correlation (47) is built using a real kernel $k : \mathbb{R}^2 \to \mathbb{R}^{2 \times 2}$ as specified by

$$\begin{bmatrix} f_{\text{out}}^{Re} \\ f_{\text{out}}^{Im} \end{bmatrix} = \begin{bmatrix} k^{Re} & -k^{Im} \\ k^{Im} & k^{Re} \end{bmatrix} \star \begin{bmatrix} f_{\text{in}}^{Re} \\ f_{\text{in}}^{Im} \end{bmatrix} .$$

The complex steerable kernel (48) is then given by

$$k(\phi) = w^{\text{Re}} \begin{bmatrix} \cos\left((m-n)\phi\right) & -\sin\left((m-n)\phi\right) \\ \sin\left((m-n)\phi\right) & \cos\left((m-n)\phi\right) \end{bmatrix} + w^{\text{Im}} \begin{bmatrix} -\sin\left((m-n)\phi\right) & -\cos\left((m-n)\phi\right) \\ \cos\left((m-n)\phi\right) & -\sin\left((m-n)\phi\right) \end{bmatrix}$$

$$= w^{\text{Re}} \qquad \psi\left((m-n)\phi\right) \qquad + w^{\text{Im}} \qquad \psi\left((m-n)\phi + \frac{\pi}{2}\right) \qquad (49)$$

While this implementation models the *complex* Harmonic Networks faithfully in real space, it does not utilize the complete SO(2)-steerable kernel space when the real feature fields are interpreted as fields transforming under the real irreps $\psi_k^{\mathbb{R}}$ as done in our work. More specifically, the kernel space used in (49) is only 2-dimensional while our basis (23) for the same case is 4-dimensional. The additional solutions with frequency $m + n$ are missing.

The lower dimensionality of the complex solution space can be understood by analyzing the relationship between SO(2)'s real and complex irreps. On the complex field, the real irreps become reducible and decomposes into two 1-dimensional complex irreps with opposite frequencies:

$$\psi_k^{\mathbb{R}}(\theta) = \frac{1}{\sqrt{2}} \begin{bmatrix} 1 & -i \\ -i & 1 \end{bmatrix} \begin{bmatrix} e^{ik\theta} & 0 \\ 0 & e^{-ik\theta} \end{bmatrix} \frac{1}{\sqrt{2}} \begin{bmatrix} 1 & i \\ i & 1 \end{bmatrix}$$

Indeed, SO(2) has only half as many real irreps as complex ones since positive and negative frequencies are conjugated to each other, i.e. they are equivalent up to a change of basis: $\psi_k^{\mathbb{R}}(\theta) = \xi(-1)\psi_{-k}^{\mathbb{R}}(\theta)\xi(-1)$. It follows that a real valued implementation of a complex $\psi_k^{\mathbb{C}}$ fields as a 2-dimensional $\psi_k^{\mathbb{R}}$ fields implicitly adds a complex $\psi_{-k}^{\mathbb{C}}$ field. The intertwiners between two real fields of type $\psi_n^{\mathbb{R}}$ and $\psi_m^{\mathbb{R}}$ therefore do not only include the single complex intertwiner between complex fields of type $\psi_n^{\mathbb{C}}$ and $\psi_m^{\mathbb{C}}$, but four complex intertwiners between fields of type $\psi_{\pm n}^{\mathbb{C}}$ and $\psi_{\pm m}^{\mathbb{C}}$. The real parts of these intertwiners correspond to our four dimensional solution space.

In conclusion, [12] indeed found the complete solution on the complex field. However, by implementing the network on the real field, negative frequencies are implicitly added to the feature fields which allows for our larger basis (23) of steerable kernels to be used without adding an overhead.

## J  Alternative approaches to compute kernel bases and their complexities

The main challenge of building steerable CNNs is to find the space of solutions of the kernel space constraint in Eq. 2. Several recent works tackle this problem for the very specific case of features which transform under *irreducible* representations of $\text{SE}(3) \cong (\mathbb{R}^3, +) \rtimes \text{SO}(3)$. The strategy followed in [35, 36, 15, 37] is based on well known analytical solutions and does not generalize to arbitrary representations. In contrast, [2] present a numerical algorithm to solve the kernel space constraint. While this algorithm was only applied to solve the constraints for irreps, it generalizes to arbitrary representations. However, the computational complexity of the algorithm scales unfavorably in comparison to the approach proposed in this work. We will in the following review the kernel space solution algorithm of [2] for general representations and discuss its complexity in comparison to our approach.

The algorithm proposed in [2] is considering the same kernel space constraint

$$k(gx) = \rho_{\text{out}}(g)k(x)\rho_{\text{in}}^{-1}(g) \quad \forall g \in G$$

as in this work. By vectorizing the kernel the constraint can be brought in the form

$$\text{vec}(k)(gx) = \left(\rho_{\text{out}} \otimes \left(\rho_{\text{in}}^{-1}\right)^T\right)(g) \, \text{vec}(k)(x)$$
$$= \left(\rho_{\text{out}} \otimes \rho_{\text{in}}\right)(g) \, \text{vec}(k)(x),$$

where the second step assumes the input representation to be unitary, that is, to satisfy $\rho_{\text{in}}^{-1} = \rho_{\text{in}}^T$. A Clebsch-Gordan decomposition, i.e. a decomposition of the tensor product representation into a direct sum of irreps $\psi_j$ of $G$, then yields[17]

$$\text{vec}(k)(gx) = Q^{-1}\left(\bigoplus_{J \in \mathcal{J}} \psi_J\right)(g) \, Q \, \text{vec}(k)(x)$$

Through a change of variables $\eta(x) := Q\operatorname{vec}(k)(x)$ this simplifies to

$$\eta(gx) = \left(\bigoplus_{J \in \mathcal{J}} \psi_J\right)(g)\eta(x)$$

which, in turn, decomposes into $|\mathcal{J}|$ independent constraints

$$\eta_J(gx) = \psi_J(g)\eta_J(x) \ .$$

Each of these constraints can be solved independently to find a basis for each $\eta_J$. The kernel basis is then found by inverting the change of basis and the vectorization, i.e. by computing $k(x) = \operatorname{unvec}\left(Q^{-1}\eta(x)\right)$.

For the case that $\rho_{\mathrm{in}} = \psi_j$ and $\rho_{\mathrm{out}} = \psi_l$ are Wigner D-matrices, i.e. irreps of $\mathrm{SO}(3)$, the change of basis $Q$ is given by the Clebsch-Gordan coefficients of $\mathrm{SO}(3)$. These well known solutions were used in [35, 36, 15, 37] to build the basis of steerable kernels. Conversely, the authors of [2] solve for the change of basis $Q$ numerically. Given *arbitrary* unitary representations $\rho_{\mathrm{in}}$ and $\rho_{\mathrm{out}}$ the numerical algorithm solves for the change of basis in

$$\left(\rho_{\mathrm{in}} \otimes \rho_{\mathrm{out}}\right)(g) \;=\; Q^{-1}\left(\bigoplus_{J \in \mathcal{J}} \psi_J(g)\right)Q \qquad\qquad \forall g \in G$$

$$\Leftrightarrow \qquad\qquad 0 \;=\; Q\left(\rho_{\mathrm{in}} \otimes \rho_{\mathrm{out}}\right)(g) \;-\; \left(\bigoplus_{J \in \mathcal{J}} \psi_J(g)\right)Q \qquad \forall g \in G\ .$$

This linear constraint on $Q$, which is a specific instance of the Sylvester equation, can be solved by vectorizing $Q$, i.e.

$$\left[I \otimes \left(\rho_{\mathrm{in}} \otimes \rho_{\mathrm{out}}\right)(g) \;-\; \left(\bigoplus_{J \in \mathcal{J}} \psi_J\right)(g) \otimes I\right]\operatorname{vec}(Q) \;=\; 0 \qquad \forall g \in G\ ,$$

where $I$ is the identity matrix on $\mathbb{R}^{\dim(\rho_{\mathrm{in}} \otimes \rho_{\mathrm{out}})} = \mathbb{R}^{\dim(\rho_{\mathrm{in}})\dim(\rho_{\mathrm{out}})}$ and $\operatorname{vec}(Q) \in \mathbb{R}^{\dim(\rho_{\mathrm{in}})^2 \dim(\rho_{\mathrm{out}})^2}$. In principle there is one Sylvester equation for each group element $g \in G$, however, it is sufficient to consider the *generators* of $G$ only, since the solutions found for the generators will automatically hold for all group elements. One can therefore stack the matrices $\left[I \otimes \left(\rho_{\mathrm{in}} \otimes \rho_{\mathrm{out}}\right)(g) - \left(\bigoplus_{J \in \mathcal{J}} \psi_J\right)(g) \otimes I\right]$ for the generators of $G$ into a bigger matrix and solve for $Q$ as the null space of this stacked matrix. The linearly independent solutions $Q^J$ in the null space correspond to the Clebsch-Gordan coefficients for $J \in \mathcal{J}$.

This approach does not rely on the analytical Clebsch-Gordan coefficients, which are only known for specific groups and representations, and therefore works for any choice of representations. However, applying it naively to large representations can be extremely expensive. Specifically, computing the null space to solve the (stacked) Sylvester equation for $\mathcal{G}$ generators of $G$ via a *SVD*, as done in [2], scales as $\mathcal{O}\left(\dim(\rho_{\mathrm{in}})^6 \dim(\rho_{\mathrm{out}})^6 \mathcal{G}\right)$. This is the case since the matrix which is multiplying $\operatorname{vec}(Q)$ is of shape $\dim(\rho_{\mathrm{in}})^2 \dim(\rho_{\mathrm{out}})^2 \mathcal{G} \times \dim(\rho_{\mathrm{in}})^2 \dim(\rho_{\mathrm{out}})^2$. Moreover, the change of basis matrix $Q$ itself has shape $\dim(\rho_{\mathrm{in}})\dim(\rho_{\mathrm{out}}) \times \dim(\rho_{\mathrm{in}})\dim(\rho_{\mathrm{out}})$ which implies that the change of variables[18] from $\eta$ to $k$ has complexity $\mathcal{O}\left(\dim(\rho_{\mathrm{in}})^2 \dim(\rho_{\mathrm{out}})^2\right)$. In [2] the authors only use irreducible representations which are relatively small such that the bad complexity of the algorithm is negligible.

In comparison, the algorithm proposed in this work is based on an *individual* decomposition of the representations $\rho_{\mathrm{in}}$ and $\rho_{\mathrm{out}}$ into irreps and leverages the analytically derived kernel space solutions between the irreps of $G \leq \mathrm{O}(2)$. The independent decomposition of the input and output representations leads to a complexity of only $\mathcal{O}\left(\left(\dim(\rho_{\mathrm{in}})^6 + \dim(\rho_{\mathrm{in}})^6\right)\mathcal{G}\right)$. We further apply the input and output changes of basis $Q_{\mathrm{in}}$ and $Q_{\mathrm{out}}$ independently to the irreps kernel solutions $\kappa^{ij}$ which leads to a complexity of $\mathcal{O}\left(\dim(\rho_{\mathrm{in}})\dim(\rho_{\mathrm{out}})^2 + \dim(\rho_{\mathrm{out}})\dim(\rho_{\mathrm{in}})^2\right)$. The improved complexity of our implementation makes working with large representations as used in this work, for instance $\dim(\rho_{\mathrm{reg}}^{\mathrm{D}_{20}}) = 40$, possible.

# K   Additional information on the training setup

| layer | output fields |
|---|---|
| conv block $7 \times 7$ (pad 1) | 16 |
| conv block $5 \times 5$ (pad 2) | 24 |
| max pooling $2 \times 2$ | 24 |
| conv block $5 \times 5$ (pad 2) | 32 |
| conv block $5 \times 5$ (pad 2) | 32 |
| max pooling $2 \times 2$ | 32 |
| conv block $5 \times 5$ (pad 2) | 48 |
| conv block $5 \times 5$ | 64 |
| invariant projection | 64 |
| global average pooling | 64 |
| fully connected | 64 |
| fully connected + softmax | 10 |

Table 13: Basic model architecture from which all models for the MNIST benchmarks in Tables 7 and 2 are being derived. Each convolution block includes a convolution layer, batch-normalization and a nonlinearity. The first fully connected layer is followed by batch-normalization and ELU. The width of each layer is expressed as the number of fields of a regular $C_{16}$ model with approximately the same number of parameters.

| layer | output fields |
|---|---|
| conv block $9 \times 9$ | 24 |
| conv block $7 \times 7$ (pad 3) | 32 |
| max pooling $2 \times 2$ | 32 |
| conv block $7 \times 7$ (pad 3) | 36 |
| conv block $7 \times 7$ (pad 3) | 36 |
| max pooling $2 \times 2$ | 36 |
| conv block $7 \times 7$ (pad 3) | 64 |
| conv block $5 \times 5$ | 96 |
| invariant projection | 96 |
| global average pooling | 96 |
| fully connected | 96 |
| fully connected | 96 |
| fully connected + softmax | 10 |

Table 14: Model architecture for the final MNIST-rot experiments (replicated from [7]). Each fully connected layer follows a dropout layer with $p = 0.3$; the first two fully connected layers are followed by batch normalization and ELU. The width of each layer is expressed in terms of regular feature fields of a $C_{16}$ model.

## K.1   Benchmarking on transformed MNIST datasets

Each model reported in Section 3.1 and Appendix D.1 is derived from the architecture reported in Table 13. The width of each model's layers is thereby scaled such that the total number of parameters is matched and the relative width of layers coincides with that reported in Table 13. Training is performed with a batch size of 64 samples, using the *Adam* optimizer. The learning rate is initialized to $10^{-3}$ and decayed exponentially by a factor of $0.8$ per epoch, starting after a burn in phase of 10 epochs. We train each model for 30 epochs and test the model which performed best on the validation set. A weight decay of $10^{-7}$ is being used for all convolutional layers and the first fully connected layer. In all experiments, we build steerable bases with Gaussian radial profiles of width $\sigma = 0.6$ for all except the outermost ring where we use $\sigma = 0.4$. We apply a strong bandlimiting policy which permits frequencies up to $0, 2, 2$ for radii $0, 1, 2$ in a $5 \times 5$ kernel and up to $0, 2, 3, 2$ for radii $0, 1, 2, 3$ in a $7 \times 7$ kernel. The strong cutoff in the rings of maximal radius is motivated by our empirical observation that these rings introduce a relatively high equivariance error for higher frequencies. This is the case since the outermost ring ranges out of the sampled kernel support. During training, data augmentation with continuous rotations and reflections is performed (if these are present in the dataset) to not disadvantage non-equivariant models. In the models using group restriction, the restriction operation is applied after the convolution layers but before batch normalization and non-linearities.

## K.2   Competitive runs on MNIST rot

In Table 3 we report the performances of some of our best models. Our experiments are based on the best performing, $C_{16}$-equivariant model of [7] which defined the state of the art on rotated MNIST at the time of writing. We replicate their model architecture, summarized in Table 14, though our models have a different frequency bandlimit and width $\sigma$ for the Gaussian radial profiles as discussed in the previous subsection. As before, the table reports the width of each layer in terms of number of fields in the $C_{16}$ regular model.

As commonly done, we train our final models on the 10000 + 2000 training and validation samples. Training is performed for 40 epochs with an initial learning rate $0.015$, which is being decayed by a factor of $0.8$, starting after 15 epochs. As before, we use the *Adam* optimizer with a batch size of 64, this time using L1 and L2 regularization with a weight of $10^{-7}$. The fully connected layers are

additionally regularized using dropout with a probability of $p = 0.3$. We are again using train time augmentation.

## K.3 CIFAR experiments

The equivariant models used in the experiments on CIFAR-10 and CIFAR-100 are adapted from the original WideResNet models by replacing conventional with $G$-steerable convolutions and scaling the number of feature fields such that the total number of parameters is preserved. For blocks which are equivariant under $\mathrm{D}_8$ or $\mathrm{C}_8$ we use $5 \times 5$ kernels instead of $3 \times 3$ kernels to allow for higher frequencies. All models use regular feature fields in all but the final convolution layer, which maps to a scalar field (conv2triv) to produce invariant predictions. We use a frequency cut-off of 3 times the ring's radius, e.g. $0, 3, 6$ for rings of radii $0, 1, 2$. These higher bandlimits in comparison to the MNIST experiments are motivated by the fact that the corresponding bases introduce small discretization errors, which is no problem for the classification of natural images. In the contrary, this leads to the models having a strong bias towards being equivariant, but might allow them to break equivariance if necessary. The widths of the bases' rings is chosen to be $\sigma = 0.45$ in all rings.

The training process is the same as used for WideResNets: we train for 200 epochs with a batch size of 128. We optimize the model with SGD, using an initial learning rate of $0.1$, momentum $0.9$ and a weight decay of $5 \cdot 10^{-4}$. The learning rate is decayed by a factor of $0.2$ every 60 epochs. We perform a standard data augmentation with random crops, horizontal flips and normalization. No CutOut is done during the normal experiments but it is used in the AutoAugment policies.

## K.4 STL-10 experiments

The models for our STL-10 experiments are adapted from [32]. However, according to an issue[19] in the authors' GitHub repository, the publication states some model parameters and the training setup wrongly. Our adaptations are therefore based on the setting reported on GitHub. Specifically, we use patches of $60 \times 60$ pixels for cutout and the stride of the first convolution layer in the first block is 2 instead of 1. Moreover, we normalize input features using CIFAR-10 statistics. Though these statistics are very close to the statistics of STL-10, they might, as the authors of [32] suggest, cause non-negligible changes in performance because of the small training set size of STL-10.

As before, regular feature fields are used throughout the whole model except for the last convolution layer which maps to trivial fields. In the small model, which does not preserve the number of parameters but the number of channels, we still scale up the number of output channels of the very first convolution layer (before the first residual block). As the first convolution layer originally has 16 output channels and our model is initially equivariant to $\mathrm{D}_8$ (whose regular representation spans 16 channels), the first convolution layer would only be able to learn 1 single independent filter (repeated 16 times, rotated and reflected). Hence, we increase the number of output channels of the first convolution layer by the square root of the group size ($\sqrt{16} = 4$) leading to $4 \cdot 16 = 64$ channels, i.e. $64/16 = 4$ regular fields. We use a ring width of $\sigma = 0.6$ for the kernel basis except for the outermost ring where we use $\sigma = 0.4$ and use a frequency cut-off factor of 3 for the rings' radii, i.e. cutoffs of $0, 3, 6, \ldots$.

We are again exactly replicating the training process as reported in the publication [32]. Only the labeled subset of the training set is used, that is, the $100000$ unlabeled training images are discarded. Training is performed for 1000 epochs with a batch size of 128, using SGD with Nesterov momentum of $0.9$ and weight decay of $5 \cdot 10^{-4}$. The learning rate is initialized to $0.1$ and decayed by a factor of 5 at $300, 400, 600$ and $800$ epochs. During training, we perform data augmentation by zero-padding with 12 pixels and randomly cropping patches of $96 \times 96$ pixels, mirroring them horizontally and applying CutOut.

## L  Additional information on the irrep models

**SO(2) models**   We experiment with some variants (rows 37-44) of the Harmonic Network model in row 30 of Table 7, varying in either the non-linearity or the invariant map applied. All of these models are therefore to be analyzed relative to this baseline. First, we try to use *squashing* nonlinearities [41]

(row 37) instead of norm-ReLUs on each non-trivial irrep. This variant performs consistently worse than the original model. In the baseline variant, we generate invariant features via a convolution to scalar fields in the last layer (*conv2triv*). This, however, reduces the utilization of high frequency irrep fields in the penultimate layer. The reason for this is that the kernel space for mappings from high frequency- to scalar fields consists of kernels of a high angular frequency, which will be cut off by our bandlimiting. To overcome this problem, we propose to instead compute the norms of all non-trivial fields to produce invariant features. This enables us to use all irreps in the output of the last convolutional layer. However, we find that combining invariant norm mappings with norm-ReLUs does not improve on the baseline model, see row 38. In row 39 we consider a variant which applies norm-ReLUs on the direct sum of multiple non-trivial irrep fields, each with multiplicity 1, together (*shared norm-ReLU*), while the scalar fields are still being acted on by ELUs. This is legitimate since the direct sum of unitary representations is itself unitary. After the last convolutional layer, the invariant projection preserves the trivial fields but computes the norm of each composed field. This model significantly outperforms all previous variants on all datasets. The model in row 40 additionally merges the scalar fields to such combined fields instead of treating them independently. This architecture performs significantly worse than the previous variants.

We further explore four different variations which are applying *gated nonlinearities* (rows 41-44). These models distinguish from each other by 1) their mapping to invariant features and 2) whether the gate is being applied to each non-trivial field independently or being shared between multiple non-trivial fields. We find that the second choice, i.e. sharing gates, does not significantly affect the performances (row 41 vs. 42 and 43 vs. 44). However, mapping to invariant features by taking the norm of all non-trivial fields performs consistently better than applying *conv2triv*. Overall, gated nonlinearities perform significantly better than any other choice of nonlinearity on the tested $\mathrm{SO}(2)$ irrep models.

**O(2) models**   Here we will give more details on the $\mathrm{O}(2)$-specific operations which we introduce to improve the performance of the $\mathrm{O}(2)$-equivariant models, reported in rows 45-57 of Table 7.

- O(2)-*conv2triv:* As invariant map of the $\mathrm{O}(2)$ irrep models in rows 46-49 and 54 we are designing a last convolution layer which is mapping to an output representation $\rho_{\mathrm{out}} = \psi_{0,0}^{\mathrm{O}(2)} \oplus \psi_{1,0}^{\mathrm{O}(2)}$, that is, to scalar fields $f_{0,0}$ and sign-flip fields $f_{1,0}$ in equal proportions. Since the latter are not invariant under reflections, we are in addition taking their absolute value. The resulting, invariant output features are then multiple fields $f_{0,0} \oplus |f_{1,0}|$. The motivation for not convolving to trivial representations of $\mathrm{O}(2)$ directly via *conv2triv* is that the steerable kernel space for mappings between irreps of $\mathrm{O}(2)$ does not allow for mapping between $\psi_{0,0}^{\mathrm{O}(2)}$ and $\psi_{1,0}^{\mathrm{O}(2)}$ (see Table 9), which would lead to dead neurons.

The models in rows 50-53, 56 and 57 operate on $\mathrm{Ind}_{\mathrm{SO}(2)}^{\mathrm{O}(2)} \psi_k^{\mathrm{SO}(2)}$-fields whose representations are induced from the irreps of $\mathrm{SO}(2)$. Per definition, this representation acts on feature vectors $f$ in $\mathbb{R}^{\dim(\psi_k^{\mathrm{SO}(2)})} \otimes \mathbb{R}^{|\,\mathrm{O}(2):\mathrm{SO}(2)|}$, which we treat in the following as functions $f : \mathrm{O}(2)/\mathrm{SO}(2) \to \mathbb{R}^{\dim(\psi_k^{\mathrm{SO}(2)})}$. We further identify the coset $s\,\mathrm{SO}(2)$ in the quotient space $\mathrm{O}(2)/\mathrm{SO}(2)$ by its representative $\mathcal{R}(s\,\mathrm{SO}(2)) := s \in (\{\pm 1\}, *)$ in the reflection group. Eq. 9 defines the action of the induced representation on a feature vector by

$$\left( \left[ \mathrm{Ind}_{\mathrm{SO}(2)}^{\mathrm{O}(2)} \psi_k^{\mathrm{SO}(2)} \right](\tilde{r}\tilde{s}) \, f \right)(s\,\mathrm{SO}(2)) := \psi_k^{\mathrm{SO}(2)} \left( \mathrm{h}\left( \tilde{r}\tilde{s}\mathcal{R}((\tilde{r}\tilde{s})^{-1}s\,\mathrm{SO}(2)) \right) \right) f\left( (\tilde{r}\tilde{s})^{-1} s\,\mathrm{SO}(2) \right)$$

$$= \psi_k^{\mathrm{SO}(2)} \left( \mathrm{h}(\tilde{r}s) \right) f\left( \tilde{s}s\,\mathrm{SO}(2) \right)$$

$$= \begin{cases} \psi_k^{\mathrm{SO}(2)}\left( \tilde{r} \right) f\left( \tilde{s}s\,\mathrm{SO}(2) \right) & \text{for } s = +1 \\ \psi_k^{\mathrm{SO}(2)}\left( \tilde{r}^{-1} \right) f\left( \tilde{s}s\,\mathrm{SO}(2) \right) & \text{for } s = -1 \,, \end{cases}$$

where we used Eq. 7 to compute

$$\mathrm{h}(\tilde{r}s) := \mathcal{R}\left( \tilde{r}s\,\mathrm{SO}(2) \right)^{-1} \tilde{r}s = s^{-1} \tilde{r}s = \begin{cases} \tilde{r} & \text{for } s = +1 \\ \tilde{r}^{-1} & \text{for } s = -1 \,. \end{cases}$$

Intuitively, this action describes a permutation of the subfields (indexed by $s$) via the reflection $\tilde{s}$ and a rotation of the subfields by $\tilde{r}$ and $\tilde{r}^{-1}$, respectively. Specifically, for $k = 0$, the induced representation is for all $\tilde{r}$ instantiated by

$$\left[ \mathrm{Ind}_{\mathrm{SO(2)}}^{\mathrm{O(2)}} \psi_0^{\mathrm{SO(2)}} \right](\tilde{r}\tilde{s}) = \begin{cases} \begin{bmatrix} 1 & 0 \\ 0 & 1 \end{bmatrix} & \text{for } \tilde{s} = +1 \\ \begin{bmatrix} 0 & 1 \\ 1 & 0 \end{bmatrix} & \text{for } \tilde{s} = -1 \ , \end{cases} \tag{50}$$

that is, it coincides with the regular representation of the reflection group. Similarly, for $k > 0$, it is for all $\tilde{r}$ given by the $4 \times 4$ matrices

$$\left[ \mathrm{Ind}_{\mathrm{SO(2)}}^{\mathrm{O(2)}} \psi_{k>0}^{\mathrm{SO(2)}} \right](\tilde{r}\tilde{s}) = \begin{cases} \left[ \begin{array}{c|c} \psi_{k>0}^{\mathrm{SO(2)}}(\tilde{r}) & 0 \\ \hline 0 & \psi_{k>0}^{\mathrm{SO(2)}}(-\tilde{r}) \end{array} \right] & \text{for } \tilde{s} = +1 \\ \left[ \begin{array}{c|c} 0 & \psi_{k>0}^{\mathrm{SO(2)}}(\tilde{r}) \\ \hline \psi_{k>0}^{\mathrm{SO(2)}}(-\tilde{r}) & 0 \end{array} \right] & \text{for } \tilde{s} = -1 \ . \end{cases}$$

We adapt the conv2triv and norm invariant maps, as well as the norm-ReLU and the gated nonlinearities to operate on $\mathrm{Ind}_{\mathrm{SO(2)}}^{\mathrm{O(2)}}$-fields as follows:

- Ind-*conv2triv:* Instead of applying *O(2)-conv2triv* to compute invariant features, we apply convolutions to $\mathrm{Ind}_{\mathrm{SO(2)}}^{\mathrm{O(2)}} \psi_0^{\mathrm{SO(2)}}$-fields which are invariant under rotations but behave like regular $(\{\pm 1\}, *)$-fields under reflections. These fields are subsequently mapped to a scalar field via $G$-pooling, i.e. by taking the maximal response over the two subfields.

- Ind-*norm:* An alternative invariant map is defined by computing the norms of the subfields of each final $\mathrm{Ind}_{\mathrm{SO(2)}}^{\mathrm{O(2)}} \psi_k^{\mathrm{SO(2)}}$-field and applying $G$-pooling over the result.

- Ind *norm-ReLU:* It would be possible to apply a norm-ReLU to a $\mathrm{Ind}_{\mathrm{SO(2)}}^{\mathrm{O(2)}} \psi_k^{\mathrm{SO(2)}}$-field for $k > 0$ as a whole, that is, to compute the norm of both subfields together. Instead, we apply two individual norm-ReLUs to the subfields. Since the fields permute under reflections, we need to choose the bias parameter of the two norm-ReLUs to be equal.

- Ind *gate:* Similarly, we could apply a single gate to each $\mathrm{Ind}_{\mathrm{SO(2)}}^{\mathrm{O(2)}} \psi_k^{\mathrm{SO(2)}}$-field. However, we apply an individual gate to each subfield. In this case it is necessary that the gates permute together with the $\mathrm{Ind}_{\mathrm{SO(2)}}^{\mathrm{O(2)}} \psi_k^{\mathrm{SO(2)}}$-fields to ensure equivariance. This is achieved by computing the gates from $\mathrm{Ind}_{\mathrm{SO(2)}}^{\mathrm{O(2)}} \psi_0^{\mathrm{SO(2)}}$-fields, which contain two permuting scalar fields.

Empirically we find that $\mathrm{Ind}_{\mathrm{SO(2)}}^{\mathrm{O(2)}}$ models perform much better than pure irrep models, despite both of them being equivalent up to a change of basis. In particular, the induced representations decompose for some change of basis matrices $Q_0$ and $Q_{>0}$ into:

$$\mathrm{Ind}_{\mathrm{SO(2)}}^{\mathrm{O(2)}} \psi_0^{\mathrm{SO(2)}} = Q_0 \left[ \psi_{0,0}^{\mathrm{O(2)}} \oplus \psi_{1,0}^{\mathrm{O(2)}} \right] Q_0^{-1}$$
$$\mathrm{Ind}_{\mathrm{SO(2)}}^{\mathrm{O(2)}} \psi_{k>0}^{\mathrm{SO(2)}} = Q_{>0} \left[ \psi_{1,k>0}^{\mathrm{O(2)}} \oplus \psi_{1,k>0}^{\mathrm{O(2)}} \right] Q_{>0}^{-1}$$

The difference between both bases is that the induced representations disentangle the action of reflections into a permutation, while the direct sum of irreps is modeling reflections in each of its sub-vectorfields independently as an inversion of the vector direction and rotation orientation. Note the analogy to the better performance of regular representations in comparison to a direct sum of the respective irreps.

## Footnotes

[4] Conversely, the equivariance under local gauge transformations $g(x) \in \mathrm{O}(2)$ implies the equivariance under active isometries. In the case of the Euclidean space $\mathbb{R}^2$ these isometries are given by the Euclidean group E(2).

[5] Note that this prevents equivariance from being exact for groups which are not symmetries of the grid. Specifically, for $\mathbb{Z}^2$ only subgroups of $D_4$ are exact symmetries which motivated their use in [6, 10, 1].

[6] The same decomposition was used in a different context in Section 2.4.

[7] The group restricted models are not listed in Table 7 but in Table 2.

[8] E.g. **6** and **9** (6 and 9) or **2** and **5** (2 and 5) are related by a rotations by $\pi$ and might therefore be confused by all models $C_{2k}$ and $D_{2k}$ for $k \in \mathbb{N}$. Similarly, **4** and **7** (4 and 7) are related by a reflection and a rotation by $\pi/2$ and might be confused by all models $D_{4k}$.

[9]The vector can equivalently be expressed as $w = \bigoplus_{gH} w_{gH}$, however, we want to make the tensor product basis explicit.

[10] Formally, a representative for each coset is chosen by a map $\mathcal{R} : G/H \to G$ such that it projects back to the same coset, i.e. $\mathcal{R}(gH)H = gH$. This map is therefore a *section* of the principal bundle $G \xrightarrow{\pi} G/H$ with fibers isomorphic to $H$ and the projection given by $\pi(g) := gH$.

[11] The rhs. of Eq. (8) corresponds to $[\text{Ind}_H^G \rho(\tilde{g}) \cdot f](\tilde{g}gH) = \rho(\text{h}(\tilde{g}\mathcal{R}(gH)))f(gH)$.

[12] Or more generally, *any* possible pattern.

[13]Trivial representations $\psi^0 \cong \rho_{\text{quot}}^{G/G}$ can themself be seen as an extreme case of quotient representations which are invariant to the full group $G$.

[14] For brevity, we suppress that frequency 0 is associated to only half the number of basis elements which does not affect the validity of the derivation.

[15]Notice that for $\mu = 0$ some of the elements of the set are zero and are therefore not part of the basis. We omit this detail to reduce clutter.

[16]This establishes an isomorphism between $\psi_k^{\mathbb{C}}(\theta)$ and $\psi_{|k|}^{\mathbb{R}}(\theta)$ depending on the sign of $k$.

[17]For the irreps of SO(3) it is well known that $\mathcal{J} = \{|j - l|, \ldots, j + l\}$ and $|\mathcal{J}| = 2\min(j, l) + 1$.

[18]No inversion from $Q$ to $Q^{-1}$ is necessary if the Sylvester equation is solved directly for $Q^{-1}$.

[19]`https://github.com/uoguelph-mlrg/Cutout/issues/2`