[Reviews · NeurIPS 2019]

Reviewer 1



While a general solution is indeed a contribution to the literature, it does not seem overly significant. Even the motivation in S1 on p1 states "Unfortunately, an empirical survey, reproducing and comparing all these different approaches, is still missing." The paper organization is questionable. Section 2 has eight subsections with new work starting in sub-section 2.4. Perhaps it would be clearer to include background information in section 2 and start new work in section 3. There are grammatical errors, including the first sentence of the Abstract. The paper flip-flops between CNNs and convolutional neural networks on p2, for instance. Pick one and be consistent.

Reviewer 2



The paper is generally well written and has sufficient experiments. The problem is an interesting one and the presented solution seems solid. The improvements in the results also seem significant.

Reviewer 3



Classical image based CNNs are equivariant to translations (modulo pooling) and this is perhaps a major reason for their immense success. A image recognition task however also contains various other types of symmetries, that are usually incorporated by means of data augmentation. In order to incorporate more symmetries in a principled manner, such that they obviate data augmentation for those symmetries, group equivariant convolutional networks were proposed. Originally they incorporated simple symmetries such as 90 degree rotations, reflections in addition to translations. This was followed by work incorporating 360 degree rotations as in harmonic networks, gated harmonic networks and so on. In this paper using the theory of steerable CNNs, the authors work out a general strategy for equivariance to the euclidean group (that is the semi-direct product of translations with O(2)). In particular, for the Euclidean group, they work out kernel constraints that enforces equivariance, while generalizing a range of work on steerable CNNs for images. Almost all of the paper is very cleanly and elegantly presented with almost no points that were unclear. The supplementary material also does a good job to fill in proofs. Thus except a nod of approval I don't have specific concerns or criticisms. In summary -- the paper is self contained and makes a strong contribution. However, one concern that could be envisaged is that this work is simply working out details for a particular group, and using machinery that has now pretty much become standard. So why is it interesting? In addition to the points mentioned in the box above, it is perhaps worth emphasizing that the most significant advances in vision research and in industry mostly rely on CNNs working on the plane. Fancier models working on more exotic spaces still remain at quite a distance in terms of the sort of impact classical CNNs have had. It is thus nice to improve models in that space even further in a principled manner. Besides, a lot of the machinery described here is directly useful for manifold CNNs. The experiments on MNIST (including those in the supplement) are really thorough. It is nice to see different combinations, while also working with restricted variants, while also checking the hypothesis that the equivariant models are able to exploit local symmetries. It is a bit surprising (although perhaps not a lot) though that for CIFAR even after autoaugment, there was a considerable improvement. I have believed that only a good sampling of the transformations is usually enough for good performance (which I think of as a valid criticism of GCNNs). Thus I might not have expected a very substantial gain. Although I do feel a bit concerned that such vetting of architectures might lead to some overfitting overall. Some minor comments -- Please defined induced and restricted representations in the supplementary. Line 81: Awkward sentence/typo: "could either be of.." Line 294: Awkward sentence. Line 313: Typo "bad choices" LIne 332: Typo "loss of informations"

Reviewer 4



As stated above, the work makes several important contributions to the field. However, there are a few problems, as will be discussed in the following. First, many, if not most, of the theoretical results of the work, including the steerable kernel constraint and its reduction to irrep constraints, have already been published elsewhere (notably, in [1–5]). The theoretical novelty of the present work is how the framework is presented, in the detailed derivation of steerable kernel bases for various subgroups of E(2). While the authors do indeed cite these previous works, they do not always make entirely clear that the new contribution is the synthesis and generalization of these previous results. This should be made clearer. One interesting aspect that is also not considered is the relationship between the framework and more general architectures, such as [11]. How does the proposed equivariant network fit into these more general notions of group equivariance? In the other direction, the authors may also want to discuss the relationship with the work of Mallat, Sifre, and collaborators, who have proposed networks equivariant (and invariant) to rigid motions, albeit with fixed, not learned, filters. On a more technical level, the authors do not fully motivate the need for steerable (i.e., non-scalar) fields. Although this may be worked out by the reader, the work misses an important opportunity to explain them before diving into the details of the steerable kernel constraint. In particular, Section 2.2 introduces steerable feature fields (with scalar and vector fields as special cases), but do not motivate them beyond stating that “the feature fields of steerable CNNs are associated with a transformation law which specifies their transformation under actions of E(2) (or subgroups) and therefore endows features with a notion of orientation.” While this is technically true, it is a subtle, but important, point that deserves to be expanded further. For example, for scalar input and output fields, eq. (2) enforces circular symmetry when SO(2) is a subgroup of G. The section on representations (Section 2.6) is a little confusing in its treatment of regular/quotient representations and irreps. First, what's the intuition for using a quotient representation instead of a full regular representation? The smaller number of parameters is certainly useful, but how does this affect the expressivity of the network? Furthermore, these are only available for discrete subgroups G of O(2). The case of infinite subgroups (such as SO(2) and O(2)) is not discussed. It seems that in this case, the only option is to use the irreps themselves (and the benchmarks seem to confirm this) such as for harmonic nets [12]. Overall, this section tries to cover many points (non-equivalence of representations for non-linearities, categories of representations, isometric representations, and types of useful non-linearities). If these discussions were better separated, it might make the text easier to follow. The discussion on implementation is missing a discussion on discretization. This is odd considering that discretizing the fields and kernels destroys exact equivariance (unless G is a subgroup of C₄ or D₄). In fact, certain previous works (such as [1]), restrict themselves to such subgroups specifically for that reason. Others, which do not (such as [12]), still discuss the effect of discretization and why the equivariance properties can still be expected to hold approximately. It's an odd omission in this paper. Further, it is not clear what exactly is meant by the statement that “we sample the analytically found bases … of the irrep kernel constrains given in Section C.3 on a square grid of size s × s.” Section C.3 specifies constraints in the angular Fourier domain, but presumably the authors mean some corresponding Cartesian grid. If so, what's the grid spacing? How do we choose s? It would be worth specifying this information, at least in the supplementary. The experimental results (Section 3) presents many interesting results, but spends very little space discussing the “final runs.” These are simply presented, without any explanation other than to state that they outperform the state of the art. It is only later (in the supplementary) that we find that this is actually a different architecture (presumably a more powerful variant of that used in the other benchmarks). Since this is what achieves the state of the art performance, it is worth describing in more detail as part of the main text. The references have capitalization errors in [2], [4], [17], and [24]. References [1], [6], [7], [8], [9], [12], [14], [22], and [24] are missing periods after middle initials and [9] is missing the name of the last author. Section C.4 derives the kernel constraints and provides a good level of detail. However, the authors only seem to cover the kernel constraints for pairs of two-dimensional irreps, neglecting pairs involving one-dimensional irreps. It may be the case that these are trivial to derive, but if so, the authors should at least give some idea of how this is to be done. If not, the authors should include those derivations for completeness. The section on benchmarking (Section D.1) does a good job describing the various experiments performed and how they differ. However, little analysis or interpretation is provided for many of them, in particular for those under the subheading “Other models.” How do scalar, vector, and mixed vector fields perform and why? Four different variants of “norm-RELU” are defined and benchmarked. Why do they perform differently and should we prefer a certain one over the others for the task at hand. The authors also mention the ELU non-linearity, which is not defined anywhere in the text. Some explanation of the “competitive” or “final” runs would also be welcome. Why was this architecture not used on the other benchmarks? How is it expected to perform better? Finally, there are several minor errors: - p. 7: “don't” → "do not”, “a bad choices” → “a bad choice” or “bad choices”, “can not” → “cannot”, “perform very bad” → “perform very badly”. - p. 8: The acronym “SOTA” has not been defined; “allows to adapt” → “allows us to adapt”. - p. 14: Missing period after “in section C.4.4” - p. 15: “real valued” → “real-valued”, “won't” → “will not” - p. 17–18: Missing spaces after periods in figure captions. Missing commas after basis elements in table. - p. 28–29: “Invariant Projection”, “State of The Art”, and “Convolution Block” should not be capitalized. - p. 28: Space between “scalar fields” and “(i.e.”; “preserve trivial fields” → “preserves trivial fields”, “don't” → “do not”. - p. 29: “5e-4” should be written out as “5·10¯⁴” or formatted correctly (in LaTeX this would be done as “$5\mathrm{e}{-4}$”).

[Author Response · NeurIPS 2019]

We thank the reviewers for their detailed feedback and suggestions. We are happy to see that R2/R3 appreciate our
"unified" description and implementation of E(2)-equivariant CNNs in an "umbrella framework" and their "extremely
thorough" evaluation. As R1 is concerned about the significance and novelty of our contributions in general, we will
clarify and accentuate our contributions as summarized below.

**R1 - Incremental work:** Indeed the idea of E(2)-equivariant CNNs is not new and we build on the well known frame-
work of steerable CNNs [1,2,3,4,5]. However, the *theoretical* framework (Sec. 2.1-2.3) was so far instantiated only for
a few choices of groups and representations [1,2,5] while a fully general solution of this important model class was still
missing. The group restriction operation (exploiting local symmetries), the possibility to build hybrid models (operating
on different field types simultaneously) and a large-scale benchmark study are also novel to the literature.

11
**R1 - Significance of the general solution of the kernel constraint:**
• Our general solution yields a unified description and implementation of a large body of E(2)-equivariant models
[1,6,7,8,9,10,12,13,22,23,24] whose specific interrelations were so far left unclear.
• While these models were proven to be equivariant, most of them lacked a proof of implementing the most general
equivariant mapping between the corresponding field types. By proving the completeness of our general solution we
add these missing pieces. In the case of Harmonic Networks [12] we found that the solution space was incomplete.
• Our solution yields an off-the-shelf method for convolutions between arbitrary field types which were not being
considered before. It describes e.g. convolutions from regular fields to vector fields which might become important
e.g. for the processing of optical flow or wind fields. A unified implementation enables application focused
researchers to start using equivariant models without needing to take the hurdle of deriving the appropriate mappings.
• Beyond planar CNNs, the solved kernel space constraint applies exactly to Gauge Equivariant CNNs [5] on 2-dim
manifolds. In contrast to the plane $\mathbb{R}^2$, the structure group of Riemannian manifolds can in general *not* be reduced
further than O(2). A general solution of the kernel space constraint is therefore strictly necessary for such models.
We agree with R1 that these points should be presented more clearly. We updated the paper accordingly and further
added two sections discussing the incompleteness of Harmonic Networks and the relations to Gauge CNNs.

**R1 - Significance of empirical evaluation:** We want to point out that the significance of the conducted experiments
goes beyond "showing that symmetries in the dataset are important". In particular:
• We benchmark 44 models (30 new ones) with different levels of equivariance, group representations and nonlineari-
ties against each other (Tab. 9, supplementary). This experiment is of great importance for the field of equivariant
deep learning since a direct comparison of the E(2)-equivariant model zoo was missing before our submission.
• The benchmark study was repeated on three transformed MNIST versions (standard, SO(2) and O(2)) which was not
being done before. This allows to separate the contributions of global and local symmetries to the final performances.
• We evaluate the new group restriction at different depths on MNIST-rot (Tab. 10, suppl.) and on CIFAR-10/100
(Tab. 3) to support our hypothesis that exploiting local symmetries is beneficial even if not being present globally.
• Two (now three) models on MNIST-rot are presented which beat the previous SOTA (Tab. 2).
• Our CIFAR-10/100 models, trained with the auto-augment (AA) policy of [21], significantly outperform the AA
baseline (Tab. 3). This shows the benefit of equivariance even when powerful, task-adapted augmentation is used.

**R1 - paper organization:** We presented the general theory of steerable CNNs (Subsec. 2.2/2.3) and our new
contributions (Subsec. 2.4-2.8) together in our theory section 2 to give a comprehensive and self-contained exposition.
In the final version we will clearly separate the new contributions from previous work.

**R2/R3 - background required:** While we tried to keep the model definitions self-contained, some knowledge in group
representation theory is required. Background material, including definitions of induced and restricted representations
as requested by R3, are added to the supplementary. We will further try to simplify the paper in general.

**R3 - details for only one particular group:** Our results in Subsec. 2.5-2.8 apply to a whole family of (orthogonal)
groups O(2), SO(2), $D_N$ and $C_N$. The general solution strategy in Subsec. 2.4 applies generally to any finite dimensional
unitary representation and thus gives a clear roadmap for deriving general solutions for other groups.

**R3 - improvements despite auto-augment (AA):** Augmentation typically applies geometric transformations globally
while GCNNs exploit local symmetries [5]. While GCNNs exploit symmetries by design, standard CNNs need to *learn*
augmented samples explicitly. The hypothesis space and parameter cost of GCNNs are thus greatly reduced which leads
to superior performances [1,2,5-10,12-14,17,18,22]. See Sec.2 and Sec.6, Par.1 in [7] for a more technical explanation.

**R3 - overfitting:** To prevent from overfitting we did not tune the hyperparameters in *any* of our experiments but simply
replaced the non-equivariant convolutions of the baseline models with G-steerable convolutions.

**R3 - new experiments:** We are running new experiments on STL-10 to 1) confirm that our previous results generalize to
higher resolutions and 2) investigate the improved sample complexity of GCNNs in a data ablation study. By replacing
conventional with steerable convolutions we 1) significantly improved upon the previous (supervised) SOTA and 2) find
our models to consistently outperform baselines on all dataset sizes tested so far (still gathering more samples).
An implementation on manifolds would indeed be interesting but would raise additional engineering questions which
we believe to deserve their own publication. Our implementation on $\mathbb{R}^2$ allows for using a regular grid and therefore an
investigation of steerable convolutions without the additional complications arising from discretizations of 2-manifolds.
We further found a bug in our CIFAR-10/100 models which now perform even (significantly) better.

[Meta-Review · NeurIPS 2019]

We had a discussion about this paper with the reviewers raising the question of novelty since the general topic of steerable neural networks (i.e., equivariant neural networks where the activations are defined as a function on a homogeneous space) has now been thoroughly explored in the literature. Also, Weiler et al have a paper on SE(3) equivariant nets, so the E(2) is arguably just a simpler variant. There are three reasons why I nonetheless recommend this paper for acceptance: 1. The kernel constraint conditions are meticulously worked out for a range of specific subgroups of O(2). One wonders whether this could be done more generally and whether it really requires so much tedious algebra. Nonetheless, this is something that practitioners are likely to find useful. 2. The paper presents extensive experimental results on a range of different architectural variants, which is informative and will be a good benchmark for future work on E(2) equivariant architectures. 3. Equivariance to rotation and translation is part of the holy grail of computer vision, so this paper has significant practical relevance.